# A saturated map of common genetic variants associated with human height

Common single-nucleotide polymorphisms (SNPs) are predicted to collectively explain 40–50% of phenotypic variation in human height, but identifying the specific variants and associated regions requires huge sample sizes[1]. Here, using data from a genome-wide association study of 5.4 million individuals of diverse ancestries, we show that 12,111 independent SNPs that are significantly associated with height account for nearly all of the common SNP-based heritability. These SNPs are clustered within 7,209 non-overlapping genomic segments with a mean size of around 90 kb, covering about 21% of the genome. The density of independent associations varies across the genome and the regions of increased density are enriched for biologically relevant genes. In out-of-sample estimation and prediction, the 12,111 SNPs (or all SNPs in the HapMap 3 panel[2]) account for 40% (45%) of phenotypic variance in populations of European ancestry but only around 10–20% (14–24%) in populations of other ancestries. Effect sizes, associated regions and gene prioritization are similar across ancestries, indicating that reduced prediction accuracy is likely to be explained by linkage disequilibrium and differences in allele frequency within associated regions. Finally, we show that the relevant biological pathways are detectable with smaller sample sizes than are needed to implicate causal genes and variants. Overall, this study provides a comprehensive map of specific genomic regions that contain the vast majority of common height-associated variants. Although this map is saturated for populations of European ancestry, further research is needed to achieve equivalent saturation in other ancestries.

Since 2007, genome-wide association studies (GWASs) have identified thousands of associations between common SNPs and height, mainly using studies with participants of European ancestry. The largest GWAS published so far for adult height focused on common variation and reported up to 3,290 independent associations in 712 loci using a sample size of up to 700,000 individuals[3]. Adult height, which is highly heritable and easily measured, has provided a larger number of common genetic associations than any other human phenotype. In addition, a large collection of genes has been implicated in disorders of skeletal growth, and these are enriched in loci mapped by GWASs of height in the normal range. These features make height an attractive model trait for assessing the role of common genetic variation in defining the genetic and biological architecture of polygenic human phenotypes.

As available sample sizes continue to increase for GWASs of common variants, it becomes important to consider whether these larger samples can 'saturate' or nearly completely catalogue the information that can be derived from GWASs. This question of completeness can take several forms, including prediction accuracy compared with heritability attributable to common variation, the mapping of associated genomic regions that account for this heritability, and whether increasing sample sizes continue to provide additional information about the identity of prioritized genes and gene sets. Furthermore, because most GWASs continue to be performed largely in populations

of European ancestry, it is necessary to address these questions of completeness in the context of multiple ancestries. Finally, some have proposed that, when sample sizes become sufficiently large, effectively every gene and genomic region will be implicated by GWASs, rather than certain subsets of genes and biological pathways being specified[4].

Here, using data from 5.4 million individuals, we set out to map common genetic associations with adult height, using variants catalogued in the HapMap 3 project (HM3), and to assess the saturation of this map with respect to variants, genomic regions and likely causal genes and gene sets. We identify significant variants, examine signal density across the genome, perform out-of-sample estimation and prediction analyses within studies of individuals of European ancestry and other ancestries and prioritize genes and gene sets as likely mediators of the effects on height. We show that this set of common variants reaches predicted limits for prediction accuracy within populations of European ancestry and largely saturates both the genomic regions associated with height and broad categories of gene sets that are likely to be relevant; future work will be required to extend prediction accuracy to populations of other ancestries, to account for rarer genetic variation and to more definitively connect associated regions with individual probable causal genes and variants.

An overview of our study design and analysis strategy is provided in Extended Data Fig. 1.

A list of authors and their affiliations appears online. ✉e-mail: l.yengo@imb.uq.edu.au; yokada@sg.med.osaka-u.ac.jp; A.R.Wood@exeter.ac.uk; peter.visscher@uq.edu.au; Joel.Hirschhorn@childrens.harvard.edu

## Table 1 Summary of results from within-ancestry and trans-ancestry GWAS meta-analyses

| Cohort ancestry or ethnic group | Number of studies | Max $n$ (mean $n$) | Number of GWS COJO SNPs ($P_{GWAS} < 5 \times 10^{-8}$) | Number of GWS loci (35 kb) | Cumulative length of non-overlapping GWS loci in Mb (% of genome) |
|---|---|---|---|---|---|
| European (EUR) | 173 | 4,080,687 (3,612,229) | 9,863 (8,382) | 6,386 | 552.5 (18.4%) |
| East Asian (EAS) | 56 | 472,730 (320,570) | 918 (807) | 821 | 60.5 (2.0%) |
| Hispanic (HIS) | 11 | 455,180 (431,645) | 1,511 (1,195) | 1,373 | 101.0 (3.3%) |
| African (AFR) | 29 | 293,593 (222,981) | 453 (404) | 412 | 30.4 (1.0%) |
| South Asian (SAS) | 12 | 77,890 (59,420) | 69 (65) | 66 | 4.7 (0.2%) |
| Trans-ancestry meta-analysis (META_FE) | 281 | 5,314,291* (4,611,160) | 12,111 (9,920) | 7,209 | 647.5 (21.6%) |

$n$ denotes the sample size for each SNP. GWS: genome-wide significant ($P < 5 \times 10^{-8}$). COJO SNPs: near-independent GWS SNPs identified using an approximate COJO analysis implemented in the GCTA software. $P_{GWAS}$: $P$ value from a marginal association test. GWS loci were defined as genomic regions centred around each GWS SNP and including all SNPs within 35 kb on each side of the lead GWS SNP. Overlapping GWS loci were merged so that the number and cumulative length of GWS loci are calculated on non-overlapping GWS loci. The percentage of the genome covered was calculated by dividing the cumulative of GWS loci by 3,039 Mb (the approximated length of the human genome).

*The number of individuals in the trans-ancestry meta-analysis ($n = 5,314,291$) is smaller than the sum of ancestry-group-specific meta-analyses ($n = 5,380,080$) because of variation in per-SNP sample sizes for SNPs included in the final analysis.

## Meta-analysis identifies 12,111 height-associated SNPs

We performed genetic analysis of up to 5,380,080 individuals from 281 studies from the GIANT consortium and 23andMe. Supplementary Fig. 1 represents projections of these 281 studies onto principal components reflecting differences in allele frequencies across ancestry groups in the 1000 Genomes Project (1KGP)[5]. Altogether, our discovery sample includes 4,080,687 participants of predominantly European ancestries (75.8% of total sample); 472,730 participants with predominantly East Asian ancestries (8.8%); 455,180 participants of Hispanic ethnicity with typically admixed ancestries (8.5%); 293,593 participants of predominantly African ancestries—mostly African American individuals with admixed African and European ancestries (5.5%); and 77,890 participants of predominantly South Asian ancestries (1.4%). We refer to these five groups of participants or cohorts as EUR, EAS, HIS, AFR and SAS, respectively, while recognizing that these commonly used groupings oversimplify the actual genetic diversity among participants. Cohort-specific information is provided in Supplementary Tables 1–3. We tested the association between standing height and 1,385,132 autosomal bi-allelic SNPs from the HM3 tagging panel[2], which contains more than 1,095,888 SNPs with a minor allele frequency (MAF) greater than 1% in each of the five ancestral groups included in our meta-analysis. Supplementary Fig. 2 shows the frequency and imputation quality distribution of HM3 SNPs across all five groups of cohorts.

We first performed separate meta-analyses in each of the five groups of cohorts. We identified 9,863, 1,511, 918, 453 and 69 quasi-independent genome-wide significant (GWS; $P < 5 \times 10^{-8}$) SNPs in the EUR, HIS, EAS, AFR and SAS groups, respectively (Table 1 and Supplementary Tables 4–8). Quasi-independent associations were obtained after performing approximate conditional and joint (COJO) multiple-SNP analyses[6], as implemented in GCTA[7] (Methods). Supplementary Note 1 presents sensitivity analyses of these COJO results, highlights biases due to relatively long-range linkage disequilibrium (LD) in admixed AFR and HIS individuals[8] (Supplementary Fig. 3), and shows how to correct those biases by varying the GCTA input parameters (Supplementary Fig. 4). Moreover, previous studies have shown that confounding due to population stratification may remain uncorrected in large GWAS meta-analyses[9,10]. Therefore, we specifically investigated confounding effects in all ancestry-specific GWASs, and found that our results are minimally affected by population stratification (Supplementary Note 2 and Supplementary Figs. 5–7).

To compare results across the five groups of cohorts, we examined the genetic and physical colocalization between SNPs identified in the largest group (EUR) with those found in the other (non-EUR) groups. We found that more than 85% of GWS SNPs detected in the non-EUR groups are in strong LD ($r_{LD}^2 > 0.8$) with at least one variant reaching

marginal genome-wide significance ($P_{GWAS} < 5 \times 10^{-8}$) in EUR (Supplementary Tables 5–8). Furthermore, more than 91% of associations detected in non-EUR meta-analyses fall within 100 kb of a GWS SNP identified in EUR (Extended Data Fig. 2). By contrast, a randomly sampled HM3 SNP (matched with GWS SNPs identified in non-EUR meta-analyses on 24 functional annotations; Methods) falls within 100 kb of a EUR GWS SNP 55% of the time on average (s.d. = 1% over 1,000 draws). Next, we quantified the cross-ancestry correlation of marginal allele substitution effects ($\rho_b$) at GWS SNPs for all pairs of ancestry groups. We estimated $\rho_b$ using five subsets of GWS SNPs identified in each of the ancestry groups, which also reached marginal genome-wide significance in at least one group. After correction for winner's curse[11,12], we found that $\rho_b$ ranged between 0.64 and 0.99 across all pairs of ancestry groups and all sets of GWS SNPs (Supplementary Figs. 8–12). We also extended the estimation of $\rho_b$ for SNPs that did not reach genome-wide significance and found that $\rho_b > 0.5$ across all comparisons (Supplementary Fig. 13). Thus, the observed GWS height associations are substantially shared across major ancestral groups, consistent with previous studies based on smaller sample sizes[13,14].

To find signals that are specific to certain groups, we tested whether any individual SNPs detected in non-EUR GWASs are conditionally independent of signals detected in EUR GWASs. We fitted an approximate joint model that includes GWS SNPs identified in EUR and non-EUR, using LD reference panels specific to each ancestry group. After excluding SNPs in strong LD ($r_{LD}^2 > 0.8$ in either ancestry group), we found that 2, 17, 49 and 63 of the GWS SNPs detected in SAS, AFR, EAS and HIS GWASs, respectively, are conditionally independent of GWS SNPs identified in EUR GWASs (Supplementary Table 9). On average, these conditionally independent SNPs have a larger MAF and effect size in non-EUR than in EUR cohorts, which may have contributed to an increased statistical power of detection. The largest frequency difference relative to EUR was observed for rs2463169 (height-increasing G allele frequency: 23% in AFR versus 84% in EUR) within the intron of PAWR, which encodes the prostate apoptosis response-4 protein. Of note, rs2463169 is located within the 12q21.2 locus, where a strong signal of positive selection in West African Yoruba populations was previously reported[15]. The estimated effect at rs2463169 is $\beta \approx 0.034$ s.d. per G allele in AFR versus $\beta \approx -0.002$ s.d. per G allele in EUR, and the $P$ value of marginal association in EUR is $P_{EUR} = 0.08$, suggesting either a true difference in effect size or nearby causal variant(s) with differing LD to rs2463169.

Given that our results show a strong genetic overlap of GWAS signals across ancestries, we performed a fixed-effect meta-analysis of all five ancestry groups to maximize statistical power for discovering associations due to shared causal variants. The mean Cochran's heterogeneity

*Q*-statistic is around 34% across SNPs, which indicates moderate heterogeneity of SNP effects between ancestries. The mean chi-square association statistic in our fixed-effect meta-analysis (hereafter referred to as META$_{FE}$) is around 36, and around 18% of all HM3 SNPs are marginally GWS. Moreover, we found that allele frequencies in our META$_{FE}$ were very similar to that of EUR (mean fixation index of genetic differentiation ($F_{ST}$) across SNPs between EUR and META$_{FE}$ is around 0.001), as expected because our META$_{FE}$ consists of more than 75% EUR participants and around 14% participants with admixed European and non-European ancestries that is, HIS and AFR). To further assess whether LD in our META$_{FE}$ could be reasonably approximated by the LD from EUR, we performed an LD score regression[16] analysis of our META$_{FE}$ using LD scores estimated in EUR. In this analysis, we focused on the attenuation ratio statistic ($R_{LDSC-EUR}$), for which large values can also indicate strong LD inconsistencies between a given reference and GWAS summary statistics. A threshold of $R_{LDSC} > 20\%$ was recommended by the authors of the LDSC software as a rule-of-thumb to detect such inconsistencies. Using EUR LD scores in the GWAS of HIS, which is the non-EUR group that is genetically closest to EUR ($F_{ST} \approx 0.02$), yields an estimated $R_{LDSC-EUR}$ of around 25% (standard error (s.e.) 1.8%), consistent with strong LD differences between HIS and EUR. By contrast, in our META$_{FE}$, we found an estimated $R_{LDSC-EUR}$ of around 4.5% (s.e. 0.8%), which is significantly lower than 20% and not statistically different from 3.8% (s.e. 0.8%) in our EUR meta-analysis. Furthermore, we show in Supplementary Note 1 that using a composite LD reference containing samples from various ancestries (with proportions matching that in our META$_{FE}$) does not improve signal detection over using an EUR LD reference. Altogether, these analyses suggest that LD in our META$_{FE}$ can be reasonably approximated by LD from EUR.

We therefore proceeded to identify quasi-independent GWS SNPs from the multi-ancestry meta-analysis by performing a COJO analysis of our META$_{FE}$, using genotypes from around 350,000 unrelated EUR participants in the UK Biobank (UKB) as an LD reference. We identified 12,111 quasi-independent GWS SNPs, including 9,920 (82%) primary signals with a GWS marginal effect and 2,191 secondary signals that only reached GWS in a joint regression model (Supplementary Table 10). Figure 1 represents the relationship between frequency and joint effect sizes of minor alleles at these 12,111 associations. Of the GWS SNPs obtained from the non-EUR meta-analyses above that were conditionally independent of the EUR GWS SNPs, 0/2 in SAS, 5/17 in AFR, 27/49 in EAS and 27/63 in HIS were marginally significant in our META$_{FE}$ (Supplementary Table 9), and 24 of those (highlighted in Fig. 2) overlapped with our list of 12,111 quasi-independent GWS SNPs.

We next sought to replicate the 12,111 META$_{FE}$ signals using GWAS data from 49,160 participants in the Estonian Biobank (EBB). We first re-assessed the consistency of allele frequencies between our META$_{FE}$ and the EBB set. We found a correlation of allele frequencies of around 0.98 between the two datasets and a mean $F_{ST}$ across SNPs of around 0.005, similar to estimates that were obtained between populations from the same continent. Of the 12,111 GWS SNPs identified through our COJO analysis, 11,847 were available in the EBB dataset, 97% of which (11,529) have a MAF greater than 1% (Supplementary Table 10). Given the large difference in sample size between our discovery and replication samples, direct statistical replication of individual associations at GWS is not achievable for most SNPs identified (Extended Data Fig. 3a). Instead, we assessed the correlation of SNP effects between our discovery and replication GWASs as an overall metric of replicability[3,17]. Among the 11,529 out of 11,847 SNPs that had a MAF greater than 1% in the EBB, we found a correlation of marginal SNP effects of $\rho_b = 0.93$ (jackknife standard error; s.e. 0.01) and a correlation of conditional SNP effects using the same LD reference panel of $\rho_b = 0.80$ (s.e. 0.03; Supplementary Fig. 14). Although we had limited power to replicate associations with 238 GWS variants that are rare in the EBB (MAF < 1%), we found, consistent with expectations (Methods and Extended Data Fig. 3b), that 60% of them had a marginal SNP effect that was sign-consistent

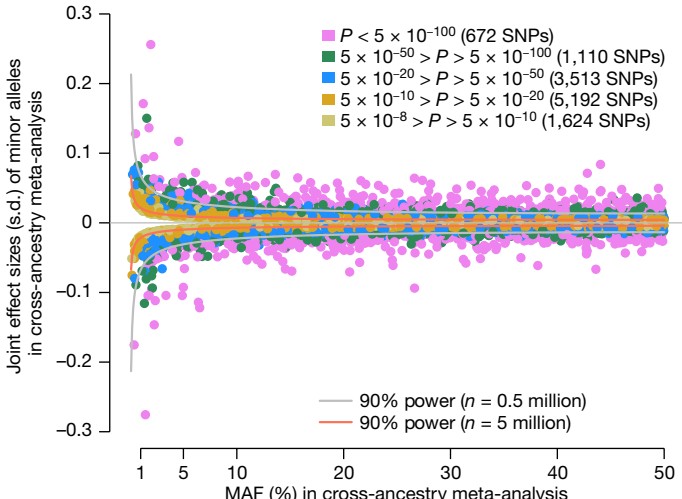

**Fig. 1 | Relationship between frequency and estimated effect sizes of minor alleles.** Each dot represents one of the 12,111 quasi-independent GWS SNPs that were identified in our cross-ancestry GWAS meta-analysis. Data underlying this figure are available in Supplementary Table 10. SNP effect estimates (*y* axis) are expressed in height standard deviation (s.d.) per minor allele as defined in our cross-ancestry GWAS meta-analysis. SNPs were stratified in five classes according to their *P* value (*P*) of association. We show two curves representing the theoretical relationship between frequency and expected magnitude of SNP effect detectable at $P < 5 \times 10^{-8}$ with a statistical power of 90%. Statistical power was assessed under two experimental designs with sample sizes equal to $n = 0.5$ million and $n = 5$ million.

with that from our discovery GWAS (Fisher's exact test; $P = 0.001$). The proportion of sign-consistent SNP effects was greater than 75% (Fisher's exact test; $P < 10^{-50}$) for variants with a MAF greater than 1%− also consistent with expectations (Extended Data Fig. 3b). Altogether, our analyses demonstrate the robustness of our findings and show their replicability in an independent sample.

## Genomic distribution of height-associated SNPs

To examine signal density among the 12,111 GWS SNPs detected in our META$_{FE}$, we defined a measure of local density of association signals for each GWS SNP on the basis of the number of additional independent associations within 100 kb (Supplementary Fig. 15). Supplementary Fig. 16 shows the distributions of signal density for GWS SNPs identified in each ancestry group and in our META$_{FE}$. We observed that 69% of GWS SNPs shared their location with another associated, conditionally independent, GWS SNP (Fig. 2). The mean signal density across the entire genome is 2.0 (s.e. 0.14), consistent with a non-random genomic distribution of GWS SNPs. Next, we evaluated signal density around 462 autosomal genes curated from the Online Mendelian Inheritance in Man (OMIM) database[18] as containing pathogenic mutations that cause syndromes of abnormal skeletal growth ('OMIM genes'; Methods and Supplementary Table 11). We found that a high density of height-associated SNPs is significantly correlated with the presence of an OMIM gene nearby[19,20] (enrichment fold of OMIM gene when density is greater than 1: 2.5×; $P < 0.001$; Methods and Extended Data Fig. 4a). Notably, the enrichment of OMIM genes almost linearly increases with the density of height-associated SNPs (Extended Data Fig. 4b). Thus, these 12,111 GWS SNPs nonrandomly cluster near each other and near known skeletal growth genes.

The largest density of conditionally independent associations was observed on chromosome 15 near *ACAN*, a gene mutated in short stature and skeletal dysplasia syndromes, where 25 GWS SNPs co-localize within 100 kb of one another (Fig. 2 and Supplementary Fig. 17). We show in

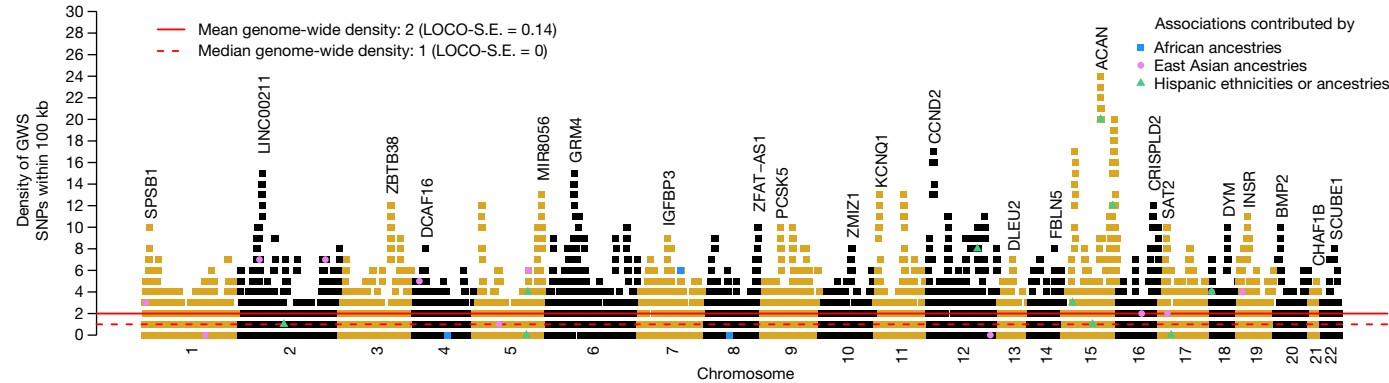

**Fig. 2 | Brisbane plot showing the genomic density of independent genetic associations with height.** Each dot represents one of the 12,111 quasi-independent GWS ($P < 5 \times 10^{-8}$) height-associated SNPs identified using approximate COJO analyses of our cross-ancestry GWAS meta-analysis. Data underlying this figure are available in Supplementary Table 10. GWS SNPs with the largest density on each chromosome were annotated with the closest gene. We highlight 24 of 12,111 associations that are mainly contributed by groups of non-European ancestry (3 from African ancestries, 10 from Hispanic ethnicities or ancestries and 11 from East Asian ancestries). The full list of height-associated SNPs detected in groups of non-European ancestry and

independent of associations detected in European ancestry GWASs is reported in Supplementary Table 9. Signal density was calculated for each associated SNP as the number of other independent associations within 100 kb. A density of 1 means that a GWS COJO SNP shares its location with another independent GWS COJO SNP within less than 100 kb. The mean signal density across the genome is 2 and the median signal density is 1 (s.e. 0.14 and 0.0, respectively). The s.e. values were calculated using a leave-one-chromosome-out jackknife approach (LOCO-S.E.). SNPs that did not reach genome-wide significance are not represented on the figure.

Supplementary Note 3 and Extended Data Fig. 5a–d, using haplotype- and simulation-based analyses, that a multiplicity of independent causal variants is the most likely explanation of this observation. We also found that signal density is partially explained by the presence of a recently identified[21,22] height-associated variable-number tandem repeat (VNTR) polymorphism at this locus (Supplementary Note 3). In fact, the 25 independent GWS SNPs clustered within 100 kb of rs4932198 explain more than 40% of the VNTR length variation in multiple ancestries (Extended Data Fig. 5e), and an additional approximately 0.24% ($P = 8.7 \times 10^{-55}$) of phenotypic variance in EUR above what is explained by the VNTR alone (Extended Data Fig. 5f). Altogether, our conclusion is consistent with previous evidence of multiple types of common variation influencing height through *ACAN* gene function, involving multiple enhancers[23], missense variants[24] and tandem repeat polymorphisms[21,22].

## Variance explained by SNPs within identified loci

To quantify the proportion of height variance that is explained by GWS SNPs identified in our META_FE, we stratified all HM3 SNPs into two groups: SNPs in the close vicinity of GWS SNPs, hereafter denoted GWS loci; and all remaining SNPs. We defined GWS loci as non-overlapping genomic segments that contain at least one GWS SNP, such that GWS SNPs in adjacent loci are more than 2 × 35 kb away from each other (that is, a 35-kb window on each side). We chose this size window because it was predicted that causal variants are located within 35 kb of GWS SNPs with a probability greater than 80% (ref. [25]). Accordingly, we grouped the 12,111 GWS SNPs identified in our META_FE into 7,209 non-overlapping loci (Supplementary Table 12) with lengths ranging from 70 kb (for loci containing only one signal) to 711 kb (for loci containing up to 25 signals). The average length of GWS loci is around 90 kb (s.d. 46 kb). The cumulative length of GWS loci represents around 647 Mb, or about 21% of the genome (assuming a genome length of around 3,039 Mb)[26].

To estimate the fraction of heritability that is explained by common variants within the 21% of the genome overlapping GWS loci, we calculated two genomic relationship matrices (GRMs)—one for SNPs within these loci and one for SNPs outside these loci—and then used both matrices to estimate a stratified SNP-based heritability ($h^2_{SNP}$) of height in eight independent samples of all five population groups represented in our META_FE (Fig. 3 and Methods). Altogether, our stratified

estimation of SNP-based heritability shows that SNPs within these 7,209 GWS loci explain around 100% of $h^2_{SNP}$ in EUR and more than 90% of $h^2_{SNP}$ across all non-EUR groups, despite being drawn from less than 21% of the genome (Fig. 3). We also varied the window size used to define GWS loci and found that 35 kb was the smallest window size for which this level of saturation of SNP-based heritability could be achieved (Supplementary Fig. 18).

To further assess the robustness of this key result, we tested whether the 7,209 height-associated GWS loci are systematically enriched for trait heritability. We chose body-mass index (BMI) as a control trait, given its small genetic correlation with height ($r_g = −0.1$, ref. [27]) and found no significant enrichment of SNP-based heritability for BMI within height-associated GWS loci (Supplementary Fig. 19). Furthermore, we repeated our analysis using a random set of SNPs matched with the 12,111 height-associated GWS SNPs on EUR MAF and LD scores. We found that this control set of SNPs explained only around 27% of $h^2_{SNP}$ for height, consistent with the proportion of SNPs within the loci defined by this random set of SNPs (Supplementary Figs. 18 and 19). Finally, we extended our stratified estimation of SNP-based heritability to all well-imputed common SNPs (that is, beyond the HM3 panel) and found, consistently across population groups, that although more genetic variance can be explained by common SNPs that are not included in the HM3 panel, all information remains concentrated within these 7,209 GWS loci (Extended Data Fig. 6). Thus, with this large GWAS, nearly all of the variability in height that is attributable to common genetic variants can be mapped to regions comprising around 21% of the genome. Further work is required in cohorts of non-European ancestries to map the remaining 5–10% of the SNP-based heritability that is not captured within those regions.

## Out-of-sample prediction accuracy

We quantified the accuracy of multiple polygenic scores (PGSs) for height on the basis of GWS SNPs (hereafter referred to as PGS_GWS) and on the basis of all HM3 SNPs (hereafter referred to as PGS_HM3). PGS_GWS were calculated using joint SNP effects from COJO, and PGS_HM3 using joint effects calculated using the SBayesC method[28] (Methods). We denote $R^2_{GWS}$ and $R^2_{HM3}$ as the prediction accuracy of PGS_GWS and PGS_HM3, respectively. For conciseness, we also use the abbreviations PGS_GWS-X and PGS_HM3-X (and $R^2_{GWS-X}$ and $R^2_{HM3-X}$) to specify which GWAS

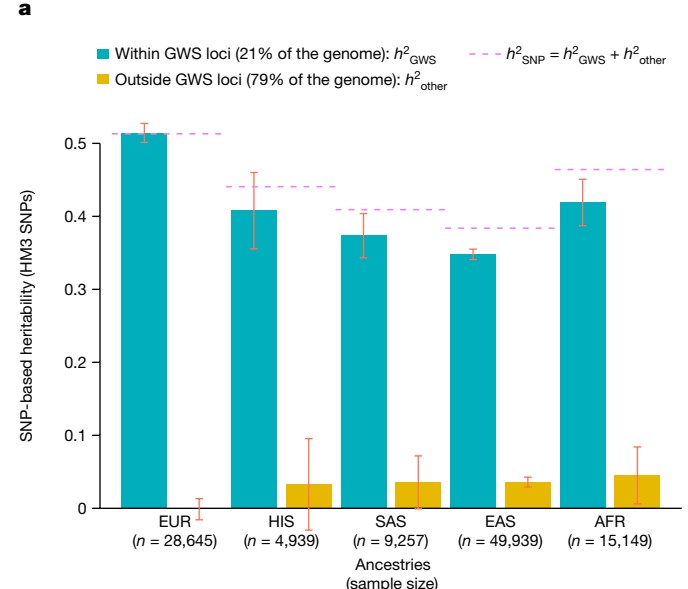

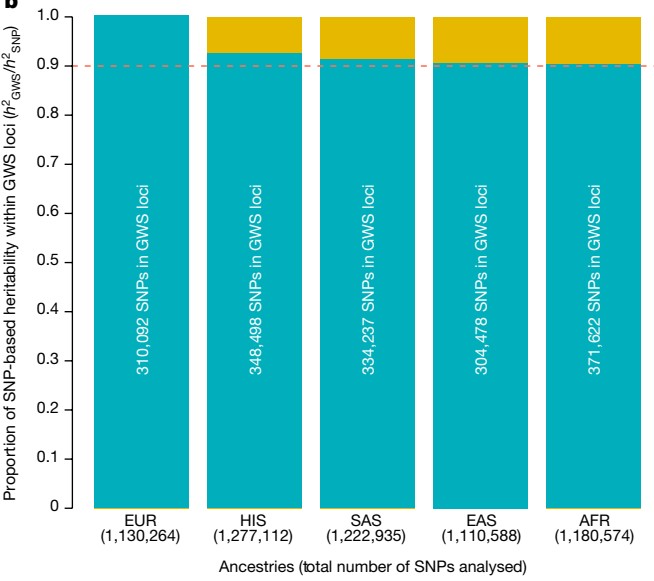

**Fig. 3 | Variance of height explained by HM3 SNPs within GWS loci.**
**a**, Stratified SNP-based heritability ($h^2_{SNP}$) estimates obtained after partitioning the genome into SNPs within 35 kb of a GWS SNP ('GWS loci' label) versus SNPs that are more than 35 kb away from any GWS SNP. Analyses were performed in samples of five different ancestries or ethnic groups: European (EUR: meta-analysis of UK Biobank (UKB) + Lifelines study), African (AFR: meta-analysis of UKB + PAGE study), East Asian (EAS: meta-analysis of UKB +

China Kadoorie Biobank), South Asian (SAS: UKB) and Hispanic (HIS: PAGE). Error bars represent standard errors. **b**, More than 90% of $h^2_{SNP}$ in all ancestries is explained by SNPs within GWS loci identified in this study. The cumulative length of non-overlapping GWS loci is around 647 Mb; that is, around 21% of the genome, assuming a genome length of around 3,039 Mb (ref. [26]). The proportion of HM3 SNPs in GWS loci is around 27%.

meta-analysis each PGS (and corresponding prediction accuracy) was trained from. For example, $PGS_{GWS-METAFE}$ refers to PGSs based on 12,111 GWS SNPs identified from our $META_{FE}$.

We first present results from $PGS_{GWS}$ across different ancestry groups. $PGS_{GWS-METAFE}$ yielded prediction accuracies greater than or equal to that of all other $PGS_{GWS}$ (Fig. 4a), partly reflecting sample size differences between ancestry-specific GWASs and also consistent with previous studies[29]. $PGS_{GWS-EUR}$ (based on 9,863 SNPs) was the second best of all $PGS_{GWS}$ across ancestry groups except in AFR. Indeed, $PGS_{GWS-AFR}$ (based on 453 SNPs) yielded an accuracy of 8.5% (s.e. 0.6%) in AFR individuals from UKB and PAGE; that is, significantly larger than the 5.9% (s.e. 0.6%) and 7.0% (s.e. 0.6%) achieved by $PGS_{GWS-EUR}$ in these two samples, respectively (Fig. 4a). $PGS_{GWS-METAFE}$ was the best of all $PGS_{GWS}$ in AFR participants with an accuracy $R^2_{GWS-METAFE} = (12.3\% + 9.9\%)/2 = 10.8\%$ (s.e. 0.5%) on average between UKB and PAGE (Fig. 4a). Across ancestry groups, the highest accuracy of $PGS_{GWS-METAFE}$ was observed in EUR participants ($R^2_{GWS-METAFE}$ ~40%; s.e. 0.6%) and the lowest in AFR participants from the UKB ($R^2_{GWS-METAFE} \approx 9.4\%$; s.e. 0.7%). Note that the difference in $R^2_{GWS-METAFE}$ between the EUR and AFR ancestry cohorts is expected because of the over-representation of EUR in our $META_{FE}$, and consistent with a relative accuracy ($R^2_{GWS-METAFE}$ in AFR)/($R^2_{GWS-METAFE}$ in EUR) of around 25% that was previously reported[30]. We extended analyses of $PGS_{GWS}$ to PGS based on SNPs identified with COJO at lower significance thresholds (Extended Data Fig. 7). As in previous studies[3,20], the inclusion of sub-significant SNPs increased the accuracy of ancestry-specific PGSs. However, lowering the significance thresholds in our $META_{FE}$ mostly improved accuracy in EUR (from 40% to 42%), whereas it slightly decreased the accuracy in AFR.

Overall, ancestry-specific $PGS_{HM3}$ consistently outperform their corresponding $PGS_{GWS}$ in most ancestry-groups. However, $PGS_{HM3}$ was sometimes less transferable across ancestry groups than $PGS_{GWS}$, in particular in AFR and HIS individuals from PAGE. In EUR, $PGS_{HM3}$ reaches an accuracy of 44.7% (s.e. 0.6%), which is higher than previously published SNP-based predictors of height derived from individual-level

data[31–33] and from GWAS summary statistics[28,34,35] across various experimental designs (different SNP sets, different sample sizes and so on). Finally, the largest improvement of $PGS_{HM3}$ over $PGS_{GWS}$ was observed in AFR individuals from the PAGE study ($R^2_{GWS-AFR} = 8.5\%$ versus $R^2_{HM3} = 15.4\%$; Fig. 4a) and the UKB ($R^2_{GWS-AFR} = 8.5\%$ versus $R^2_{HM3} = 14.4\%$; Fig. 4a).

Furthermore, we sought to evaluate the prediction accuracy of PGSs relative to that of familial information as well as the potential improvement in accuracy gained from combining both sources of information. We analysed 981 unrelated EUR trios (that is, two parents and one child) and 17,492 independent EUR sibling pairs from the UKB, who were excluded from our $META_{FE}$. We found that height of any first-degree relative yields a prediction accuracy between 25% and 30% (Fig. 4b). Moreover, the accuracy of the parental average is around 43.8% (s.e. 3.2%), which is lower than yet not significantly different from the accuracy of $PGS_{HM3-EUR}$ in EUR. In addition, we found that a linear combination of the average height of parents and of the child's PGS yields an accuracy of 54.2% (s.e. 3.2%) with $PGS_{GWS-EUR}$ and 55.2% (s.e. 3.2%) with $PGS_{HM3-EUR}$. This observation reflects the fact that PGSs can explain within-family differences between siblings, whereas average parental height cannot. To show this empirically, we estimate that our PGSs based on GWS SNPs explain around 33% (s.e. 0.7%) of height variance between siblings (Methods). Finally, we show that the optimal weighting between parental average and PGS can be predicted theoretically as a function of the prediction accuracy of the PGS, the full narrow sense heritability and the phenotypic correlation between spouses (Supplementary Note 4 and Supplementary Fig. 20).

In summary, the estimation of variance explained and prediction analyses in samples with European ancestry show that the set of 12,111 GWS SNPs accounts for nearly all of $h^2_{SNP}$, and that combining SNP-based PGS with family history significantly improves prediction accuracy. By contrast, both estimation and prediction results show clear attenuation in samples with non-European ancestry, consistent with previous studies[30,36–38].

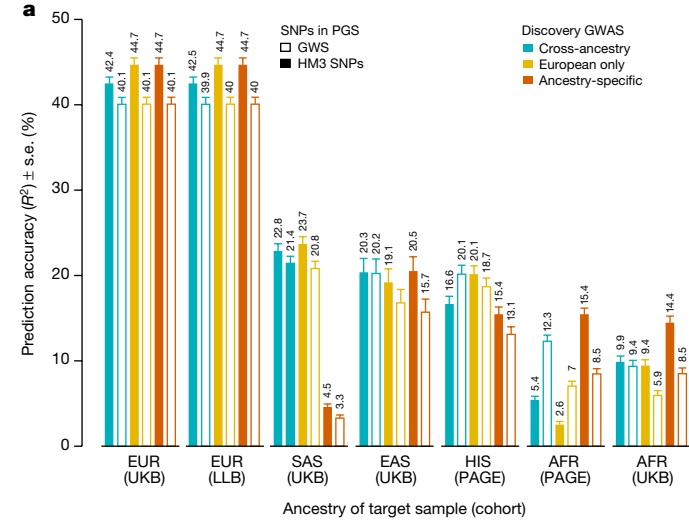

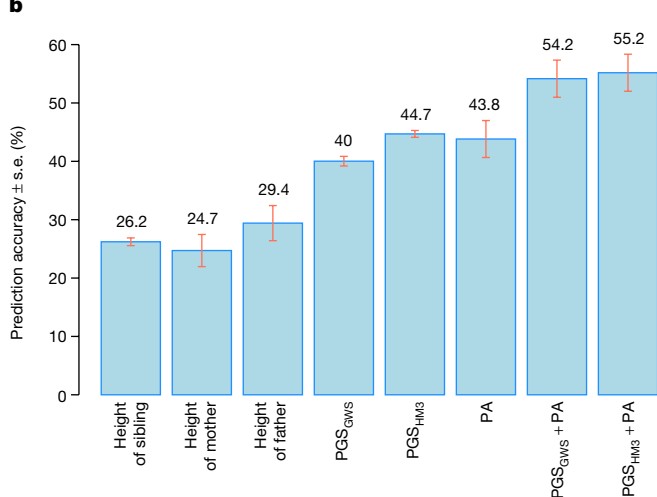

**Fig. 4 | Accuracy of PGSs within families and across ancestries.** Prediction accuracy ($R^2$) was measured as the squared correlation between PGS and actual height adjusted for age, sex and 10 genetic principal components. **a**, Accuracy of PGSs assessed in participants of five different ancestry groups: European (EUR) from the UKB ($n = 14,587$) and the Lifelines Biobank ($n = 14,058$); South Asian (SAS; $n = 9,257$) from UKB; East Asian (EAS; $n = 2,246$) from UKB; Hispanic (HIS; $n = 5,798$) from the PAGE study; and admixed African (AFR) from UKB ($n = 6,911$) and PAGE ($n = 8,238$). PGSs used for prediction, in **a**, are based on GWS SNPs or around 1.1 million HM3 SNPs. When using all HapMap 3 SNPs, SNP effects were calculated using the SBayesC method (Methods), whereas PGSs based on GWS SNPs used joint SNP effects estimated using the COJO method

(Methods). Both SBayesC and COJO were applied to (1) our cross-ancestry meta-analysis (turquoise bar); (2) our EUR meta-analysis (yellow bar); and (3) each ancestry-specific meta-analysis (red bar). **b**, Squared correlation of height between EUR participants in UKB and their first-degree relatives, and the accuracy of a predictor combining PGS (denoted PGS$_{GWS}$, as based on GWS SNPs) and familial information. The accuracies of PGS$_{GWS}$ and PGS$_{HM3}$ shown in **b** are the average of the respective accuracies of these PGSs in EUR participants from UKB and the Lifelines Biobank as shown in **a**. Sibling correlation was calculated in 17,492 independent EUR sibling pairs from the UKB and parent–offspring correlations in 981 EUR unrelated trios (that is, two parents and one child) from the UKB. PA, parental average.

## GWAS discoveries, sample size and ancestry diversity

Our large study offers the opportunity to quantify empirically how much increasing GWAS sample sizes and ancestry diversity affects the discovery of variants, genes and biological pathways. To address this question, we re-analysed three previously published GWASs of height[3,19,20] and also down-sampled our meta-analysis into four subsets (including our EUR and META$_{FE}$ GWASs). Altogether, we analysed seven GWASs with a sample size increasing from around 0.13 million up to around 5.3 million individuals (Table 2).

For each GWAS, we quantified eight metrics grouped into four variant- and locus-based metrics (number of GWS SNPs; number of GWS loci; prediction accuracy ($R^2_{GWS}$) of PGS based on GWS SNPs; and proportion of the genome covered by GWS loci), a functional-annotation-based metric (enrichment statistics from stratified LDSC[39,40]), two gene-based metrics (number of genes prioritized by summary-data-based Mendelian randomization[41] (SMR; Methods) and proximity of variants with OMIM genes) and a gene-set-based metric (enrichment within clusters of gene sets or pathways). Overall, we found different patterns for the relationship between those metrics and GWAS sample size and ancestry composition, consistent with varying degrees of saturation achieved at different sample sizes.

We observed the strongest saturation for the gene-set and functional-annotation metrics, which capture how well general biological functions can be inferred from GWAS results using currently available computational methods. Using two popular gene-set prioritization methods (DEPICT[42] and MAGMA[43]), we found that the same broad clusters of related gene sets (including most of the clusters enriched for OMIM genes) are prioritized at all GWAS sample sizes (Supplementary Fig. 21, Extended Data Fig. 8, Supplementary Tables 13–15 and Supplementary Note 5). Similarly, stratified LDSC estimates of heritability enrichment within 97 functional annotations also remain stable across the range of sample sizes (Extended Data Fig. 9). Overall, we found no significant improvement for all these higher-level metrics from adding non-EUR

samples to our analyses. The latter observation is consistent with other analyses showing that GWASs expectedly implicate similar biology across major ancestral groups (Supplementary Note 5 and Supplementary Fig. 22).

For the gene-level metric, the excess in the number of OMIM genes that are proximate to a GWS SNP (compared with matched sets of random genes) plateaus at sample sizes of larger than 1.5 million, whereas the relative enrichment of GWS SNPs near OMIM genes first decreases with sample size, then plateaus when $n$ is greater than 1.5 million (Supplementary Fig. 23a–c). Notably, the decrease observed for $n$ values of less than 1.5 million reflects the preferential localization of larger effect variants (those identified with smaller sample sizes) closer to OMIM genes (Supplementary Fig. 23d) and, conversely, that more recently identified variants with smaller effects tend to localize further away from OMIM genes (Supplementary Fig. 23e). We also investigated the number of genes prioritized using SMR (hereafter referred to as SMR genes; Methods) using expression quantitative trait loci (eQTLs) as genetic instruments (Supplementary Table 16) as an alternative gene-level metric and found it to saturate for $n$ values greater than 4 million (Supplementary Fig. 23f). Note that saturation of SMR genes is partly affected by the statistical power of current eQTL studies, which do not always survey biologically relevant tissues and cell types for height. Therefore, we can expect more genes to be prioritized when integrating GWAS summary statistics from this study with those from larger eQTL studies that may be available in the future and may involve more tissue types. Gene-level metrics were also not substantially affected by adding non-EUR samples, again consistent with broadly similar sets of genes affecting height across ancestries.

At the level of variants and genomic regions, we saw a steady and almost linear increase in the number of GWS SNPs as a function of sample size, as previously reported[44]. However, given that newly identified variants tend to cluster near ones identified at smaller sample sizes, we also saw a saturation in the number of loci identified for $n$ values greater than 2.5 million, where the upward trend starts to weaken (Supplementary

Table 2 Overview of five European-ancestry GWASs re-analysed in our study to quantify the relationship between sample size and discovery

| Down-sampled GWAS | Max $n$ (mean $n$) | Number of GWS COJO SNPs | Percentage of the genome covered by GWS loci (35 kb) (%) |
|---|---|---|---|
| Lango Allen et al. (2010)[19a] | 130,010 (128,942) | 240 | 0.5 |
| Wood et al. (2014)[20] | 241,724 (239,227) | 633 | 1.4 |
| Yengo et al. (2018)[3] | 695,648 (688,927) | 2,794 | 5.8 |
| GIANT-EUR (no 23andMe) | 1,632,839 (1,502,499) | 4,867 | 9.7 |
| 23andMe-EUR | 2,502,262 (2,498,336) | 7,020 | 13.6 |

Summary statistics from the three published GWASs were imputed using the ImpG-Summary software to maximize the coverage of HM3 SNPs (Methods). GWS loci are defined as in the legend of Table 1.

[a]Summary statistics from the Lango Allen et al. study[19], initially over-corrected for population stratification using a double genomic control correction, were re-inflated such that the LD score regression intercept estimated from re-inflated test statistics equals 1.

Fig. 24a). We found a similar pattern for the percentage of the genome covered by GWS loci, with the degree of saturation varying as a function of the window size used to define loci (Supplementary Fig. 24b). The observed saturation in PGS prediction accuracy (both within ancestry—that is, in EUR—and multi-ancestry) was more noticeable than that of the number and genomic coverage of GWS loci. In fact, increasing the sample size from 2.5 million to 4 million by adding another 1.5 million EUR samples increased the number of GWS SNPs from 7,020 to 9,863—that is, an increase of around 1.4-fold ((9,863 − 7,020)/7,020)— but the absolute increase in prediction accuracy is less than 2.7%. This improvement is mainly observed in EUR but remains lower than 1.3% in individuals of the EAS and AFR ancestry groups. However, adding another approximately 1 million participants of non-EUR improves the multi-ancestry prediction accuracy by more than 3.4% (Supplementary Fig. 24c), highlighting the value of including non-EUR populations.

Altogether, these analyses show that increasing the GWAS sample size not only increases the prediction accuracy, but also sheds more light on the genomic distribution of causal variants and, at all but the largest sample sizes, the genes proximal to these variants. By contrast, enrichment of higher-level, broadly defined biological categories such as gene sets and pathways and functional annotations can be identified using relatively small sample sizes ($n ≈ 0.25$ million for height). Of note, we confirm that increased genetic diversity in GWAS discovery samples significantly improves the prediction accuracy of PGSs in under-represented ancestries.

## Discussion

By conducting one of the largest GWASs so far in 5.4 million individuals, with a primary focus on common genetic variation, we have provided insights into the genetic architecture of height—including a saturated genomic map of 12,111 genetic associations for height. Consistent with previous studies[19,20], we have shown that signal density of associations (known and novel) is not randomly distributed across the genome; rather, associated variants are more likely to be detected around genes that have been previously associated with Mendelian disorders of growth. Furthermore, we observed a strong genetic overlap of association across cohorts with various ancestries. Effect estimates of associated SNPs are moderately to highly correlated (minimum = 0.64; maximum = 0.99), suggesting even larger correlations of effect sizes of underlying causal variants[13]. Moreover, although there are significant differences in power to detect an association between cohorts with European and non-European ancestries, most genetic associations for height observed in populations with non-European ancestry lie in close proximity and in linkage disequilibrium to associations identified within populations of European ancestry.

By increasing our experimental sample size to more than seven times that of previous studies, we have explained up to 40% of the inter-individual variation in height in independent European-ancestry samples using GWS SNPs alone, and more than 90% of $h^2_{SNP}$ across diverse populations when incorporating all common SNPs within 35 kb of GWS SNPs. This result highlights that future investigations of common (MAF > 1%) genetic variation associated with height in many ancestries will be most likely to detect signals within the 7,209 GWS loci that we have identified in the present study. A question for the future is whether rare genetic variants associated with height are also concentrated within the same loci. We provide suggestive evidence supporting this hypothesis from analysing imputed SNPs with 0.1% < MAF < 1% (Supplementary Note 6, Extended Data Fig. 10 and Supplementary Fig. 25). Our results are consistent with findings from a previous study[45], which showed across 492 traits a strong colocalization between common and rare coding variants associated with the same trait. Nevertheless, our conclusions remain limited by the relatively low performances of imputation in this MAF regime[46,47]. Therefore, large samples with whole-genome sequences will be required to robustly address this question. Such datasets are increasingly becoming available[48–50]. Separately, previous studies have reported a significant enrichment of height heritability near genes as compared to inter-genic regions (that is, >50 kb away from the start or stop genomic position of genes)[51]. Our findings are consistent with but not reducible to that observation, given that up to 31% of GWS SNPs identified in this study lie more than 50 kb away from any gene.

Our study provides a powerful genetic predictor of height based on 12,111 GWS SNPs, for which accuracy reaches around 40% (that is, 80% of $h^2_{SNP}$) in individuals of European ancestries and up to around 10% in individuals of predominantly African ancestries. Notably, we show using a previously developed method[38] that LD and MAF differences between European and African ancestries can explain up to around 84% (s.e. 1.5%) of the loss of prediction accuracy between these populations (Methods), with the remaining loss being presumably explained by differences in heritability between populations and/or differences in effect sizes across populations (for example, owing to gene-by-gene or gene-by-environment interactions). This observation is consistent with common causal variants for height being largely shared across ancestries. Therefore, we anticipate that fine-mapping of GWS loci identified in this study, ideally using methods that can accommodate dense sets of signals and large populations with African ancestries, would substantially improve the accuracy of a derived height PGS for populations of non-European ancestry. Our study has a large number of participants with African ancestries as compared with previous efforts. However, we emphasize that further increasing the size of GWASs in populations of non-European ancestry, including those with diverse African ancestries, is essential to bridge the gap in prediction accuracy—particularly as most studies only partially capture the wide range of ancestral diversity both within Africa and globally. Such increased sample sizes would help to identify potential ancestry-specific causal variants, to facilitate ancestry-specific fine-mapping and to inform gene–environment and gene–ancestry interactions. Another important finding of our study is to show how individual PGS can be optimally combined with familial information and thereby improve the overall accuracy of height prediction to above 54% in populations of European ancestry.

Although large sample sizes are needed to pinpoint the variants responsible for the heritability of height (and larger samples in multiple ancestries will probably be required to map these at finer scale), the prioritization of relevant genes and gene sets is feasible at smaller

sample sizes than that required to account for the common variant heritability. Thus, the sample sizes required for saturation of GWAS are smaller for identifying enriched gene sets, with the identification of genes implicated as potentially causal and mapping of genomic regions containing associated variants requiring successively larger sample sizes. Furthermore, unlike prediction accuracy, prioritization of genes that are likely to be causal and even mapping of associated regions is consistent across ancestries, reflecting the expected similarity in the biological architecture of human height across populations. Recent studies using UKB data predicted that GWAS sample sizes of just over 3 million individuals are required to identify 6,000–7,000 GWS SNPs explaining more than 90% of the SNP-based heritability of height[52]. We showed empirically that these predictions are downwardly biased given that around 10,000 independent associations are, in fact, required to explain 80–90% of the SNP-based heritability of height in EUR individuals. Discrepancies between observed and predicted levels of saturation could be explained by several factors, such as (i) heterogeneity of SNP effects between cohorts and background ancestries, which may have reduced the statistical power of our study as compared to a homogenous sample like UKB; (ii) inconsistent definitions of GWS SNPs (using COJO in this study versus standard clumping in ref. [52]); and, most importantly, (iii) misspecification of the SNP-effects distribution assumed to make these predictions. Nevertheless, if these predictions reflect proportional levels of saturation between traits, then we could expect that two- to tenfold larger samples would be required for GWASs of inflammatory bowel disease (×2, that is, $n = 10$ million), schizophrenia (×7; $n = 35$ million) or BMI (×10; $n = 50$ million) to reach a similar saturation of 80–90% of SNP-based heritability.

Our study has a number of limitations. First, we focused on SNPs from the HM3 panel, which only partially capture common genetic variation. However, although a significant fraction of height variance can be explained by common SNPs outside the HM3 SNPs panel, we showed that the extra information (also referred to as 'hidden heritability') remains concentrated within GWS loci identified in our HM3-SNP-based analyses (Extended Data Fig. 6). This result underlines the widespread allelic heterogeneity at height-associated loci. Another limitation of our study is that we determined conditional associations using a EUR LD reference ($n \approx 350,000$), which is sub-optimal given that around 24% of our discovery sample is of non-European ancestry. We emphasize that no analytical tool with an adequately large multi-ancestry reference panel is at present available to properly address how to identify conditionally independent associations in a multi-ancestry study. Fine-mapping of variants remains a particular challenge when attempted across ancestries in loci containing multiple signals (as is often the case for height). A third limitation of our study is our inability to perform well-powered replication analyses of genetic associations specific to populations with non-European ancestries, owing to the current limited availability of such data. Finally, as with all GWASs, definitive identification of effector genes and the mechanisms by which genes and variants influence phenotype remains a key bottleneck. Therefore, progress towards identifying causal genes from GWAS of height may be achieved by a combination of increasingly large whole-exome sequencing studies, allowing straightforward SNP-to-gene mapping[45], the use of relevant complementary data (for example, context-specific eQTLs in relevant tissues and cell types) and the development of computational methods that can integrate these data.

In summary, our study has been able to show empirically that the combined additive effects of tens of thousands of individual variants, detectable with a large enough experimental sample size, can explain substantial variation in a human phenotype. For human height, we show that studies of the order of around 5 million participants of various ancestries provide enough power to map more than 90% (around 100% in populations of European ancestry) of genetic variance explained by common SNPs down to around 21% of the genome. Mapping the missing 5–10% of SNP-based heritability not accounted for in the four non-European ancestries studied here will require additional and directed efforts in the future.

Height has been used as a model trait for the study of human polygenic traits, including common diseases, because of its high heritability and relative ease of measurement, which enable large sample sizes and increased power. Conclusions about the genetic architecture, sample size requirements for additional GWAS discovery and scope for polygenic prediction that were initially made for height have by-and-large agreed with those for common disease. If the results from this study can also be extrapolated to disease, this would suggest that substantially increased sample sizes could largely resolve the heritability attributed to common variation to a finite set of SNPs (and small genomic regions). These variants and regions would implicate a particular subset of genes, regulatory elements and pathways that would be most relevant to address questions of function, mechanism and therapeutic intervention.

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

Loïc Yengo[1,500✉], Sailaja Vedantam[2,3,500], Eirini Marouli[4,500], Julia Sidorenko[1], Eric Bartell[2,3,5], Saori Sakaue[3,6,7,8], Marielisa Graff[9], Anders U. Eliasen[10,11], Yunxuan Jiang[12], Sridharan Raghavan[13,14], Jenkai Miao[2,3], Joshua D. Arias[15], Sarah E. Graham[16], Ronen E. Mukamel[3,17,18], Cassandra N. Spracklen[19,20], Xianyong Yin[21], Shyh-Huei Chen[22], Teresa Ferreira[23], Heather H. Highland[9], Yingjie Ji[24], Tugce Karaderi[25,26], Kuang Lin[27], Kreete Lüll[28], Deborah E. Malden[27], Carolina Medina-Gomez[29], Moara Machado[15], Amy Moore[30], Sina Rüeger[31,32], Xueling Sim[33], Scott Vrieze[34], Tarunveer S. Ahluwalia[35,36], Masato Akiyama[6,37], Matthew A. Allison[38], Marcus Alvarez[39], Mette K. Andersen[40], Alireza Ani[41,42], Vivek Appadurai[43], Liubov Arbeeva[44], Seema Bhaskar[45], Lawrence F. Bielak[46], Sailalitha Bollepalli[47], Lori L. Bonnycastle[48], Jette Bork-Jensen[40], Jonathan P. Bradfield[49,50], Yuki Bradford[51], Peter S. Braund[52,53], Jennifer A. Brody[54], Kristoffer S. Burgdorf[55,56], Brian E. Cade[5,57], Hui Cai[58], Qiuyin Cai[58], Archie Campbell[59], Marisa Cañadas-Garre[60], Eulalia Catamo[61], Jin-Fang Chai[33], Xiaoran Chai[62,63], Li-Ching Chang[64], Yi-Cheng Chang[64,65,66], Chien-Hsiun Chen[64], Alessandra Chesi[67,68], Seung Hoan Choi[69], Ren-Hua Chung[70], Massimiliano Cocca[61], Maria Pina Concas[61], Christian Couture[71], Gabriel Cuellar-Partida[12,72], Rebecca Danning[73], E. Warwick Daw[74], Frauke Degenhard[75], Graciela E. Delgado[76], Alessandro Delitala[77], Ayse Demirkan[78,79], Xuan Deng[80], Poornima Devineni[81], Alexander Dietl[82,83], Maria Dimitriou[84], Latchezar Dimitrov[85], Rajkumar Dorajoo[86,87], Arif B. Ekici[88], Jorgen E. Engmann[89], Zammy Fairhurst-Hunter[27], Aliki-Eleni Farmaki[84], Jessica D. Faul[90], Juan-Carlos Fernandez-Lopez[91], Lukas Forer[92], Margherita Francescatto[93], Sandra Freitag-Wolf[94], Christian Fuchsberger[95], Tessel E. Galesloot[96], Yan Gao[97], Zishan Gao[98,99,100], Frank Geller[101], Olga Giannakopoulou[4], Franco Giulianini[73], Anette P. Gjesing[40], Anuj Goel[26,102], Scott D. Gordon[103], Mathias Gorski[82], Jakob Grove[104,105,106], Xiuqing Guo[107], Stefan Gustafsson[108], Jeffrey Haessler[109], Thomas F. Hansen[43,56,110], Aki S. Havulinna[47,111], Simon J. Haworth[112,113], Jing He[58], Nancy Heard-Costa[114,115], Prashantha Hebbar[116], George Hindy[3,117], Yuk-Lam A. Ho[118], Edith Hofer[119,120], Elizabeth Holliday[121], Katrin Horn[122,123], Whitney E. Hornsby[16], Jouke-Jan Hottenga[124], Hongyan Huang[125], Jie Huang[126,127], Alicia Huerta-Chagoya[128,129,130], Jennifer E. Huffman[118], Yi-Jen Hung[131], Shaofeng Huo[132], Mi Yeong Hwang[133], Hiroyuki Iha[134], Daisuke D. Ikeda[134], Masato Isono[135], Anne U. Jackson[21], Susanne Jäger[136,137], Iris E. Jansen[138,139], Ingegerd Johansson[140,141], Jost B. Jonas[142,143,144,145], Anna Jonsson[40], Torben Jørgensen[146,147], Ioanna-Panagiota Kalafati[84], Masahiro Kanai[3,6,7], Stavroula Kanoni[4], Line L. Kårhus[146], Anuradhani Kasturiratne[148], Tomohiro Katsuya[149], Takahisa Kawaguchi[150], Rachel L. Kember[151], Katherine A. Kentistou[152,153], Han-Na Kim[154,155], Young Jin Kim[133], Marcus E. Kleber[76,156], Maria J. Knol[78], Azra Kurbasic[157], Marie Lauzon[107], Phuong Le[158,159], Rodney Lea[160], Jong-Young Lee[161], Hampton L. Leonard[162,163,164], Shengchao A. Li[15,165], Xiaohui Li[107], Xiaoyin Li[166,496], Jingjing Liang[166], Honghuang Lin[167], Shih-Yi Lin[168], Jun Liu[27,78], Xueping Liu[101], Ken Sin Lo[169], Jirong Long[58], Laura Lores-Motta[170], Jian'an Luan[171], Valeriya Lyssenko[172,173], Leo-Pekka Lyytikäinen[174,175,176], Anubha Mahajan[26,498], Vasiliki Mamakou[177], Massimo Mangino[178,179], Ani Manichaikul[180], Jonathan Marten[181], Manuel Mattheisen[104,182,183], Laven Mavarani[184], Aaron F. McDaid[31,32], Karina Meidtner[136,137], Tori L. Melendez[16], Josep M. Mercader[18,128,185,186], Yuri Milaneschi[187], Jason E. Miller[188,189], Iona Y. Millwood[27,190], Pashupati P. Mishra[174,175], Ruth E. Mitchell[112,191], Line T. Møllehave[146], Anna Morgan[61], Soeren Mucha[192], Matthias Munz[192], Masahiro Nakatochi[193], Christopher P. Nelson[52,53], Maria Nethander[194,195], Chu Won Nho[196], Aneta A. Nielsen[197], Ilja M. Nolte[41], Suraj S. Nongmaithem[45,198], Raymond Noordam[199], Ioanna Ntalla[4], Teresa Nutile[200], Anita Pandit[21], Paraskevi Christofidou[178], Katri Pärna[28,41], Marc Pauper[170], Eva R. B. Petersen[201], Liselotte V. Petersen[105,202], Niina Pitkänen[203,204], Ozren Polašek[205,206], Alaitz Poveda[157], Michael H. Preuss[207,208], Saiju Pyarajan[5,57,81], Laura M. Raffield[19], Hiromi Rakugi[149], Julia Ramirez[4,209,210], Asif Rasheed[211], Dennis Raven[212], Nigel W. Rayner[26,198,213,214], Carlos Riveros[215,216], Rebecca Rohde[9], Daniela Ruggiero[200,217], Sanni E. Ruotsalainen[47], Kathleen A. Ryan[218,219], Maria Sabater-Lleal[220,221], Richa Saxena[3,186], Markus Scholz[122,123], Anoop Sendamarai[81], Botong Shen[222], Jingchunzi Shi[12], Jae Hun Shin[223], Carlo Sidore[224], Colleen M. Sitlani[54], Roderick C. Slieker[225,226,227], Roelof A. J. Smit[207,228], Albert V. Smith[46,229], Jennifer A. Smith[46,90], Laura J. Smyth[60], Lorraine Southam[213,230], Valgerdur Steinthorsdottir[231], Liang Sun[132], Fumihiko Takeuchi[135], Divya Sri Priyanka Tallapragada[45,232], Kent D. Taylor[107], Bamidele O. Tayo[233], Catherine Tcheandjieu[234,235], Natalie Terzikhan[78], Paola Tesolin[93], Alexander Teumer[236,237], Elizabeth Theusch[238], Deborah J. Thompson[239,240], Gudmar Thorleifsson[231], Paul R. H. J. Timmers[152,181], Stella Trompet[199,241], Constance Turman[125], Simona Vaccargiu[224], Sander W. van der Laan[242], Peter J. van der Most[41], Jan B. van Klinken[243,244,245], Jessica van Setten[246], Shefali V. Verma[67], Niek Verweij[247], Yogasudha Veturi[51], Carol A. Wang[215,216], Chaolong Wang[86,248], Lihua Wang[74], Zhe Wang[207], Helen R. Warren[4,249], Wen Bin Wei[250], Ananda R. Wickremasinghe[148], Matthias Wielscher[251,252], Kerri L. Wiggins[54], Bendik S. Winsvold[253,254], Andrew Wong[255], Yang Wu[1], Matthias Wuttke[256,257], Rui Xia[258], Tian Xie[41], Ken Yamamoto[259], Jingyun Yang[260,261],

Jie Yao[107], Hannah Young[34], Noha A. Yousri[262,263], Lei Yu[260,261], Lingyao Zeng[264], Weihua Zhang[265,266], Xinyuan Zhang[51], Jing-Hua Zhao[267], Wei Zhao[46], Wei Zhou[3,268,269,270], Martina E. Zimmermann[82], Magdalena Zoledziewska[224], Linda S. Adair[271,272], Hieab H. H. Adams[273,274,275], Carlos A. Aguilar-Salinas[276,277], Fahd Al-Mulla[116], Donna K. Arnett[278], Folkert W. Asselbergs[246,279,280], Bjørn Olav Åsvold[281,282,283], John Attia[121], Bernhard Banas[284], Stefania Bandinelli[285], David A. Bennett[260,261], Tobias Bergler[284], Dwaipayan Bharadwaj[286], Ginevra Biino[287], Hans Bisgaard[10], Eric Boerwinkle[288], Carsten A. Böger[284,289,290], Klaus Bønnelykke[10], Dorret I. Boomsma[124], Anders D. Børglum[104,105,291,292], Judith B. Borja[293,294], Claude Bouchard[295], Donald W. Bowden[85,296], Ivan Brandslund[297,298], Ben Brumpton[281,299], Julie E. Buring[5,73], Mark J. Caulfield[4,249], John C. Chambers[265,266,300,301], Giriraj R. Chandak[45,302], Stephen J. Chanock[15], Nish Chaturvedi[255], Yii-Der Ida Chen[107], Zhengming Chen[27,190], Ching-Yu Cheng[62,303], Ingrid E. Christophersen[304,305], Marina Ciullo[200,217], John W. Cole[306,307], Francis S. Collins[48], Richard S. Cooper[233], Miguel Cruz[308], Francesco Cucca[224,309], L. Adrienne Cupples[80,115], Michael J. Cutler[310], Scott M. Damrauer[51,311,312], Thomas M. Dantoft[146], Gert J. de Borst[313], Lisette C. P. G. M. de Groot[314], Philip L. De Jager[3,315], Dominique P. V. de Kleijn[313], H. Janaka de Silva[148], George V. Dedoussis[84], Anneke I. den Hollander[170], Shufa Du[271,272], Douglas F. Easton[239,316], Petra J. M. Elders[317], A. Heather Eliassen[57,125,318], Patrick T. Ellinor[69,319,320], Sölve Elmståhl[321], Jeanette Erdmann[192], Michele K. Evans[222], Diane Fatkin[322,323,324], Bjarke Feenstra[101], Mary F. Feitosa[74], Luigi Ferrucci[325], Ian Ford[326], Myriam Fornage[258,327], Andre Franke[75], Paul W. Franks[157,318,328], Barry I. Freedman[329], Paolo Gasparini[61,93], Christian Gieger[99,137], Giorgia Girotto[61,93], Michael E. Goddard[330,331], Yvonne M. Golightly[9,44,332,333], Clicerio Gonzalez-Villalpando[334], Penny Gordon-Larsen[271,272], Harald Grallert[99,137], Struan F. A. Grant[49,335,336,337], Niels Grarup[40], Lyn Griffiths[160], Vilmundur Gudnason[229,338], Christopher Haiman[339], Hakon Hakonarson[49,335,340,341], Torben Hansen[40], Catharina A. Hartman[212], Andrew T. Hattersley[342], Caroline Hayward[181], Susan R. Heckbert[343], Chew-Kiat Heng[344,345], Christian Hengstenberg[346], Alex W. Hewitt[347,348,349], Haretsugu Hishigaki[134], Carel B. Hoyng[170], Paul L. Huang[5,320,350], Wei Huang[351], Steven C. Hunt[262,352], Kristian Hveem[281,282], Elina Hyppönen[353,354], William G. Iacono[34], Sahoko Ichihara[355], M. Arfan Ikram[78], Carmen R. Isasi[356], Rebecca D. Jackson[357], Marjo-Riitta Jarvelin[251,358,359,360], Zi-Bing Jin[145,361], Karl-Heinz Jöckel[184], Peter K. Joshi[153], Pekka Jousilahti[111], J. Wouter Jukema[241,362,363], Mika Kähönen[364,365], Yoichiro Kamatani[6,366], Kui Dong Kang[367], Jaakko Kaprio[47], Sharon L. R. Kardia[46], Fredrik Karpe[214,368], Norihiro Kato[135], Frank Kee[60], Thorsten Kessler[264,369], Amit V. Khera[3,186], Chiea Chuen Khor[86], Lambertus A. L. M. Kiemeney[96,370], Bong-Jo Kim[133], Eung Kweon Kim[371,372], Hyung-Lae Kim[373], Paulus Kirchhof[374,375,376,377], Mika Kivimaki[378], Woon-Puay Koh[379], Heikki A. Koistinen[111,380,381], Genovefa D. Kolovou[382], Jaspal S. Kooner[265,301,383,384], Charles Kooperberg[109], Anna Köttgen[256], Peter Kovacs[385], Adriaan Kraaijeveld[246], Peter Kraft[125], Ronald M. Krauss[238], Meena Kumari[386], Zoltan Kutalik[31,32], Markku Laakso[387], Leslie A. Lange[388], Claudia Langenberg[171,389], Lenore J. Launer[222], Loic Le Marchand[390], Hyejin Lee[391], Nanette R. Lee[293], Terho Lehtimäki[174,175], Huaixing Li[132], Liming Li[392,393], Wolfgang Lieb[394], Xu Lin[132,395], Lars Lind[108], Allan Linneberg[146,396], Ching-Ti Liu[80], Jianjun Liu[86], Markus Loeffler[122,123], Barry London[397], Steven A. Lubitz[69,319,320], Stephen J. Lye[398], David A. Mackey[347,349], Reedik Mägi[28], Patrik K. E. Magnusson[399], Gregory M. Marcus[400], Pedro Marques Vidal[401,402], Nicholas G. Martin[103], Winfried März[76,403,404], Fumihiko Matsuda[150], Robert W. McGarrah[405,406], Matt McGue[34], Amy Jayne McKnight[60], Sarah E. Medland[407], Dan Mellström[194,408], Andres Metspalu[28], Braxton D. Mitchell[218,219,409], Paul Mitchell[410], Dennis O. Mook-Kanamori[228,411], Andrew D. Morris[412], Lorelei A. Mucci[125], Patricia B. Munroe[4,249], Mike A. Nalls[162,163,164], Saman Nazarian[413], Amanda E. Nelson[44,414], Matt J. Neville[214,368], Christopher Newton-Cheh[186,320], Christopher S. Nielsen[415,416], Markus M. Nöthen[417], Claes Ohlsson[194,418], Albertine J. Oldehinkel[212], Lorena Orozco[419], Katja Pahkala[203,204,420], Päivi Pajukanta[39,421], Colin N. A. Palmer[422], Esteban J. Parra[159], Cristian Pattaro[95], Oluf Pedersen[40], Craig E. Pennell[215,216], Brenda W. J. H. Penninx[187], Louis Perusse[71,423], Annette Peters[100,137,424], Patricia A. Peyser[46], David J. Porteous[59], Danielle Posthuma[138], Chris Power[425], Peter P. Pramstaller[95], Michael A. Province[74], Qibin Qi[356], Jia Qu[361], Daniel J. Rader[51,426], Olli T. Raitakari[203,204,427], Sarju Ralhan[428], Loukianos S. Rallidis[429], Dabeeru C. Rao[430], Susan Redline[5,57], Dermot F. Reilly[431], Alexander P. Reiner[109,432], Sang Youl Rhee[433], Paul M. Ridker[5,73], Michiel Rienstra[247], Samuli Ripatti[3,47,434], Marylyn D. Ritchie[51], Dan M. Roden[435], Frits R. Rosendaal[228], Jerome I. Rotter[107], Igor Rudan[152], Femke Rutters[436], Charumathi Sabanayagam[62,303], Danish Saleheen[211,437], Veikko Salomaa[111], Nilesh J. Samani[52,53], Dharambir K. Sanghera[438,439,440,441], Naveed Sattar[442], Börge Schmidt[184], Helena Schmidt[443], Reinhold Schmidt[119], Matthias B. Schulze[136,137,444], Heribert Schunkert[445], Laura J. Scott[21], Rodney J. Scott[446], Peter Sever[384], Eric J. Shiroma[222], M. Benjamin Shoemaker[447], Xiao-Ou Shu[58], Eleanor M. Simonsick[325], Mario Sims[97], Jai Rup Singh[448],

Andrew B. Singleton[162], Moritz F. Sinner[369,449], J. Gustav Smith[450,451,452], Harold Snieder[41], Tim D. Spector[178], Meir J. Stampfer[57,125,318], Klaus J. Stark[82], David P. Strachan[453], Leen M. 't Hart[225,226,227,454], Yasuharu Tabara[150], Hua Tang[455], Jean-Claude Tardif[169,456], Thangavel A. Thanaraj[116], Nicholas J. Timpson[112,191], Anke Tönjes[385], Angelo Tremblay[71,423], Tiinamaija Tuomi[47,173,457,458], Jaakko Tuomilehto[111,459,460], Maria-Teresa Tusié-Luna[461,462], Andre G. Uitterlinden[29], Rob M. van Dam[33,463,464], Pim van der Harst[246,247], Nathalie Van der Velde[29,465], Cornelia M. van Duijn[27,78], Natasja M. van Schoor[466], Veronique Vitart[181], Uwe Völker[237,467], Peter Vollenweider[401,402], Henry Völzke[236,237], Niels H. Wacher-Rodarte[468], Mark Walker[469], Ya Xing Wang[145], Nicholas J. Wareham[171], Richard M. Watanabe[470,471,472], Hugh Watkins[26,102], David R. Weir[90], Thomas M. Werge[43,396,473], Elisabeth Widen[47], Lynne R. Wilkens[390], Gonneke Willemsen[124], Walter C. Willett[125,318], James F. Wilson[152,181], Tien-Yin Wong[62,303], Jeong-Taek Woo[433], Alan F. Wright[181], Jer-Yuarn Wu[64,474], Huichun Xu[218,219], Chittaranjan S. Yajnik[475], Mitsuhiro Yokota[476], Jian-Min Yuan[477,478], Eleftheria Zeggini[213,230,479], Babette S. Zemel[51,68,337,340], Wei Zheng[58], Xiaofeng Zhu[166], Joseph M. Zmuda[478], Alan B. Zonderman[222], John-Anker Zwart[253,480], 23andMe Research Team*, VA Million Veteran Program*, DiscovEHR (DiscovEHR and MyCode Community Health Initiative)*, eMERGE (Electronic Medical Records and Genomics Network)*, Lifelines Cohort Study*, The PRACTICAL Consortium*, Understanding Society Scientific Group*, Daniel I. Chasman[5,73], Yoon Shin Cho[223], Iris M. Heid[82], Mark I. McCarthy[26,214,497], Maggie C. Y. Ng[85,482], Christopher J. O'Donnell[5,57,483], Fernando Rivadeneira[29], Unnur Thorsteinsdottir[231,338], Yan V. Sun[484,485], E. Shyong Tai[33,463], Michael Boehnke[21], Panos Deloukas[4,486], Anne E. Justice[9,481], Cecilia M. Lindgren[3,23,26], Ruth J. F. Loos[40,207,208,487], Karen L. Mohlke[19], Kari E. North[9], Kari Stefansson[231,338], Robin G. Walters[27,190], Thomas W. Winkler[82], Kristin L. Young[9], Po-Ru Loh[3,17,18], Jian Yang[1,488,489], Tõnu Esko[28], Themistocles L. Assimes[234,235], Adam Auton[12], Goncalo R. Abecasis[21], Cristen J. Willer[16,268,490], Adam E. Locke[491], Sonja I. Berndt[15], Guillaume Lettre[169,456], Timothy M. Frayling[24], Yukinori Okada[6,7,492,493,498,499,501✉], Andrew R. Wood[24,501✉], Peter M. Visscher[1,501✉] & Joel N. Hirschhorn[2,494,495,501✉]

[1]Institute for Molecular Bioscience, The University of Queensland, Brisbane, Queensland, Australia. [2]Division of Endocrinology, Boston Children's Hospital, Boston, MA, USA. [3]Program in Medical and Population Genetics, Broad Institute of MIT and Harvard, Cambridge, MA, USA. [4]William Harvey Research Institute, Barts and the London School of Medicine and Dentistry, Queen Mary University of London, London, UK. [5]Harvard Medical School, Boston, MA, USA. [6]Laboratory for Statistical Analysis, RIKEN Center for Integrative Medical Sciences, Yokohama, Japan. [7]Department of Statistical Genetics, Osaka University Graduate School of Medicine, Osaka, Japan. [8]Divisions of Genetics and Rheumatology, Brigham and Women's Hospital and Department of Medicine, Harvard Medical School, Boston, MA, USA. [9]Department of Epidemiology, Gillings School of Global Public Health, University of North Carolina at Chapel Hill, Chapel Hill, NC, USA. [10]COPSAC, Copenhagen Prospective Studies on Asthma in Childhood, Herlev and Gentofte Hospital, University of Copenhagen, Copenhagen, Denmark. [11]Section for Bioinformatics, Department of Health Technology, Technical University of Denmark, Copenhagen, Denmark. [12]23andMe, Sunnyvale, CA, USA. [13]Department of Veterans Affairs, Eastern Colorado Healthcare System, Aurora, CO, USA. [14]Division of Biomedical Informatics and Personalized Medicine, University of Colorado Anschutz Medical Campus, Aurora, CO, USA. [15]Division of Cancer Epidemiology and Genetics, National Cancer Institute, Rockville, MD, USA. [16]Department of Internal Medicine, Division of Cardiology, University of Michigan, Ann Arbor, MI, USA. [17]Division of Genetics, Department of Medicine, Brigham and Women's Hospital, Boston, MA, USA. [18]Department of Medicine, Harvard Medical School, Boston, MA, USA. [19]Department of Genetics, University of North Carolina at Chapel Hill, Chapel Hill, NC, USA. [20]Department of Biostatistics and Epidemiology, School of Public Health and Health Sciences, University of Massachusetts, Amherst, MA, USA. [21]Department of Biostatistics and Center for Statistical Genetics, University of Michigan School of Public Health, Ann Arbor, MI, USA. [22]Department of Biostatistics and Data Science, Wake Forest School of Medicine, Winston-Salem, NC, USA. [23]Big Data Institute, Li Ka Shing Centre for Health Information and Discovery, University of Oxford, Oxford, UK. [24]Genetics of Complex Traits, College of Medicine and Health, University of Exeter, Exeter, UK. [25]Center for Health Data Science, Faculty of Health and Medical Sciences, University of Copenhagen, Copenhagen, Denmark. [26]Wellcome Centre for Human Genetics, Nuffield Department of Medicine, University of Oxford, Oxford, UK. [27]Nuffield Department of Population Health, University of Oxford, Oxford, UK. [28]Institute of Genomics, Estonian Genome Centre, University of Tartu, Tartu, Estonia. [29]Department of Internal Medicine, Erasmus MC, University Medical Center Rotterdam, Rotterdam, The Netherlands. [30]Division of Biostatistics and Epidemiology, RTI International, Durham, NC, USA. [31]Center for Primary Care and Public Health, University of Lausanne, Lausanne, Switzerland. [32]Swiss Institute of Bioinformatics, Lausanne, Switzerland. [33]Saw Swee Hock School of Public Health, National University of Singapore and National University Health System, Singapore, Singapore. [34]Department of Psychology, University of Minnesota, Minneapolis, MN, USA. [35]Steno Diabetes Center Copenhagen, Herlev, Denmark. [36]Department of Biology, The Bioinformatics Center, University of Copenhagen, Copenhagen, Denmark. [37]Department of Ophthalmology, Graduate School of Medical Sciences, Kyushu University, Fukuoka, Japan. [38]Department of Family Medicine, University of California, San Diego, La Jolla, CA, USA. [39]Department of Human Genetics, David Geffen School of Medicine at UCLA, Los Angeles, CA, USA. [40]Novo Nordisk Foundation Center for Basic Metabolic Research, Faculty of Health and Medical Sciences, University of Copenhagen, Copenhagen, Denmark. [41]Department of Epidemiology, University of Groningen, University Medical Center Groningen, Groningen, The Netherlands. [42]Department of Bioinformatics, Isfahan University of Medical Sciences, Isfahan, Iran. [43]Institute of Biological Psychiatry, Mental Health Services, Copenhagen University Hospital, Copenhagen, Denmark. [44]Thurston Arthritis Research Center, University of North Carolina at Chapel Hill, Chapel Hill, NC, USA. [45]Genomic Research on Complex diseases (GRC-Group), CSIR-Centre for Cellular and Molecular Biology, Hyderabad, India. [46]Department of Epidemiology, University of Michigan School of Public Health, Ann Arbor, MI, USA. [47]Institute for Molecular Medicine Finland (FIMM), HiLIFE, University of Helsinki, Helsinki, Finland. [48]Molecular Genetics Section, Center for Precision Health Research, National Human Genome Research Institute, National Institutes of Health, Bethesda, MD, USA. [49]Center for Applied Genomics, Children's Hospital of Philadelphia, Philadelphia, PA, USA. [50]Quantinuum Research, Wayne, PA, USA. [51]Department of Genetics, University of Pennsylvania, Philadelphia, PA, USA. [52]Department of Cardiovascular Sciences, University of Leicester, Leicester, UK. [53]NIHR Leicester Biomedical Research Centre, Glenfield Hospital, Leicester, UK. [54]Cardiovascular Health Research Unit, Department of Medicine, University of Washington, Seattle, WA, USA. [55]Department of Clinical Immunology, Copenhagen University Hospital, Rigshospitalet, Copenhagen, Denmark. [56]NovoNordic Center for Protein Research, Copenhagen University, Copenhagen, Denmark. [57]Department of Medicine, Brigham and Women's Hospital, Boston, MA, USA. [58]Division of Epidemiology, Department of Medicine, Vanderbilt University Medical Center, Nashville, TN, USA. [59]Centre for Genomic and Experimental Medicine, Institute of Genetics and Cancer, University of Edinburgh, Edinburgh, UK. [60]Centre for Public Health, Queen's University of Belfast, Belfast, UK. [61]Institute for Maternal and Child Health – IRCCS, Burlo Garofolo, Trieste, Italy. [62]Ocular Epidemiology, Singapore Eye Research Institute, Singapore National Eye Centre, Singapore, Singapore. [63]Department of Ophthalmology, National University of Singapore and National University Health System, Singapore, Singapore. [64]Institute of Biomedical Sciences, Academia Sinica, Taipei, Taiwan. [65]Graduate Institute of Medical Genomics and Proteomics, Medical College, National Taiwan University, Taipei, Taiwan. [66]Department of Internal Medicine, National Taiwan University Hospital, Taipei, Taiwan. [67]Department of Pathology and Laboratory Medicine, University of Pennsylvania, Philadelphia, PA, USA. [68]Center for Spatial and Functional Genomics, Division of Human Genetics, Children's Hospital of Philadelphia, Philadelphia, PA, USA. [69]Cardiovascular Disease Initiative, Broad Institute of MIT and Harvard, Cambridge, MA, USA. [70]Institute of Population Health Sciences, National Health Research Institutes, Zhunan, Taiwan. [71]Department of Kinesiology, Faculty of Medicine, Université Laval, Québec City, Quebec, Canada. [72]Diamantina Institute, The University of Queensland, Brisbane, Queensland, Australia. [73]Division of Preventive Medicine, Brigham and Women's Hospital, Boston, MA, USA. [74]Division of Statistical Genomics, Department of Genetics, Washington University School of Medicine, St Louis, MO, USA. [75]Institute of Clinical Molecular Biology, Christian-Albrechts University of Kiel, Kiel, Germany. [76]Vth Department of Medicine, Medical Faculty Mannheim, Heidelberg University, Mannheim, Germany. [77]Dipartimento di Scienze Mediche Chirurgiche e Sperimentali, Università degli Studi di Sassari, Sassari, Italy. [78]Department of Epidemiology, Erasmus MC, University Medical Center Rotterdam, Rotterdam, The Netherlands. [79]Section of Statistical Multi-omics, Department of Clinical and Experimental Medicine, University of Surrey, Guildford, UK. [80]Department of Biostatistics, Boston University School of Public Health, Boston, MA, USA. [81]Center for Data and Computational Sciences, VA Boston Healthcare System, Boston, MA, USA. [82]Department of Genetic Epidemiology, University of Regensburg, Regensburg, Germany. [83]Department of Internal Medicine II, University Hospital Regensburg, Regensburg, Germany. [84]Department of Nutrition and Dietetics, School of Health and Education, Harokopio University of Athens, Athens, Greece. [85]Center for Precision Medicine, Wake Forest School of Medicine, Medical Center Boulevard, Winston-Salem, NC, USA. [86]Genome Institute of Singapore, Agency for Science, Technology and Research, Singapore, Singapore. [87]Health Services and Systems Research, Duke-NUS Medical School, Singapore, Singapore. [88]Institute of Human Genetics, Universitätsklinikum Erlangen, Friedrich-Alexander-Universität Erlangen-Nürnberg, Erlangen, Germany. [89]Institute of Cardiovascular Science, Faculty of Population Health, University College London, London, UK. [90]Survey Research Center, Institute for Social Research, University of Michigan, Ann Arbor, MI, USA. [91]Computational Genomics Department, National Institute of Genomic Medicine, Mexico City, Mexico. [92]Institute of Genetic Epidemiology, Medical University of Innsbruck, Innsbruck, Austria. [93]Department of Medicine, Surgery and Health Sciences, University of Trieste, Trieste, Italy. [94]Institute of Medical Informatics and Statistics, Kiel University, Kiel, Germany. [95]Eurac Research, Institute for Biomedicine, Affiliated Institute of the University of Lübeck, Bolzano, Italy. [96]Radboud University Medical Center, Radboud Institute for Health Sciences, Department for Health Evidence, Nijmegen, The Netherlands. [97]Jackson Heart Study, Department of Medicine, University of Mississippi, Jackson, MS, USA. [98]Nanjing University of Chinese Medicine, Nanjing, China. [99]Research Unit of Molecular Epidemiology, Institute of Epidemiology, Helmholtz Zentrum München Research Center for Environmental Health, Neuherberg, Germany. [100]Institute of Epidemiology, Helmholtz Zentrum München Research Center for Environmental Health, Neuherberg, Germany. [101]Department of Epidemiology Research, Statens Serum Institut, Copenhagen, Denmark. [102]Cardiovascular Medicine, Radcliffe Department of Medicine, University of Oxford, John Radcliffe Hospital, Oxford, UK. [103]Genetic Epidemiology, QIMR Berghofer Medical Research Institute, Brisbane, Queensland, Australia. [104]Department of Biomedicine (Human Genetics) and iSEQ Center, Aarhus University, Aarhus, Denmark. [105]The Lundbeck Foundation Initiative for Integrative Psychiatric Research, iPSYCH, Aarhus, Denmark.

[106]BiRC—Bioinformatics Research Centre, Aarhus University, Aarhus, Denmark. [107]The Institute for Translational Genomics and Population Sciences, Department of Pediatrics, The Lundquist Institute for Biomedical Innovation at Harbor-UCLA Medical Center, Torrance, CA, USA. [108]Department of Medical Sciences, Uppsala University, Uppsala, Sweden. [109]Division of Public Health Sciences, Fred Hutchinson Cancer Research Center, Seattle, WA, USA. [110]Danish Headache Center, Department of Neurology, Copenhagen University Hospital, Rigshospitalet, Rigshospitalet, Copenhagen, Denmark. [111]Department of Public Health and Welfare, Finnish Institute for Health and Welfare, Helsinki, Finland. [112]MRC Integrative Epidemiology Unit, University of Bristol, Bristol, UK. [113]Bristol Dental School, University of Bristol, Bristol, UK. [114]Department of Neurology, Boston University School of Medicine, Boston, MA, USA. [115]Framingham Heart Study, Framingham, MA, USA. [116]Department of Genetics and Bioinformatics, Dasman Diabetes Institute, Kuwait City, Kuwait. [117]Department of Clinical Sciences in Malmö, Lund University, Malmö, Sweden. [118]Veterans Affairs Boston Healthcare System, Boston, MA, USA. [119]Clinical Division of Neurogeriatrics, Department of Neurology, Medical University of Graz, Graz, Austria. [120]Institute for Medical Informatics, Statistics and Documentation, Medical University of Graz, Graz, Austria. [121]School of Medicine and Public Health, University of Newcastle, Callaghan, New South Wales, Australia. [122]Institute for Medical Informatics, Statistics and Epidemiology, University of Leipzig, Medical Faculty, Leipzig, Germany. [123]LIFE Research Center for Civilization Diseases, University of Leipzig, Medical Faculty, Leipzig, Germany. [124]Department of Biological Psychology, Behaviour and Movement Sciences, Vrije Universiteit Amsterdam, Amsterdam, The Netherlands. [125]Department of Epidemiology, Harvard T.H. Chan School of Public Health, Boston, MA, USA. [126]School of Public Health and Emergency Management, Southern University of Science and Technology, Shenzhen, China. [127]Institute for Global Health and Development, Peking University, Beijing, China. [128]Programs in Metabolism and Medical and Population Genetics, Broad Institute of Harvard and MIT, Cambridge, MA, USA. [129]Departamento de Medicina Genómica y Toxicología Ambiental, Instituto de Investigaciones Biomédicas, Universidad Nacional Autónoma de México Ciudad Universitaria, Mexico City, Mexico. [130]Unidad de Biología Molecular y Medicina Genómica, Instituto Nacional de Ciencias Médicas y Nutrición, Mexico City, Mexico. [131]Division of Endocrine and Metabolism, Tri-Service General Hospital Songshan Branch, Taipei, Taiwan. [132]Shanghai Institute of Nutrition and Health, University of Chinese Academy of Sciences, Chinese Academy of Sciences, Shanghai, China. [133]Division of Genome Science, Department of Precision Medicine, National Institute of Health, Cheongju, Republic of Korea. [134]Biomedical Technology Research Center, Tokushima Research Institute, Otsuka Pharmaceutical Co., Tokushima, Japan. [135]Research Institute, National Center for Global Health and Medicine, Tokyo, Japan. [136]Department of Molecular Epidemiology, German Institute of Human Nutrition Potsdam-Rehbruecke, Nuthetal, Germany. [137]German Center for Diabetes Research (DZD), Neuherberg, Germany. [138]Department of Complex Trait Genetics, Center for Neurogenomics and Cognitive Research, Amsterdam Neuroscience, Vrije Universiteit Amsterdam, Amsterdam, The Netherlands. [139]Department of Child and Adolescent Psychiatry and Pediatric Psychology, Section Complex Trait Genetics, Amsterdam Neuroscience, Vrije Universiteit Medical Center, Amsterdam, The Netherlands. [140]Department of Biobank Research, Umeå University, Umeå, Sweden. [141]Department of Odontology, Umeå University, Umeå, Sweden. [142]Institute of Molecular and Clinical Ophthalmology Basel, Basel, Switzerland. [143]Privatpraxis Prof Jonas und Dr Panda-Jonas, Heidelberg, Germany. [144]Department of Ophthalmology, Medical Faculty Mannheim, Heidelberg University, Mannheim, Germany. [145]Beijing Institute of Ophthalmology, Beijing Tongren Eye Center, Beijing Tongren Hospital, Capital Medical University, Beijing Ophthalmology and Visual Sciences Key Laboratory, Beijing, China. [146]Center for Clinical Research and Prevention, Copenhagen University Hospital - Bispebjerg and Frederiksberg, Copenhagen, Denmark. [147]Department of Public Health, Faculty of Health and Medical Sciences, University of Copenhagen, Copenhagen, Denmark. [148]Faculty of Medicine, University of Kelaniya, Ragama, Sri Lanka. [149] Department of Geriatric and General Medicine, Osaka University Graduate School of Medicine, Suita, Japan. [150]Center for Genomic Medicine, Kyoto University Graduate School of Medicine, Kyoto, Japan. [151]Department of Psychiatry, University of Pennsylvania, Philadelphia, PA, USA. [152]Centre for Global Health, Usher Institute, University of Edinburgh, Edinburgh, UK. [153]Centre for Cardiovascular Sciences, Queens Medical Research Institute, University of Edinburgh, Edinburgh, UK. [154]Medical Research Institute, Kangbuk Samsung Hospital, Sungkyunkwan University School of Medicine, Seoul, Republic of Korea. [155]Department of Clinical Research Design and Evaluation (SAIHST), Sungkyunkwan University, Seoul, Republic of Korea. [156]SYNLAB MVZ Humangenetik Mannheim, Mannheim, Germany. [157]Department of Clinical Sciences, Genetic and Molecular Epidemiology Unit, Lund University, Malmö, Sweden. [158]Department of Computer Science, University of Toronto, Toronto, Ontario, Canada. [159]Department of Anthropology, University of Toronto at Mississauga, Mississauga, Ontario, Canada. [160]Genomics Research Centre, Centre for Genomics and Personalised Health, School of Biomedical Sciences, Queensland University of Technology, Kelvin Grove, Queensland, Australia. [161]Oneomics, Soonchunhyang Mirai Medical Center, Bucheon-si, Republic of Korea. [162]Laboratory of Neurogenetics, National Institute on Aging, National Institutes of Health, Bethesda, MD, USA. [163]Center for Alzheimer's and Related Dementias, National Institutes of Health, Bethesda, MD, USA. [164]Data Tecnica International, Glen Echo, MD, USA. [165]Cancer Genomics Research Laboratory, Leidos Biomedical Research, Rockville, MD, USA. [166]Department of Population and Quantitative Health Sciences, Case Western Reserve University, Cleveland, OH, USA. [167]Department of Medicine, University of Massachusetts Chan Medical School, Worcester, MA, USA. [168]Center for Geriatrics and Gerontology, Taichung Veterans General Hospital, Taichung, Taiwan. [169]Montreal Heart Institute, Montreal, Quebec, Canada. [170]Departments of Ophthalmology and Human Genetics, Radboud University Nijmegen Medical Center, Nijmegen, The Netherlands.

[171]MRC Epidemiology Unit, University of Cambridge School of Clinical Medicine, Cambridge, UK. [172]Department of Clinical Science, Center for Diabetes Research, University of Bergen, Bergen, Norway. [173]Department of Clinical Sciences, Lund University Diabetes Centre, Malmö, Sweden. [174]Department of Clinical Chemistry, Fimlab Laboratories, Tampere, Finland. [175]Department of Clinical Chemistry, Finnish Cardiovascular Research Center - Tampere, Faculty of Medicine and Health Technology, Tampere University, Tampere, Finland. [176]Department of Cardiology, Heart Center, Tampere University Hospital, Tampere, Finland. [177]National and Kapodistrian University of Athens, Dromokaiteio Psychiatric Hospital, Athens, Greece. [178]Department of Twin Research and Genetic Epidemiology, King's College London, London, UK. [179]NIHR Biomedical Research Centre at Guy's and St Thomas' Foundation Trust, London, UK. [180]Center for Public Health Genomics, University of Virginia School of Medicine, Charlottesville, VA, USA. [181]MRC Human Genetics Unit, Institute of Genetics and Cancer, University of Edinburgh, Western General Hospital, Edinburgh, UK. [182]Department of Psychiatry and Department of Community Health and Epidemiology, Dalhousie University, Halifax, Nova Scotia, Canada. [183]Institute of Psychiatric Phenomics and Genomics (IPPG), University Hospital, LMU Munich, Munich, Germany. [184]Institute for Medical Informatics, Biometry and Epidemiology, University Hospital Essen, Essen, Germany. [185]Diabetes Unit, Massachusetts General Hospital, Boston, MA, USA. [186]Center for Genomic Medicine, Massachusetts General Hospital, Boston, MA, USA. [187]Department of Psychiatry, Amsterdam Public Health and Amsterdam Neuroscience, Amsterdam UMC and Vrije Universiteit, Amsterdam, The Netherlands. [188]Biomedical and Translational Informatics Institute, Geisinger, Danville, PA, USA. [189]Department of Genetics, Institute for Biomedical Informatics, Perelman School of Medicine, University of Pennsylvania, Philadelphia, PA, USA. [190]MRC Population Health Research Unit, Nuffield Department of Population Health, University of Oxford, Oxford, UK. [191]Population Health Sciences, Bristol Medical School, University of Bristol, Bristol, UK. [192]Institute for Cardiogenetics, University of Lübeck, DZHK (German Research Centre for Cardiovascular Research) partner site Hamburg/Lübeck/Kiel and University Heart Center Lübeck, Lübeck, Germany. [193]Public Health Informatics Unit, Department of Integrated Health Sciences, Nagoya University Graduate School of Medicine, Nagoya, Japan. [194]Centre for Bone and Arthritis Research, Department of Internal Medicine and Clinical Nutrition, Institute of Medicine, Sahlgrenska Academy, University of Gothenburg, Gothenburg, Sweden. [195]Bioinformatics Core Facility, Sahlgrenska Academy, University of Gothenburg, Gothenburg, Sweden. [196]Korea Institute of Science and Technology, Gangneung Institute of Natural Products, Gangneung, Republic of Korea. [197]Department of Clinical Biochemistry, Lillebaelt Hospital, Kolding, Denmark. [198]Department of Human Genetics, Wellcome Sanger Institute, Hinxton, UK. [199]Department of Internal Medicine, Section of Gerontology and Geriatrics, Leiden University Medical Center, Leiden, The Netherlands. [200]Institute of Genetics and Biophysics A. Buzzati-Traverso, CNR, Naples, Italy. [201]Department of Clinical Biochemistry and Immunology, Hospital of Southern Jutland, Aabenraa, Denmark. [202]The National Centre for Register-based Research, University of Aarhus, Aarhus, Denmark. [203]Centre for Population Health Research, University of Turku and Turku University Hospital, Turku, Finland. [204]Research Centre of Applied and Preventive Cardiovascular Medicine, University of Turku, Turku, Finland. [205]Medical School, University of Split, Split, Croatia. [206]Algebra University College, Zagreb, Croatia. [207]The Charles Bronfman Institute for Personalized Medicine, Icahn School of Medicine at Mount Sinai, New York, NY, USA. [208]Department of Environmental Medicine and Public Health, Icahn School of Medicine at Mount Sinai, New York, NY, USA. [209]Aragon Institute of Engineering Research, University of Zaragoza, Zaragoza, Spain. [210]Centro de Investigación Biomédica en Red en Bioingeniería, Biomateriales y Nanomedicina (CIBER-BBN), Madrid, Spain. [211]Center for Non-Communicable Diseases, Karachi, Pakistan. [212]Department of Psychiatry, Interdisciplinary Center Psychopathology and Emotion Regulation, University of Groningen, University Medical Center Groningen, Groningen, The Netherlands. [213]Institute of Translational Genomics, Helmholtz Zentrum München, German Research Center for Environmental Health, Neuherberg, Germany. [214]Present address: Oxford Centre for Diabetes, Endocrinology and Metabolism, Radcliffe Department of Medicine, University of Oxford, Churchill Hospital, Oxford, UK. [215]Hunter Medical Research Institute, New Lambton Heights, New South Wales, Australia. [216]School of Medicine and Public Health, College of Health, Medicine and Wellbeing, The University of Newcastle, New Lambton Heights, New South Wales, Australia. [217]IRCCS Neuromed, Pozzilli, Italy. [218]Department of Medicine, Division of Endocrinology, Diabetes and Nutrition, University of Maryland School of Medicine, Baltimore, MD, USA. [219]Program for Personalized and Genomic Medicine, University of Maryland School of Medicine, Baltimore, MD, USA. [220]Unit of Genomics of Complex Diseases, Sant Pau Biomedical Research Institute (IIB Sant Pau), Barcelona, Spain. [221]Cardiovascular Medicine Unit, Department of Medicine, Karolinska Institutet, Center for Molecular Medicine, Stockholm, Sweden. [222]Laboratory of Epidemiology and Population Sciences, National Institute on Aging, National Institutes of Health, Baltimore, MD, USA. [223]Department of Biomedical Science, Hallym University, Chuncheon, Republic of Korea. [224]Istituto di Ricerca Genetica e Biomedica, Consiglio Nazionale delle Ricerche (CNR), Cagliari, Italy. [225]Department of Cell and Chemical Biology, Leiden University Medical Center, Leiden, The Netherlands. [226]Epidemiology and Data Science, Amsterdam UMC, location Vrije Universiteit Amsterdam, Amsterdam, The Netherlands. [227]Amsterdam Cardiovascular Sciences, Amsterdam, The Netherlands. [228]Department of Clinical Epidemiology, Leiden University Medical Center, Leiden, The Netherlands. [229]Icelandic Heart Association, Kópavogur, Iceland. [230]Wellcome Sanger Institute, Hinxton, UK. [231]deCODE Genetics/Amgen, Reykjavik, Iceland. [232]Mohn Nutrition Research Laboratory, Department of Clinical Science, University of Bergen, Bergen, Norway. [233]Department of Public Health Sciences, Parkinson School of Health Sciences and Public Health, Loyola University Chicago, Maywood, IL, USA. [234]VA Palo Alto Health Care System, Palo Alto, CA, USA. [235]Department of Medicine, Stanford University School of

Medicine, Stanford, CA, USA. [236]Institute for Community Medicine, University Medicine Greifswald, Greifswald, Germany. [237]DZHK (German Centre for Cardiovascular Research), partner site Greifswald, Greifswald, Germany. [238]Cardiology Division, Department of Pediatrics, University of California, San Francisco, Oakland, CA, USA. [239]Centre for Cancer Genetic Epidemiology, University of Cambridge, Cambridge, UK. [240]Department of Public Health and Primary Care, University of Cambridge, Cambridge, UK. [241]Department of Cardiology, Leiden University Medical Center, Leiden, The Netherlands. [242]Central Diagnostics Laboratory, Division Laboratories, Pharmacy and Biomedical Genetics, University Medical Center Utrecht, Utrecht University, Utrecht, The Netherlands. [243]Department of Human Genetics, Leiden University Medical Center, Leiden, The Netherlands. [244]Laboratory Genetic Metabolic Diseases, Department of Clinical Chemistry, Amsterdam UMC, University of Amsterdam, Amsterdam, The Netherlands. [245]Core Facility Metabolomics, Amsterdam UMC, University of Amsterdam, Amsterdam, The Netherlands. [246]Department of Cardiology, Division Heart and Lungs, University Medical Center Utrecht, Utrecht University, Utrecht, The Netherlands. [247]Department of Cardiology, University of Groningen, University Medical Center Groningen, Groningen, The Netherlands. [248]Department of Epidemiology and Biostatistics, School of Public Health, Tongji Medical College, Huazhong University of Science and Technology, Wuhan, China. [249]NIHR Barts Cardiovascular Biomedical Research Centre, Barts and The London School of Medicine and Dentistry, Queen Mary University of London, London, UK. [250]Department of Ophthalmology, Beijing Tongren Hospital, Capital Medical University, Beijing, China. [251]Department of Epidemiology and Biostatistics, MRC-PHE Centre for Environment and Health, School of Public Health, Imperial College London, London, UK. [252]Department of Dermatology, Medical University of Vienna, Vienna, Austria. [253]Department of Research and Innovation, Division of Clinical Neuroscience, Oslo University Hospital, Oslo, Norway. [254]Department of Neurology, Oslo University Hospital, Oslo, Norway. [255]MRC Unit for Lifelong Health and Ageing at UCL, Institute of Cardiovascular Science, University College London, London, UK. [256]Institute of Genetic Epidemiology, Faculty of Medicine and Medical Center, University of Freiburg, Freiburg, Germany. [257]Department of Medicine IV – Nephrology and Primary Care, Faculty of Medicine and Medical Center, University of Freiburg, Freiburg, Germany. [258]Institute of Molecular Medicine, McGovern Medical School, University of Texas Health Science Center at Houston, Houston, TX, USA. [259]Department of Medical Biochemistry, Kurume University School of Medicine, Kurume, Japan. [260]Rush Alzheimer's Disease Center, Rush University Medical Center, Chicago, IL, USA. [261]Department of Neurological Sciences, Rush University Medical Center, Chicago, IL, USA. [262]Department of Genetic Medicine, Weill Cornell Medicine-Qatar, Doha, Qatar. [263]Department of Computer and Systems Engineering, Alexandria University, Alexandria, Egypt. [264]Department of Cardiology, German Heart Centre Munich, Technical University Munich, Munich, Germany. [265]Department of Cardiology, Ealing Hospital, London North West University Healthcare NHS Trust, London, UK. [266]Department of Epidemiology and Biostatistics, Imperial College London, London, UK. [267]Cardiovascular Epidemiology Unit, Department of Public Health and Primary Care, University of Cambridge, Strangeways Research Laboratory, Cambridge, UK. [268]Department of Computational Medicine and Bioinformatics, University of Michigan, Ann Arbor, MI, USA. [269]Analytic and Translational Genetics Unit, Massachusetts General Hospital, Boston, MA, USA. [270]Stanley Center for Psychiatric Research, Broad Institute of MIT and Harvard, Cambridge, MA, USA. [271]Department of Nutrition, Gillings School of Global Public Health, University of North Carolina at Chapel Hill, Chapel Hill, NC, USA. [272]Carolina Population Center, University of North Carolina at Chapel Hill, Chapel Hill, NC, USA. [273]Department of Clinical Genetics, Erasmus MC, Rotterdam, The Netherlands. [274]Department of Radiology and Nuclear Medicine, Erasmus MC, Rotterdam, The Netherlands. [275]Latin American Brain Health (BrainLat), Universidad Adolfo Ibáñez, Santiago, Chile. [276]Unidad de Investigacion de Enfermedades Metabolicas and Direction of Nutrition, Instituto Nacional de Ciencias Medicas y Nutricion, Mexico City, Mexico. [277]Escuela de Medicina y Ciencias de la Salud, Tecnologico de Monterrey, Monterrey, Mexico. [278]Department of Epidemiology and Dean's Office, College of Public Health, University of Kentucky, Lexington, KY, USA. [279]Institute of Cardiovascular Science, Faculty of Population Health Sciences, University College London, London, UK. [280]Health Data Research UK and Institute of Health Informatics, University College London, London, UK. [281]KG Jebsen Center for Genetic Epidemiology, Department of Public Health and Nursing, Faculty of Medicine and Health Sciences, Norwegian University of Science and Technology, NTNU, Trondheim, Norway. [282]HUNT Research Centre, Department of Public Health and Nursing, Norwegian University of Science and Technology, Levanger, Norway. [283]Department of Endocrinology, Clinic of Medicine, St. Olavs Hospital, Trondheim University Hospital, Trondheim, Norway. [284]Department of Nephrology, University Hospital Regensburg, Regensburg, Germany. [285]Geriatric Unit, Azienda Toscana Centro, Florence, Italy. [286]Systems Genomics Laboratory, School of Biotechnology, Jawaharlal Nehru University (JNU), New Delhi, India. [287]Institute of Molecular Genetics, National Research Council of Italy, Pavia, Italy. [288]Human Genetics Center and Department of Epidemiology, University of Texas Health Science Center at Houston, Houston, TX, USA. [289]Department of Nephrology and Rheumatology, Kliniken Südostbayern, Regensburg, Germany. [290]KfH Kidney Center Traunstein, Traunstein, Germany. [291]Center for Genomics and Personalized Medicine (CGPM), Aarhus University, Aarhus, Denmark. [292]Bioinformatics Research Centre, Aarhus University, Aarhus, Denmark. [293]USC-Office of Population Studies Foundation, University of San Carlos, Cebu City, Philippines. [294]Department of Nutrition and Dietetics, University of San Carlos, Cebu City, Philippines. [295]Human Genomics Laboratory, Pennington Biomedical Research Center, Baton Rouge, LA, USA. [296]Department of Biochemistry, Wake Forest School of Medicine, Medical Center Boulevard, Winston-Salem, NC, USA. [297]Department of Clinical Biochemistry, Lillebaelt Hospital, Vejle, Denmark. [298]Institute of Regional Health Research, University of Southern Denmark, Odense, Denmark. [299]Clinic of Medicine, St. Olavs Hospital,

Trondheim University Hospital, Trondheim, Norway. [300]Lee Kong Chian School of Medicine, Nanyang Technological University, Singapore, Singapore. [301]Imperial College Healthcare NHS Trust, Imperial College London, London, UK. [302]Adjunct Faculty, JSS University Academy of Higher Education and Research (JSSAHER), JSS (Deemed to be) University, Mysuru, India. [303]Ophthalmology and Visual Sciences Academic Clinical Program (Eye ACP), Duke-NUS Medical School, Singapore, Singapore. [304]Department of Medical Genetics, Oslo University Hospital, Oslo, Norway. [305]Department of Medical Research, Bærum Hospital, Vestre Viken Hospital Trust, Gjettum, Norway. [306]Department of Neurology, Division of Vascular Neurology, University of Maryland School of Medicine, Baltimore, MD, USA. [307]Baltimore Veterans Affairs Medical Center, Department of Neurology, Baltimore, MD, USA. [308]Unidad de Investigación Médica en Bioquímica, Hospital de Especialidades, Centro Médico Nacional Siglo XXI, Instituto Mexicano del Seguro Social, Mexico City, Mexico. [309]Dipartimento di Scienze Biomediche, Università degli Studi di Sassari, Sassari, Italy. [310]Intermountain Heart Institute, Intermountain Medical Center, Murray, UT, USA. [311]Department of Surgery, University of Pennsylvania, Philadelphia, PA, USA. [312]Corporal Michael J. Crescenz VA Medical Center, Philadelphia, PA, USA. [313]Department of Vascular Surgery, University Medical Center Utrecht, University of Utrecht, Utrecht, The Netherlands. [314]Department of Human Nutrition, Wageningen University, Wageningen, The Netherlands. [315]Center for Translational and Computational Neuroimmunology, Department of Neurology, Columbia University Medical Center, New York, NY, USA. [316]Department of Oncology, University of Cambridge, Cambridge, UK. [317]Department of General Practice, Amsterdam Public Health Institute, Amsterdam UMC, location VUmc, Amsterdam, The Netherlands. [318]Department of Nutrition, Harvard T.H. Chan School of Public Health, Boston, MA, USA. [319]Cardiac Arrhythmia Service, Massachusetts General Hospital, Boston, MA, USA. [320]Cardiovascular Research Center, Massachusetts General Hospital, Boston, MA, USA. [321]Department of Clinical Sciences in Malmö, Division of Geriatric Medicine, Lund University, Malmö, Sweden. [322]Molecular Cardiology Division, Victor Chang Cardiac Research Institute, Darlinghurst, New South Wales, Australia. [323]Cardiology Department, St Vincent's Hospital, Darlinghurst, New South Wales, Australia. [324]Faculty of Medicine, UNSW Sydney, Kensington, New South Wales, Australia. [325]Translational Gerontology Branch, National Institute on Aging, National Institutes of Health, Baltimore, MD, USA. [326]Robertson Center for Biostatistics, University of Glasgow, Glasgow, UK. [327]Human Genetics Center, School of Public Health, University of Texas Health Science Center at Houston, Houston, TX, USA. [328]Department of Public Health and Clinical Medicine, Umeå University, Umeå, Sweden. [329]Department of Internal Medicine, Wake Forest School of Medicine, Medical Center Boulevard, Winston-Salem, NC, USA. [330]Faculty of Veterinary and Agricultural Science, University of Melbourne, Parkville, Victoria, Australia. [331]Agriculture Victoria Research, Department of Jobs, Precincts and Regions, Bundoora, Victoria, Australia. [332]Injury Prevention Research Center, University of North Carolina at Chapel Hill, Chapel Hill, NC, USA. [333]Division of Physical Therapy, University of North Carolina at Chapel Hill, Chapel Hill, NC, USA. [334]Centro de Investigacion en Salud Poblacional Instituto Nacional de Salud Publica and Centro de Estudios en Diabetes, Cuernavaca, Mexico. [335]Division of Human Genetics, Children's Hospital of Philadelphia, Philadelphia, PA, USA. [336]Departments of Pediatrics and Genetics, Perelman School of Medicine, University of Pennsylvania, Philadelphia, PA, USA. [337]Division of Endocrinology and Diabetes, Children's Hospital of Philadelphia, Philadelphia, PA, USA. [338]Faculty of Medicine, University of Iceland, Reykjavik, Iceland. [339]Department of Preventive Medicine, Keck School of Medicine of USC, Los Angeles, CA, USA. [340]Department of Pediatrics, Perelman School of Medicine, University of Pennsylvania, Philadelphia, PA, USA. [341]Division of Pulmonary Medicine, Children's Hospital of Philadelphia, Philadelphia, PA, USA. [342]Institute of Biomedical and Clinical Science, University of Exeter Medical School, Exeter, UK. [343]Cardiovascular Health Research Unit, Department of Epidemiology, University of Washington, Seattle, WA, USA. [344]Department of Paediatrics, Yong Loo Lin School of Medicine, National University of Singapore, Singapore, Singapore. [345]Khoo Teck Puat - National University Children's Medical Institute, National University Health System, Singapore, Singapore. [346]Department of Internal Medicine II, Division of Cardiology, Medical University of Vienna, Vienna, Austria. [347]Menzies Research Institute Tasmania, University of Tasmania, Hobart, Tasmania, Australia. [348]Centre for Eye Research Australia, Royal Victorian Eye and Ear Hospital, University of Melbourne, Melbourne, Victoria, Australia. [349]Lions Eye Institute, Centre for Ophthalmology and Vision Science, University of Western Australia, Perth, Western Australia, Australia. [350]Cardiology Division, Massachusetts General Hospital, Boston, MA, USA. [351]Department of Genetics, Shanghai-MOST Key Laboratory of Heath and Disease Genomics, Chinese National Human Genome Center and Shanghai Industrial Technology Institute, Shanghai, China. [352]Department of Internal Medicine, University of Utah, Salt Lake City, UT, USA. [353]Australian Centre for Precision Health, Clinical and Health Sciences, University of South Australia, Adelaide, South Australia, Australia. [354]South Australian Health and Medical Research Institute, Adelaide, South Australia, Australia. [355]Department of Environmental and Preventive Medicine, Jichi Medical University School of Medicine, Shimotsuke, Japan. [356]Department of Epidemiology and Population Health, Albert Einstein College of Medicine, Bronx, NY, USA. [357]Division of Endocrinology, Diabetes and Metabolism, School of Medicine, Ohio State University, Columbus, OH, USA. [358]Center for Life Course Health Research, Faculty of Medicine, University of Oulu, Oulu, Finland. [359]Unit of Primary Health Care, Oulu University Hospital, OYS, Oulu, Finland. [360]Department of Life Sciences, College of Health and Life Sciences, Brunel University London, Uxbridge, UK. [361]The Eye Hospital, School of Ophthalmology and Optometry, Wenzhou Medical University, Wenzhou, China. [362]Einthoven Laboratory for Experimental Vascular Medicine, LUMC, Leiden, The Netherlands. [363]Netherlands Heart Institute, Utrecht, The Netherlands. [364]Department of Clinical Physiology, Tampere University Hospital, Tampere, Finland. [365]Department of Clinical Physiology, Finnish Cardiovascular Research

Center - Tampere, Faculty of Medicine and Health Technology, Tampere University, Tampere, Finland. [366]Laboratory of Complex Trait Genomics, Department of Computational Biology and Medical Sciences, Graduate School of Frontier Sciences, The University of Tokyo, Tokyo, Japan. [367]Department of Ophthalmology, The Catholic University of Korea Incheon St. Mary's Hospital, Incheon, Republic of Korea. [368]NIHR Oxford Biomedical Research Centre, Churchill Hospital, Oxford, UK. [369]German Centre for Cardiovascular Research (DZHK), partner site Munich Heart Alliance, Munich, Germany. [370]Radboud University Medical Center, Radboud Institute for Health Sciences, Department of Urology, Nijmegen, The Netherlands. [371]Corneal Dystrophy Research Institute, Yonsei University College of Medicine, Seoul, Republic of Korea. [372]Saevit Eye Hospital, Goyang, Republic of Korea. [373]Department of Biochemistry, College of Medicine, Ewha Womans University, Seoul, Republic of Korea. [374]Department of Cardiology, University Heart and Vascular Center UKE Hamburg, Hamburg, Germany. [375]Institute of Cardiovascular Sciences, College of Medical and Dental Sciences, University of Birmingham, Birmingham, UK. [376]German Center for Cardiovascular Research, partner site Hamburg/Kiel/Lübeck, Hamburg, Germany. [377]Atrial Fibrillation NETwork, Münster, Germany. [378]Department of Epidemiology and Public Health, UCL Institute of Epidemiology and Health Care, University College London, London, UK. [379]Healthy Longevity Translational Research Programme, Yong Loo Lin School of Medicine, National University of Singapore, Singapore, Singapore. [380]University of Helsinki and Department of Medicine, Helsinki University Hospital, Helsinki, Finland. [381]Minerva Foundation Institute for Medical Research, Helsinki, Finland. [382]Department of Preventive Cardiology, Lipoprotein Apheresis Unit and Lipid Disorders Clinic, Metropolitan Hospital, Athens, Greece. [383]MRC-PHE Centre for Environment and Health, Imperial College London, London, UK. [384]National Heart and Lung Institute, Imperial College London, London, UK. [385]Medical Department III – Endocrinology, Nephrology, Rheumatology, University of Leipzig Medical Center, Leipzig, Germany. [386]Institute for Social and Economic Research, University of Essex, Colchester, UK. [387]Institute of Clinical Medicine, Internal Medicine, University of Eastern Finland and Kuopio University Hospital, Kuopio, Finland. [388]Department of Medicine, University of Colorado at Denver, Aurora, CO, USA. [389]Berlin Institute of Health at Charité – Universitätsmedizin Berlin, Berlin, Germany. [390]Epidemiology Program, University of Hawaii Cancer Center, Honolulu, HI, USA. [391]Department of Internal Medicine, Ewha Womans University School of Medicine, Seoul, Republic of Korea. [392]Department of Epidemiology and Biostatistics, Peking University Health Science Center, Beijing, China. [393]Peking University Center for Public Health and Epidemic Preparedness and Response, Beijing, China. [394]Institute of Epidemiology and Biobank Popgen, Kiel University, Kiel, Germany. [395]Key Laboratory of Systems Health Science of Zhejiang Province, Hangzhou Institute for Advanced Study, University of Chinese Academy of Sciences, Chinese Academy of Sciences, Hangzhou, China. [396]Department of Clinical Medicine, Faculty of Health and Medical Sciences, University of Copenhagen, Copenhagen, Denmark. [397]Division of Cardiovascular Medicine and Abboud Cardiovascular Research Center, University of Iowa Hospitals and Clinics, Iowa City, IA, USA. [398]Alliance for Human Development, Lunenfeld-Tanenbaum Research Institute, Sinai Health System, Toronto, Ontario, Canada. [399]Department of Medical Epidemiology and Biostatistics, Karolinska Institutet, Stockholm, Sweden. [400]Division of Cardiology, University of California, San Francisco, San Francisco, CA, USA. [401]Department of Medicine, Internal Medicine, Lausanne University Hospital, Lausanne, Switzerland. [402]University of Lausanne, Lausanne, Switzerland. [403]SYNLAB Academy, SYNLAB Holding Deutschland, Mannheim, Germany. [404]Clinical Institute of Medical and Chemical Laboratory Diagnostics, Medical University of Graz, Graz, Austria. [405]Department of Medicine, Division of Cardiology, Duke University School of Medicine, Durham, NC, USA. [406]Duke Molecular Physiology Institute, Duke University School of Medicine, Durham, NC, USA. [407]Psychiatric Genetics, QIMR Berghofer Medical Research Institute, Brisbane, Queensland, Australia. [408]Geriatric Medicine, Institute of Medicine, Sahlgrenska Academy, University of Gothenburg, Gothenburg, Sweden. [409]Geriatrics Research and Education Clinical Center, Baltimore Veterans Administration Medical Center, Baltimore, MD, USA. [410]Centre for Vision Research and Department of Ophthalmology, Westmead Millennium Institute of Medical Research, University of Sydney, Sydney, New South Wales, Australia. [411]Department of Public Health and Primary Care, Leiden University Medical Center, Leiden, The Netherlands. [412]Usher Institute of Population Health Sciences and Informatics, University of Edinburgh, Edinburgh, UK. [413]Electrophysiology Section, Division of Cardiovascular Medicine, Perelman School of Medicine, University of Pennsylvania, Philadelphia, PA, USA. [414]Department of Medicine, University of North Carolina at Chapel Hill, Chapel Hill, NC, USA. [415]Department of Chronic Diseases, Norwegian Institute of Public Health, Oslo, Norway. [416]Department of Pain Management and Research, Oslo University Hospital, Oslo, Norway. [417]Institute of Human Genetics, School of Medicine and University Hospital Bonn, Bonn, Germany. [418]Sahlgrenska University Hospital, Department of Drug Treatment, Gothenburg, Sweden. [419]Laboratorio de Inmunogenómica y Enfermedades Metabólicas, Instituto Nacional de Medicina Genómica, CDMX, Mexico City, Mexico. [420]Paavo Nurmi Centre, Sports and Exercise Medicine Unit, Department of Physical Activity and Health, University of Turku, Turku, Finland. [421]Institute for Precision Health, David Geffen School of Medicine at UCLA, Los Angeles, CA, USA. [422]Pat MacPherson Centre for Pharmacogenetics and Pharmacogenomics, Division of Population Health and Genomics, School of Medicine, University of Dundee, Ninewells Hospital and Medical School, Dundee, UK. [423]Centre Nutrition, Santé et Société (NUTRISS), Institute of Nutrition and Functional Foods, Université Laval, Québec City, Quebec, Canada. [424]IBE-Chair of Epidemiology, LMU Munich, Neuherberg, Germany. [425]Population, Policy and Practice, UCL Great Ormond Street Hospital Institute of Child Health, London, UK. [426]Department of Medicine, University of Pennsylvania, Philadelphia, PA, USA. [427]Department of Clinical Physiology and Nuclear Medicine, Turku University Hospital, Turku, Finland. [428]Hero DMC Heart Institute, Dyanand Medical College, Ludhiana, India. [429]Second Department of Cardiology, Medical School, National and Kapodistrian University of Athens, University General Hospital Attikon, Athens, Greece. [430]Division of Biostatistics, Washington University School of Medicine, St Louis, MO, USA. [431]Genetics, Merck Sharp & Dohme, Kenilworth, NJ, USA. [432]Department of Epidemiology, University of Washington, Seattle, WA, USA. [433]Department of Endocrinology and Metabolism, Kyung Hee University School of Medicine, Seoul, Korea. [434]Department of Public Health, Clinicum, Faculty of Medicine, University of Helsinki, Helsinki, Finland. [435]Departments of Medicine, Pharmacology, and Biomedical Informatics, Vanderbilt University Medical Center, Nashville, TN, USA. [436]Department of Epidemiology and Data Science, Amsterdam Public Health Institute, Amsterdam Cardiovascular Sciences Institute, Amsterdam UMC, location VUmc, Amsterdam, The Netherlands. [437]Department of Cardiology and Department of Medicine, Columbia University, New York, NY, USA. [438]Department of Pediatrics, Section of Genetics, College of Medicine, University of Oklahoma Health Sciences Center, Oklahoma City, OK, USA. [439]Department of Pharmaceutical Sciences, University of Oklahoma Health Sciences Center, Oklahoma City, OK, USA. [440]Department of Physiology, University of Oklahoma Health Sciences Center, Oklahoma City, OK, USA. [441]Oklahoma Center for Neuroscience, University of Oklahoma Health Sciences Center, Oklahoma City, OK, USA. [442]Institute of Cardiovascular and Medical Sciences, University of Glasgow, Glasgow, UK. [443]Gottfried Schatz Research Center (for Cell Signaling, Metabolism and Aging), Medical University of Graz, Graz, Austria. [444]Institute of Nutritional Science, University of Potsdam, Nuthetal, Germany. [445]Deutsches Herzzentrum München, Cardiology, Deutsches Zentrum für Herz- und Kreislaufforschung (DZHK) – Munich Heart Alliance, and Technische Universität München, München, Germany. [446]School of Biomedical Science and Pharmacy, University of Newcastle, New Lambton Heights, New South Wales, Australia. [447]Department of Medicine, Vanderbilt University Medical Center, Nashville, TN, USA. [448]Central University of Punjab, Bathinda, India. [449]Department of Medicine I, University Hospital, LMU Munich, Munich, Germany. [450]Department of Cardiology, Clinical Sciences, Lund University and Skåne University Hospital, Lund, Sweden. [451]The Wallenberg Laboratory, Department of Molecular and Clinical Medicine, Institute of Medicine, Gothenburg University and the Department of Cardiology, Sahlgrenska University Hospital, Gothenburg, Sweden. [452]Wallenberg Center for Molecular Medicine and Lund University Diabetes Center, Lund University, Lund, Sweden. [453]Population Health Research Institute, St George's, University of London, London, UK. [454]Molecular Epidemiology Section, Department of Biomedical Data Sciences, Leiden University Medical Center, Leiden, The Netherlands. [455]Department of Genetics, Stanford University School of Medicine, Stanford, CA, USA. [456]Department of Medicine, Faculty of Medicine, Université de Montréal, Montreal, Quebec, Canada. [457]Helsinki University Central Hospital, Research Program for Clinical and Molecular Metabolism, University of Helsinki, Helsinki, Finland. [458]Folkhälsan Research Center, Helsinki, Finland. [459]Department of Public Health, University of Helsinki, Helsinki, Finland. [460]Diabetes Research Group, King Abdulaziz University, Jeddah, Saudi Arabia. [461]Unidad de Biología Molecular y Medicina Genómica, Instituto de Investigaciones Biomédicas, UNAM, Mexico City, Mexico. [462]Instituto Nacional de Ciencias Médicas y Nutrición Salvador Zubirán, Mexico City, Mexico. [463]Yong Loo Lin School of Medicine, National University of Singapore and National University Health System, Singapore, Singapore. [464]Milken Institute School of Public Health, The George Washington University, Washington, DC, USA. [465]Department Geriatric Medicine, Amsterdam Public Health, Amsterdam UMC location University of Amsterdam, Amsterdam, The Netherlands. [466]Department of Epidemiology and Data Science, Amsterdam UMC, Amsterdam, The Netherlands. [467]Interfaculty Institute for Genetics and Functional Genomics, University Medicine Greifswald, Greifswald, Germany. [468]Unidad de Investigación Médica en Epidemiología Clínica, Hospital de Especialidades, Centro Médico Nacional Siglo XXI, Instituto Mexicano del Seguro Social, Mexico City, Mexico. [469]Institute of Cellular Medicine, Newcastle University, Newcastle upon Tyne, UK. [470]Department of Population and Public Health Sciences, Keck School of Medicine of USC, Los Angeles, CA, USA. [471]Department of Physiology and Neuroscience, Keck School of Medicine of USC, Los Angeles, CA, USA. [472]USC Diabetes and Obesity Research Institute, Keck School of Medicine of USC, Los Angeles, CA, USA. [473]Lundbeck Foundation Center for GeoGenetics, GLOBE Institute, University of Copenhagen, Copenhagen, Denmark. [474]School of Chinese Medicine, China Medical University, Taichung, Taiwan. [475]Diabetes Unit, KEM Hospital and Research Centre, Pune, India. [476]Kurume University School of Medicine, Kurume, Japan. [477]Division of Cancer Control and Population Sciences, UPMC Hillman Cancer Center, University of Pittsburgh, Pittsburgh, PA, USA. [478]Department of Epidemiology, Graduate School of Public Health, University of Pittsburgh, Pittsburgh, PA, USA. [479]TUM School of Medicine, Technical University of Munich and Klinikum Rechts der Isar, Munich, Germany. [480]Institute of Clinical Medicine, Faculty of Medicine, University of Oslo, Oslo, Norway. [481]Department of Population Health Sciences, Geisinger, Danville, PA, USA. [482]Vanderbilt Genetics Institute, Division of Genetic Medicine, Vanderbilt University Medical Center, Nashville, TN, USA. [483]Department of Medicine, Veterans Affairs Boston Healthcare System, Boston, MA, USA. [484]Department of Epidemiology, Emory University Rollins School of Public Health, Atlanta, GA, USA. [485]Atlanta VA Health Care System, Decatur, GA, USA. [486]Princess Al-Jawhara Al-Brahim Centre of Excellence in Research of Hereditary Disorders (PACER-HD), King Abdulaziz University, Jeddah, Saudi Arabia. [487]The Mindich Child Health and Development Institute, Icahn School of Medicine at Mount Sinai, New York, NY, USA. [488]School of Life Sciences, Westlake University, Hangzhou, China. [489]Westlake Laboratory of Life Sciences and Biomedicine, Hangzhou, China. [490]Department of Human Genetics, University of Michigan, Ann Arbor, MI, USA. [491]McDonnell Genome Institute and Department of Medicine, Washington University School of Medicine, St Louis, MO, USA. [492]Laboratory of Statistical Immunology, Immunology Frontier Research Center (WPI-IFReC), Osaka, Japan. [493]Integrated Frontier Research for Medical Science Division, Institute for Open and

Transdisciplinary Research Initiatives, Osaka University, Osaka, Japan. [494]Programs in Metabolism and Medical and Population Genetics, Broad Institute of MIT and Harvard, Cambridge, MA, USA. [495]Departments of Pediatrics and Genetics, Harvard Medical School, Boston, MA, USA. [496]Present address: Department of Mathematics and Statistics, St Cloud State University, St Cloud, MN, USA. [497]Genentech, South San Francisco, CA, USA. [498]Present address: Laboratory for Systems Genetics, RIKEN Center for Integrative Medical Sciences, Kanagawa, Japan. [499]Department of Genome Informatics, Graduate School of Medicine, The University of Tokyo, Tokyo, Japan. [500]These authors contributed equally: Loïc Yengo, Sailaja Vedantam, Eirini Marouli. [501]These authors jointly supervised this work: Yukinori Okada, Andrew R. Wood, Peter M. Visscher, Joel N. Hirschhorn. *A list of authors and their affiliations appears online.

**23andMe Research Team**

Adam Auton[12], Gabriel Cuellar Partida[12], Yunxuan Jiang[12] & Jingchunzi Shi[12]

Full lists of members and their affiliations appear in the Supplementary Information.

**VA Million Veteran Program**

Saiju Pyarajan[5,57,81] & Yan Sun[484,485]

Full lists of members and their affiliations appear in the Supplementary Information.

**DiscovEHR (DiscovEHR and MyCode Community Health Initiative)**

Jason E. Miller[188,189], Shefali S. Verma[67] & Anne E. Justice[9,481]

Full lists of members and their affiliations appear in the Supplementary Information.

**eMERGE (Electronic Medical Records and Genomics Network)**

Damien Croteau-Chonka[57]

Full lists of members and their affiliations appear in the Supplementary Information.

**Lifelines Cohort Study**

Ilja M. Nolte[41], Harold Snieder[41], Peter M. Visscher[1,501] & Judith M. Vonk[41]

Full lists of members and their affiliations appear in the Supplementary Information.

**The PRACTICAL Consortium**

Sonja I. Berndt[15], Stephen Chanock[15], Christopher Haiman[339] & Loic Le Marchand[390]

Full lists of members and their affiliations appear in the Supplementary Information.

**Understanding Society Scientific Group**

Meena Kumari[386]

Full lists of members and their affiliations appear in the Supplementary Information.

## Methods

A summary of the methods, together with a full description of genome-wide association analyses and follow-up analyses is described below. Written informed consent was obtained from every participant in each study, and the study was approved by relevant ethics committees (Supplementary Table 1).

### Quality control checks of individual studies

All study files were checked for quality using the software EasyQC[53] that was adapted to the format from RVTESTS (versions listed in Supplementary Table 2)[54]. The checks performed included allele frequency differences with ancestry-specific reference panels, total number of markers, total number of markers not present in the reference panels, imputation quality, genomic inflation factor and trait transformation. We excluded two studies that did not pass our quality checks in the data.

### GWAS meta-analysis

We first performed ancestry-group-specific GWAS meta-analyses of 173 studies of EUR, 56 studies of EAS, 29 studies of AFR, 11 studies of HIS and 12 studies of SAS. Meta-analyses within ancestry groups were performed as described before[19,20] using a modified version of RAREMETAL[55] (v.4.15.1), which accounts for multi-allelic variants in the data. Study-specific GWASs are described in Supplementary Tables 1–3. Details about imputation procedures implemented by each study are also given in Supplementary Table 2. We kept in our analyses SNPs with an imputation accuracy ($r^2_{INFO}$) > 0.3, Hardy–Weinberg Equilibrium (HWE) $P$ value ($P_{HWE}$) > $10^{-8}$ and a minor allele count (MAC) > 5 in each study. Next, we performed a fixed-effect inverse variance weighted meta-analysis of summary statistics from all five ancestry groups GWAS meta-analysis using a custom R script using the R package meta (see 'URLs' section).

### Hold-out sample from the UK Biobank

We excluded 56,477 UK Biobank (UKB) participants from our discovery GWAS for following analyses including quantification of population stratification. More precisely, our hold-out EUR sample consists of 17,942 sibling pairs and 981 trios (two parents and one child) plus all UKB participants with an estimated genetic relationship larger than 0.05 with our set of sibling pairs and trios. We identified 14,587 individuals among these 56,477 UKB participants who were unrelated (unrelatedness was determined as when the genetic relationship coefficient estimated from HM3 SNPs was lower than 0.05) to each other and used their data to quantify the variance explained by SNPs within GWS loci (described below) and the prediction accuracy of PGSs.

### COJO analyses

We performed COJO analyses of each of the five ancestry group-specific GWAS meta-analyses using the software GCTA (version v.1.93)[6,7]. We used default parameters for all ancestry groups except in AFR and HIS, for which we found that default parameters could yield biased estimates of joint SNP effects because of long-range LD. This choice is discussed in Supplementary Note 1. The GCTA-COJO method implements a stepwise model selection that aims at retaining a set of SNPs the joint effects of which reach genome-wide significance, defined in this study as $P < 5 \times 10^{-8}$. In addition to GWAS summary statistics, COJO analyses also require genotypes from an ancestry-matched sample that is used as a LD reference. For all sets of genotypes used as LD reference panels, we selected HM3 SNPs with $r^2_{INFO}$ > 0.3 and $P_{HWE}$ > $10^{-6}$. For EUR, we used genotypes at 1,318,293 HM3 SNPs (MAC > 5) from 348,501 unrelated EUR participants in the UKB as our LD reference. For EAS, we used genotypes at 1,034,263 quality-controlled (MAF > 1%, SNP missingness < 5%) HM3 SNPs from a merged panel of $n$ = 5,875 unrelated participants from the UKB ($n$ = 2,257) and Genetic Epidemiology Research on Aging (GERA; $n$ = 3,618). Data from the GERA study were obtained from the database of Genotypes and Phenotypes (dbGaP; accession number: phs000788.v2.p3.c1) under project 15096. For SAS, we used genotypes at 1,222,935 HM3 SNPs (MAC > 5; SNP missingness < 5%) from 9,448 unrelated individuals. For AFR, we used genotypes at 1,007,949 quality-controlled (MAF > 1%, SNP missingness < 5%) HM3 SNPs from a merged panel of 15,847 participants from the Women's Health Initiative (WHI; $n$ = 7,480), and the National Heart, Lung, and Blood Institute's Candidate Gene Association Resource (CARe[56], $n$ = 8,367). Both WHI and CARe datasets were obtained from dbGaP (accession numbers: phs000386 for WHI; CARe including phs000557.v4.p1, phs000286.v5.p1, phs000613.v1.p2, phs000284.v2.p1, phs000283.v7.p3 for ARIC, JHS, CARDIA, CFS and MESA cohorts) and processed following the protocol provided by the dbGaP data submitters. After excluding samples with more than 10% missing values and retaining only unrelated individuals, our final LD reference included data from $n$ = 10,636 unrelated AFR individuals. For HIS, we used genotypes at 1,246,763 sequenced HM3 SNPs (MAF > 1%) from $n$ = 4,883 unrelated samples from the Hispanic Community Health Study/Study of Latinos (HCHS/SOL; dbGaP accession number: phs001395.v2.p1) cohorts. Finally, we performed a COJO analysis of the combined meta-analysis of all ancestries (referred to as META$_{FE}$ in the main text) using 348,501 unrelated EUR participants in the UKB as the reference panel.

To assess whether SNPs detected in non-EUR were independent of signals detected in EUR, we performed another COJO analysis of ancestry groups GWAS by fitting jointly SNPs detected in EUR with those detected in each of the non-EUR GWAS meta-analyses. For each non-EUR GWAS, we performed a single-step COJO analysis only including SNPs identified in that non-EUR GWAS and for which the LD squared correlation ($r^2_{LD}$) with any of the EUR signals (marginally or conditionally GWS) is lower than 0.8 in both EUR and corresponding non-EUR data. Single-step COJO analyses were performed using the --cojo-joint option of GCTA, which does not involve model selection and simply approximates a multivariate regression model in which all selected SNPs on a chromosome are fitted jointly. LD correlations used in these filters were estimated in ancestry-matched samples of the 1000 Genomes Project (1KGP; release 3). More specifically, LD was estimated in 661 AFR, 347 HIS (referred to with the AMR label in 1KGP), 504 EAS, 503 EUR and 489 SAS 1KGP participants. We used the same LD reference samples in these analyses as for our main discovery analysis described at the beginning of the section.

### $F_{ST}$ calculation and (stratified) LD score regression

We used two statistics to evaluate whether an EUR LD reference could approximate well enough the LD structure in our trans-ancestry GWAS meta-analysis. The first statistic that we used is the Wright fixation index[57], which measures allele frequency divergence between two populations. We used the Hudson's estimator of $F_{ST}$[58] as previously recommended[59] to compare allele frequencies from our META$_{FE}$ with that from our EUR GWAS meta-analysis and an independent replication sample from the EBB. The other statistic that we used is the attenuation ratio statistic from the LD score regression methodology. These LD score regression analyses were performed using version 1.0 of the LDSC software and using LD scores calculated from EUR participants in the 1KGP (see 'URLs' section). Moreover, we performed a stratified LD score regression analysis to quantify the enrichment of height heritability in 97 genomic annotations curated and described previously[40]. as the baseline-LD model. Annotation-weighted LD scores used for those analyses were also calculated using data from 1KGP (see 'URLs' section).

### Density of GWS signal and enrichment near OMIM genes

We defined the density of independent signals around each GWS SNP as the number of other independent associations identified with COJO within a 100-kb window on both sides. Therefore, a SNP with no other associations within 100 kb has a density of 0, whereas a SNP colocalizing with 20 other GWS associations within 100 kb will have a density of 20.

We quantified the standard error of the mean signal density across the genome using a leave-one-chromosome-out jackknife procedure. We then quantified the enrichment of 462 curated OMIM[18] genes near GWS SNPs with a large signal density, by counting the number of OMIM genes within 100 kb of a GWS SNP, then comparing that number for SNPs with a density of 0 and those with a density of at least 1. The strength of the enrichment was measured using an odds ratio calculated from a 2×2 contingency table: 'presence/absence of an OMIM gene' versus 'density of 0 or larger than 0'. To assess the significance of the enrichment, we simulated the distribution of enrichment statistics for a random set of 462 length-matched genes. We used 22 length classes (<10 kb; between $i \times 10$ kb and $(i+1) \times 10$ kb, with $i = 1,...,9$; between i × 100 kb and $(i+1) \times$ 100 kb, with $i = 1,...,10$; between 1 Mb and 1.5 Mb; between 1.5 Mb and 2 Mb; and >2 Mb) to match OMIM genes with random genes. OMIM genes within a given length class were matched with the same number of non-OMIM genes present in the class. We sampled 1,000 random sets of genes and calculated for each them an enrichment statistic. Enrichment $P$ value was calculated as the number of times enrichment statistics of random genes exceeded that of OMIM genes. The list of OMIM genes is provided in Supplementary Table 11.

**Genomic colocalization of GWS SNPs identified across ancestries**
We assessed the genomic colocalization between 2,747 GWS SNPs identified in non-EUR (Supplementary Tables 5–8) and 9,863 GWS SNPs identified in EUR (Supplementary Table 4) by quantifying the proportion of EUR GWS SNPs identified within 100 kb of any non-EUR GWS SNP. We tested the statistical significance of this proportion by comparing it with the proportion of EUR GWS SNPs identified within 100 kb of random HM3 SNPs matched with non-EUR GWS SNPs on 24 binary functional annotations[39].

These 24 annotations (for example, coding or conserved) are thoroughly described in a previous study[39] and were downloaded from https://alkesgroup.broadinstitute.org/LDSCORE/baselineLD_v2.1_annots/.

Our matching strategy consists of three steps. First, we calibrated a statistical model to predict the probability for a given HM3 SNP to be GWS in any of our non-EUR GWAS meta-analyses as a function of their annotation. For that, we used a logistic regression of the non-EUR GWS status (1 = if the SNP is GWS in any of the non-EUR GWAS; 0 = otherwise) onto the 24 annotations as regressors. Second, we used that model to predict the probability to be GWS in non-EUR. Thirdly, we used the predicted probability to sample (with replacement) 1,000 random sets of 2,747 SNPs. Finally, we estimated the proportion of EUR GWS SNPs within 100 kb of SNPs in each sampled SNP set. We report in the main text the mean and s.d. over these 1,000 proportions.

To validate our matching strategy, we compared the mean value of each of these 24 annotations (for example, proportion of coding SNPs) between non-EUR GWS SNPs and each of the 1,000 random sets of SNPs, using a Fisher's exact test. For each of the 24 annotations, both the mean and median $P$ value were greater than 0.6 and the proportion of $P$ values < 5% was less than 1%, suggesting no significant differences in the distribution of these 24 annotations between non-EUR GWS SNPs and matched SNPs.

**Replication analyses**
To assess the replicability of our results, we tested whether the correlation $\rho_b$ of estimated SNP effects between our discovery GWAS and our replication sample of 49,160 participants of the EBB was statistically different from 1. We used the estimator of $\rho_b$ from a previous study[60], which accounts for sampling errors in both discovery and replication samples. Standard errors were calculated using a leave-one-SNP-out jackknife procedure. We quantified the correlation of marginal and also that of joint SNP effects. Joint SNP effects in our replication sample were obtained by performing a single-step COJO analysis of GWAS summary statistics from our EBB sample, using the same LD reference as in the discovery GWAS. Correlation of SNP effects were

calculated after correcting SNP effects for winner's curse using a previously described method[12]. We provide the R scripts used to apply these corrections and estimate the correlation of SNP effects (see 'URLs' section). The expected proportion, $E[P]$, of sign-consistent SNP effects between discovery and replication was calculated using the quadrant probability of a standard bivariate Gaussian distribution with correlation $E[\rho_b]$, denoting the expected correlation between estimated SNP effects in the discovery and replication sample:

$$E[P] = \frac{1}{2} + \frac{\sin^{-1}(E[\rho_b t])}{\pi}, \tag{1}$$

where $\sin^{-1}$ denotes the inverse of the sine function and $E[\rho_b]$ the expectation of the $\rho_b$ statistic under the assumption that the true SNP effects are the same across discovery and replications cohorts. $E[\rho_b]$ was calculated as

$$E[\rho_b] = \frac{\sigma_b^2}{\sqrt{(\sigma_b^2 + [1 - \sigma_b^2 h_d]/(N_d h_d))(\sigma_b^2 + [1 - \sigma_b^2 h_r]/(N_r h_r))}}, \tag{2}$$

where $N_d$ and $N_r$ denote the sizes of the discovery and replication samples, respectively; $h_d$ and $h_r$ the average heterozygosity under Hardy–Weinberg equilibrium (that is, 2 × MAF × (1 − MAF)) across GWS SNPs in the discovery and replication samples, respectively; and $\sigma_b^2$ the mean per-SNP variance explained by GWS SNPs, which we calculated (as per ref. [60].) as the sample variance of estimated SNP effects in the discovery sample minus the median squared standard error.

**Variance explained by GWS SNPs and loci**
We estimated the variance explained by GWS SNPs using the genetic relationship-based restricted maximum likelihood (GREML) approach implemented in GCTA[1,7]. This approach involves two main steps: (i) calculation of genetic relationships matrices (GRM); and (ii) estimation of variance components corresponding to each of these matrices using a REML algorithm. We partitioned the genome in two sets containing GWS loci on the one hand and all other HM3 SNPs on the other hand. GWS loci were defined as non-overlapping genomic segments containing at least one GWS SNP and such that GWS SNPs in adjacent loci are more than 2 × 35 kb away from each other (that is, a 35-kb window on each side). We then calculated a GRM based on each set of SNPs and estimated jointly a variance explained by GWS alone and that explained by the rest of the genome. We performed these analyses in multiple samples independent of our discovery GWAS, which include participants of diverse ancestry. Details about the samples used for these analyses are provided below. We extended our analyses to also quantify the variance explained by GWS loci using alternative definitions based on a window size of 0 kb and 10 kb around GWS SNPs (Supplementary Figs. 18 and 19).

We also repeated our analyses using a random set of 12,111 SNPs matched with GWS SNPs on MAF and LD. Loci for these 12,111 random SNPs were defined similarly as for GWS loci. To match random SNPs with GWS SNPs on MAF and LD, we first created 28 MAF-LD classes of HM3 SNPs (7 MAF classes × 4 LD score classes). MAF classes were defined as <1%; between 1% and 5%; between 5% and 10%; between 10% and 20%; between 20% and 30%; between 30% and 40%; and between 40% and 50%. LD score classes were defined using quartiles of the HM3 LD score distribution. We next matched GWS SNPs in each of the 28 MAF-LD classes, with the same number of SNPs randomly sampled from that MAF-LD class.

**Prediction analyses**
Height was first mean-centred and scaled to variance 1 within each sex. We quantified the prediction accuracy of height predictors as the difference between the variance explained by a linear regression model of sex-standardized height regressed on the height predictor, age, 20

genotypic principal components and study-specific covariates (full model) minus that explained by a reduced linear regression not including the height predictor. Genetic principal components were calculated from LD pruned HM3 SNPs ($r^2_{LD} < 0.1$). We used height of siblings or parents as a predictor of height as well as various polygenic scores (PGSs) calculated as a weighted sum of height-increasing alleles. The direction and magnitude of these weights was determined by estimated SNP effects from our discovery GWAS meta-analyses. No calibration of tuning parameters in a validation was performed.

**Between-family prediction.** We analysed two classes of PGS. The first class is based on SNPs ascertained using GCTA-COJO. We applied GCTA-COJO to ancestry-specific and cross-ancestry GWAS meta-analyses using an ancestry-matched and an EUR LD reference, respectively. We compared PGSs based on SNPs ascertained at different significance thresholds: $P < 5 \times 10^{-8}$ (GWS: reported in the main text) and $P < 5 \times 10^{-7}$, $P < 5 \times 10^{-6}$ and $P < 5 \times 10^{-5}$. For all COJO-based PGS, we used estimated joint effects to calculate the PGS. The second class of PGS uses weights for all HM3 SNPs obtained from applying the SBayesC method[28] to ancestry-specific and cross-ancestry GWAS meta-analyses with ancestry-matched and EUR-specific LD matrices, respectively. The SBayesC method is a Bayesian PGS-method implemented in the GCTB software (v.2.0), which uses the same prior as the LDpred method[61,62]. In brief, SBayesC models the distribution of joint effects of all SNPs using a two-component mixture distribution. The first component is a point-mass Dirac distribution on zero and the other component a Gaussian distribution (for each SNP) with mean 0 and a variance parameter to estimate. Full LD matrices (that is, not sparse) were calculated using GCTB across around 250 overlapping (50% overlap) blocks of around 8,000 SNPs (average size is around 20 Mb). These LD matrices were calculated using the same sets of genotypes used for COJO analyses (described above). We ran SBayesC in each block separately with 100,000 Monte Carlo Markov Chain iterations. In each run, we initialized the proportion of causal SNPs in a block at 0.0001 and the heritability explained by SNPs in the block at 0.001. Posterior SNP effects of SNPs present in two blocks were meta-analysed using inverse-variance meta-analysis.

Prediction accuracy was quantified in 61,095 unrelated individuals from three studies, including 33,001 participants of the UKB who were not included in our discovery GWAS (that is, 14,587 EUR; 9,257 SAS; 6,911 AFR and 2,246 EAS; Methods section 'Samples used for prediction and estimation of variance explained'); 14,058 EUR participants from the Lifelines cohort study; and 8,238 HIS and 5,798 AFR participants from the PAGE study.

**Within-family prediction.** The prediction accuracy of sibling's height was assessed in 17,942 unrelated sibling pairs from the UKB. Those pairs were determined by intersecting the list of UKB sibling pairs determined by Bycroft et al.[63] with a list of genetically determined European ancestry participants from the UKB also described previously[3]. We then filtered the resulting list for SNP-based genetic relationship between members of different families to be smaller than 0.05. The prediction accuracy of parental height (each parent and their average) was assessed in 981 unrelated trios obtained as described above by crossing information from Bycroft et al.[63] (calling of relatives) with that from Yengo et al.[3] (calling of European ancestry participants). We quantified the within-family variance explained by PGS as the squared correlation of height difference between siblings with PGS difference between siblings. We describe in Supplementary Note 4 how familial information and PGS were combined to generate a single predictor.

**Samples used for prediction and estimation of variance explained**
We quantified the accuracy of a PGS based on GWS SNPs as well as the variance explained by SNPs within GWS loci, in eight different datasets independent of our discovery GWAS meta-analyses. These datasets

include two samples of EUR from the UKB ($n = 14,587$) and the Lifelines study ($n = 14,058$), two samples of AFR from the UKB ($n = 6,911$) and the PAGE study ($n = 8,238$), two samples of EAS ($n = 2,246$) from the UKB and the China Kadoorie Biobank (CKB; $n = 47,693$), one sample of SAS from the UKB ($n = 9,257$) and one sample of HIS from the PAGE study ($n = 4,939$). Analyses were adjusted for age, sex, 20 genotypic principal components and study-specific covariates (for example, recruitment centres). Genotypes of EUR UKB participants were imputed to the Haplotype Reference Consortium (HRC) and to a combined reference panel including haplotypes from the 1KG Project and the UK10K Project. To improve variant coverage in non-EUR participants of UKB, we re-imputed their genotypes to the 1KG reference panel, as described previously[38]. Lifelines samples were imputed to the HRC panel. PAGE and CKB were imputed to the 1KG reference panel. Standard quality control ($r^2_{INFO} > 0.3$, $P_{HWE} > 10^{-6}$ and MAC > 5) were applied to imputed genotypes in each dataset.

## Contribution of LD and MAF to the loss of prediction accuracy
We defined the EUR-to-AFR relative accuracy as the ratio of prediction accuracies from an AFR sample over that from a EUR sample. We used a previously published method[38] to quantify the expectation of that relative accuracy under the assumption that causal variants and their effects are shared between EUR and AFR, whereas MAF and LD structures can differ. In brief, this method contrasts LD and MAF patterns within 100-kb windows around each GWS SNPs and uses them to predict the expected loss of accuracy. As previously described[38], we used genotypes from 503 EUR and 661 AFR participants of the 1KGP as a reference sample to estimate ancestry-specific MAF and LD correlations between GWS SNPs and SNPs in their close vicinity, and defined candidate causal variants as any sequenced SNP with an $r^2_{LD} > 0.45$ with a GWS SNP within that 100-kb window. Standard errors were calculated using a delta-method approximation as previously described[38].

## Down-sampled GWAS analyses
In addition to our EUR GWAS meta-analysis and our trans-ancestry meta-analysis (META_FE), we re-analysed five down-sampled GWASs as shown in Table 2. These down-sampled GWASs include various iterations of previous efforts of the GIANT consortium and have a sample size varying between around 130,000 and 2.5 million (EUR participants from 23andMe). To ensure sufficient genomic coverage of HM3 SNPs we imputed GWAS summary statistics from Lango Allen et al.[19], Wood et al.[20] and Yengo et al.[3]. with ImpG-Summary (v.1.0.1)[64] using haplotypes from 1KGP as a LD reference. GWAS summary statistics from Lango Allen et al. only contain $P$ values ($P$), height-increasing alleles and per-SNP sample sizes ($N$). Therefore, we first calculated $Z$-scores ($Z$) from $P$ values assuming that $Z$-scores are normally distributed, then derived SNP effects ($\beta$) and corresponding standard errors (s.e.) using linear regression theory as $\beta = Z / \sqrt{2MAF \times (1 - MAF) \times (N + Z^2)}$ and SE = $\beta/Z$. Imputed GWAS summary statistics from these three studies are made publicly available on the GIANT consortium website (see 'URLs' section). We next performed a COJO analysis of all down-sampled GWAS using genotypes of 348,501 unrelated EUR participants in the UKB as a LD reference panel, as for our META_FE and EUR GWAS meta-analysis.

## Gene prioritization using SMR
We used SMR to identify genes whose expression could mediate the effects of SNPs on height. SMR analyses were performed using the SMR software v.1.03. We used publicly available gene eQTLs identified from two large eQTL studies; namely, the GTEx[65] v.8 and the eQTLgen studies (see 'URLs' section). To ensure that our SMR results robustly reflect causality or pleiotropic effects of height-associated SNPs on gene expression, we only report here significant SMR results (that is, $P < 5 \times 10^{-8}$), which do not pass the heterogeneity in dependent instrument (HEIDI) test (that is, $P > 0.01$; Methods). The significance threshold for the HEIDI test was chosen on the basis of recommendations from another study[66].

## Selection of OMIM genes

To generate a list of genes that are known to underlie syndromes of abnormal skeletal growth, we queried the Online Mendelian Inheritance in Man database (OMIM; https://www.omim.org/). From July 2019 to August 2020, we performed queries using search terms of "short stature", "tall stature", "overgrowth", "skeletal dysplasia" and "brachydactyly." We then used the free text descriptions in OMIM to manually curate the resulting combined list of genes, as well as genes in our earlier list from Wood et al.[20] and all genes listed as causing skeletal disease in an online endocrine textbook (https://www.endotext.org/, accessed September 2020). For short stature, we only included genes that underlie syndromes in which short stature was either consistent (less than −2 s.d. in the vast majority of patients with data recorded), or present in multiple families or sibships and accompanied by (a) more severe short stature (−3 s.d.), (b) presence of skeletal dysplasia (beyond poor bone quality/fractures); or (c) presence of brachydactyly, shortened digits, disproportionate short stature or limb shortening (not simply absence of specific bones). We removed genes underlying syndromes in which short stature was likely to be attributable to failure to thrive, specific metabolic disturbances, intestinal failure or enteropathy and/or very severe disease (for example, early lethality or severe neurological disease). For tall stature or overgrowth, we only included genes underlying syndromes in which tall stature was consistent (more than +2 s.d. in the vast majority of patients with data recorded) or present in multiple families or sibships and accompanied by either (a) more severe tall stature (>+3 s.d.) or (b) arachnodactyly. For brachydactyly, we required more than only fifth finger involvement, and that brachydactyly be either consistent (present in the vast majority of patients) or accompanied by consistent short stature or other skeletal dysplasias. For skeletal dysplasias, we only considered genes that underlie syndromes in which the skeletal dysplasia involved long bones or the spine and was accompanied by short stature, brachydactyly or limb or digit shortening. We also included all genes in a list we generated in Lango Allen et al.[19], which was curated using similar criteria. The resulting list contained 536 genes, of which 462 (Supplementary Table 11) are autosomal on the basis of annotation from PLINK (https://www.cog-genomics.org/static/bin/plink/glist-hg19).

## URLs

GIANT consortium data files: https://portals.broadinstitute.org/collaboration/giant/index.php/GIANT_consortium_data_files. Analysis script for within- and across-ancestry meta-analysis: https://github.com/loic-yengo/ScriptsForYengo2022_HeightGWAS/blob/main/run-meta-analyses-within-ancestries.R and https://github.com/loic-yengo/ScriptsForYengo2022_HeightGWAS/blob/main/run-meta-analyses-across-ancestries.R. Analysis script for correction of winner's curse: https://github.com/loic-yengo/ScriptsForYengo2022_HeightGWAS/blob/main/WC_correction.R. Genotypes from 1KG: https://ftp.1000genomes.ebi.ac.uk/vol1/ftp/release/20130502/. eQTL data for SMR: GTEx v.8: https://yanglab.westlake.edu.cn/data/SMR/GTEx_V8_cis_eqtl_summary.html; eQTLgen: https://www.eqtlgen.org/cis-eqtls.html. Annotation-weighted LD scores for stratified LD score regression analyses: https://alkesgroup.broadinstitute.org/LDSCORE/LDSCORE/. LDSC software: https://github.com/bulik/ldsc.

## Reporting summary

Further information on research design is available in the Nature Research Reporting Summary linked to this article.

## Data availability

Summary statistics for ancestry-specific and multi-ancestry GWASs (excluding data from 23andMe) as well as SNP weights for polygenic scores derived in this study are made publicly available on the GIANT consortium website (see 'URLs' for GIANT consortium data files). GWAS summary statistics derived involving 23andMe participants will be made available to qualified researchers under an agreement with 23andMe that protects the privacy of participants. Application for data access can be submitted at https://research.23andme.com/dataset-access/. We used genotypes from various publicly available databases to estimate linkage disequilibrium correlations required for conditional analyses and genome-wide prediction analyses. These databases include the UK Biobank under project 12505 and the database of Genotypes and Phenotypes (dbGaP) under project 15096. Accession numbers for dbGaP datasets are phs000788.v2.p3.c1, phs000386, phs000557.v4.p1, phs000286.v5.p1, phs000613.v1.p2, phs000284.v2.p1, phs000283.v7.p3 and phs001395.v2.p1 cohorts. Details for each dbGaP dataset are given in the Methods. Source data are provided with this paper.

## Code availability

We used publicly available software tools for all analyses. These software tools are listed in the main text and in the Methods. Source data are provided with this paper.

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

**Acknowledgements** We acknowledge the participants in each cohorts contributing to this study. Additional acknowledgements are provided in the Supplementary Information. Support for title page creation and format was provided by AuthorArranger, a tool developed at the National Cancer Institute. This research was supported by the following funding bodies. US National Institutes of Health (NIH): 75N92021D00001, 75N92021D00002, 75N92021D00003, 75N92021D00004, 75N92021D00005, AA07535, AA10248, AA014041, AA13320, AA13321, AA13326, DA12854, U01 DK062418, HHSN268201800005I, HHSN268201800007I, HHSN268201800003I, HHSN268201800006I, HHSN268201800004I, R01 CA55069, R35 CA53890, R01 CA80205, R01 CA144034, HHSN268201200008I, EY022310, 1X01HG006934-01, R01DK118427, R21DK105913, HHSN268201200036C, HHSN268200800007C, HHSN268200960009C, HHSN268201800001C, N01HC55222, N01HC85079, N01HC85080, N01HC85081, N01HC85082, N01HC85083, N01HC85086, 75N92021D00006, U01HL080295, R01HL085251, R01HL087652, R01HL105756, R01HL103612, R01HL120393, U01HL130114, R01AG023629, UL1TR001881, DK063491, R01 HL095056, 1R01HL139731 (S. A. Lubitz), R01HL157635 (S. A. Lubitz, P.T.E.), 1R01HL092577 (P.T.E.), K24HL105780 (P.T.E.), HHSC268200782096C, R01 DK087914, R01 DK066358, R01 DK053591, 1K08HG010155 (A.V.K.), 1U01HG011719 (A.V.K.), U01 HG004436, P30 DK072488, HHSN268200782096C, U01 HG 004446, R01 NS45012, U01 NS069208-01, R01-NS114045 (J.W.C.), R01-NS100178 (J.W.C.), R01-NS105150 (J.W.C.), HL043851, HL080467, CA047988, UM1CA182913, U01HG008657, U01HG008685, U01HG008672, U01HG008666, U01HG006379, U01HG008679, U01HG008680, U01HG008684, U01HG008673, U01HG008701, U01HG008676, U01HG008664, U54MD007593, UL1TR001878, R01-DK062370 (M.B.), R01-DK072193 (K.L.M.), intramural project number 1Z01-HG000024 (F.S.C.), N01-HG-65403, DA044283, DA042755, DA037904, AA009367, DA005147, DA036216, 5-P60-AR30701, 5-P60-AR49465, N01-AG-1-2100, HHSN271201200022C, National Institute on Aging Intramural Research Program, R-35-HL135824 (C.J.W.), AA-12502, AA-00145, AA-09203, AA15416, K02AA018755, UM1 CA186107, P01 CA87969, R01 CA49449, U01 CA176726, R01 CA67262, UM1CA167552, CA141298, P01CA055075, CA141298, HL54471, HL54472, HL54473,

HL54495, HL54496, HL54509, HL54515, U24 MH068457-06, R01D0042157-01A1, RO1 MH58799-03, MH081802, 1RC2MH089951-01, 1RC2 MH089995, R01 DK092127-04, R01DK110113 (R.J.F.L.), R01DK075787 (R.J.F.L.), R01DK107786 (R.J.F.L.), R01HL142302 (R.J.F.L.), R01HG010297 (R.J.F.L.), R01DK124097 (R.J.F.L.), R01HL151152 (R.J.F.L.), R01-HL046380, KL2-RR024990, R35-HL135818, R01-HL113338, R35HL135818 (S. Redline), HL 046389 (S. Redline), HL113338 (S. Redline), K01 HL135405 (B.E.C.), R03 HL154284 (B.E.C.), R01HL086718, HG011052 (X. Zhu), N01-HC-25195, HHSN268201500001I, N02-HL-6-4278, R01-DK122503, U01AG023746, U01AG023712, U01AG023749, U01AG023755, U01AG023744, U19AG063893, R01-DK-089256, R01HL117078, R01 HL09135701, R01 HL091357, R01 HL104135, R37-HL045508, R01-HL053353, R01-DK075787, U01-HL054512, R01-HL074166, R01-HL086718, R01-HG003054, U01HG004423, U01HG004446, U01HG004438, DK078150, TW005596, HL085144, RR020649, ES010126, DK056350, R01DK072193, R01 HD30880, R01 AG065357, R01DK104371, R01HL108427, Fogarty grant D43 TW009077, 263 MD 9164, 263 MD 821336, N.1-AG-1-1, N.1-AG-1-2111, HHSN268201800013I, HHSN268201800014I, HHSN268201800015I, HHSN268201800010I, HHSN268201800011I and HHSN268201800012I, KL2TR002490 (L.M.R.), T32HL129982 (L.M.R.), R01AG056477, R01AG034454, R01 HD056465, U01 HL054457, U01 HL054464, U01 HL054481, R01 HL119443, R01 HL087660, U01AG009740, RC2 AG036495, RC4 AG039029, U01AG009740 (W. Zhao.), RC2 AG036495 (W. Zhao.), RC4 AG039029 (W. Zhao.), 75N92020D00001, HHSN268201500003I, N01-HC-95159, 75N92020D00005, N01-HC-95160, 75N92020D00002, N01-HC-95161, 75N92020D00003, N01-HC-95162, 75N92020D00006, N01-HC-95163, 75N92020D00004, N01-HC-95164, 75N92020D00007, N01-HC-95165, N01-HC-95166, N01-HC-95167, N01-HC-95168, N01-HC-95169, UL1-TR-000040, UL1-TR-001079, UL1-TR-001420, N02-HL-64278, UL1TR001881, DK063491, R01-HL088457, R01-HL-60030, R01-HL067974, R01-HL-55005, R01-HL 067974, R01HL111249, R01HL111249-04S1, U01HL54527, U01HL54498 R01HL54684, EY014684-04S1, EY014684, EY014684-04S1, DK063491, S10OD017985, S10RR025141, UL1TR002243, UL1TR000445, UL1RR024975, U01HG004798, R01NS032830, RC2GM092618, P50GM115305, U01HG006378, U19HL065962, R01HD074711, 5K08HL135275 (R.W.M.), R01 HL77398 (B.L.), NR013520 (Y.V.S.), DK125187 (Y.V.S.), HHSN268201700000I, HHSN268201700002I, HHSN268201700003I, HHSN268201700004I, HHSN268201700005I, R01HL087641, R01HL086694, U01HG004402, HHSN268200625226C, UL1RR025005, U01HG007416, R01DK101855, 15GRNT25880008, N01-HC65233, N01-HC65234, N01-HC65235, N01-HC65236, N01-HC65237, U01HG007376, HHSN268201100046C, HHSN268201100001C, HHSN268201100002C, HHSN268201100003C, HHSN268201100004C, HHSN271201100004C, N01-AG-6-2101, N01-AG-6-2103, N01-AG-6-2106, R01-AG028050, R01-NR012459, P30AG10161, P30AG72975, R01AG17917, RF1AG15819, R01AG30146, U01AG46152, U01AG61256, AG000513, R01 HD58886, R01 HD100406, N01-HD-1-3228, N01-HD-1-3329, N01-HD-1-3330, N01-HD-1-3331, N01-HD-1-3332, N01-HD-1-3333, UL1 TR000077, R01 HD056465 (S.F.A.G.), R01 HG010067 (S.F.A.G.), R01CA64277, R01CA15847, UM1CA182910, R01CA148677, R01CA144034, UM1 CA182876, R01DK075787, R01DK075787 (J.N.H.), ZIA CP010152-20, U19 CA 148537-01, U01 CA188392, X01HG007492, HHSN268201200008I, Z01CP010119, R01-CA080122, R01-CA056678, R01-CA082664, R01-CA092579, K05-CA175147, P30-CA015704, CA063464, CA054281, CA098758, CA164973, R01CA128813, K25 HL150334 (R. E. Mukamel), DP2 ES030554 (P.-R.L.), U19 CA148065, CA128978, 1U19 CA148537, 1U19 CA148065, 1U19 CA148112, U01 DK062418, U01-DK105535 (M.I.M.), R01HL24799 NIHHLB, U01 DK105556, DK093757 (K.L.M.), HL129982 and T32 HL007055 (H. H. Highland). Wellcome Trust: 068545/Z/02, 076113/B/04/Z, Strategic Award 079895, 090532/Z/09/Z, 203141/Z/16/Z, 201543/B/16/Z, 084723/Z/08/Z, 090532, 098381, 217065/Z/19/Z, WT088806, WT09283O/Z/10/Z, 202802/Z/16/Z (N.J.T.), 217065/Z/19/Z (N.J.T.), 216767/Z/19/Z, 104036/Z/14/Z, 098051, WT098051, 212946/Z/18/Z, 202922/Z/16/Z, 104085/Z/14/Z, 088185/Z/09/Z, 221854/Z/20/Z, 212904/Z/18/Z, WT095219MA, 068545/Z/02, 076113, 090532 (M.I.M.), 098381 (M.I.M.), 106130 (M.I.M.), 203141 (M.I.M.), 212259 (M.I.M.), 072960/Z/03/Z, 084726/Z/08/Z, 084727/Z/08/Z, 085475/Z/08/Z, 085475/B/08/Z, 212945/Z/18/Z (J.S.K.). UK Medical Research Council: G0000934, MR/N013166/1 (P.R.H.J.T.), MR/N013166/1 (K.A.K.), U. MC_UU_00007/10, G0601966, G0700931, MRC Integrative Epidemiology Unit MC_UU_00011/1 (N.J.T., R. E. Mitchell), MC_UU_00019/1, G9521010D (the BRIGHT Study), MC_UU_12015/1, MC_PC_13046, MC_PC_13049, MC-PC-14135, MC_UU_00017/1, MC_UU_12026/2, MC_U137686851, K013351, R024227, MC_UU_00007/10, MR/M016560/1, G1001799, MC_PC_20026 (L. J. Smyth). Cancer Research UK: CRUK Integrative Cancer Epidemiology Programme C18281/A29019 (N.J.T.), C16077/A29186, C500/A16896, C5047/A7357, C1287/A10118, C1287/A16563, C5047/A3354, C5047/A10692, C16913/A6135, C5047/A1232, C490/A10124, C1287/A16563, C1287/A10118, C1287/A10710, C12292/A11174, C1281/A12014, C5047/A8384, C5047/A15007, C5047/A10692, C8197/A16565. Australian Research Council: DP0770096 (P. M. Visscher), DP1093502 (M.E.G.), DE200100425 (L. Yengo), FL180100072 (P. M. Visscher). Australian National Health and Medical Research Council: 241944, 389875, 389891, 389892, 389938, 442915, 442981, 496739, 496688, 552485, 613672, 613601, 1011506, 1172917 (S.E.M.), 572613, 403981, 1059711, 1027449, 1044840, 1021858, 974159, 211069, 457349, 512423, 302010, 571013, 1154518 (D.A.M.), 1103329 (A.W.H.), 1186500 (D.F.), 209057, 396414, 1074383, 390130, 1009458, 1113400 (P. M. Visscher, Jian Yang). UK National Institute for Health Research Centres: Barts Biomedical Research Centre (P. Deloukas, S.K.), Comprehensive Biomedical Research Centre Imperial College Healthcare NHS Trust, Health Protection Research Unit on Health Impact of Environmental Hazards, RP-PG-0407-10371, Official Development Assistance award 16/136/68, the University of Bristol NIHR Biomedical Research Centre BRC-1215-2001 (N.J.T.), Academic Clinical Fellowship (S.J.H.), Leicester Cardiovascular Biomedical Research Centre BRC-1215-20010 (C.P.N., P.S.B., N.J.S.), Barts Biomedical Research Centre and Queen Mary University of London, Exeter Clinical Research Facility, Clinical Research Facility and Biomedical Research Centre based at Guy's and St Thomas' NHS Foundation Trust and King's College London (M. Mangino, P.C.), Biomedical Research Centre at The Institute of Cancer Research and The Royal Marsden NHS Foundation Trust, Biomedical Research Centre at the University of Cambridge, Oxford Biomedical Research Centre. European Union: 018996, LSHG-CT-2006-018947, HEALTH-F2-2013-601456, ERA-CVD program grant 01KL1802 (S.W.v.d.L.), 305739, 727565, FP/2007-2013 ERC grant agreement number 310644 MACULA, LSHM-CT-2007-037273, SOC 95201408 05F02, SOC 98200769 05F02, LSHM-CT-2006-037593, 279143, iHealth-T2D 643774, 223004, Marie Sklodowska-Curie grant agreement number 786833 (J.R.), 810645, FP7-HEALTH-F4-2007 grant number 201413 and 9602768, QLG1-CT-2001-01252, LSHG-CT-2006-01894 (I.R., A.F.W., V.V.), 733100, HEALTH-F2-2009-223175, LSHG-CT-2006-01947), HEALTH-F4-2007-201413, QLG2-CT-2002-01254, FP7 project number

602633, H2020 project numbers 634935 and 633784, HEALTH-F2-2009-223175, IMI-SUMMIT program, H2020 grants 755320 and 848146 (S.W.v.d.L.), BigData@Heart grant EU IMI 116074 (P. Kirchhof). European Regional Development Fund: 2014-2020.4.01.15-0012, 2014-2020.4.01.16-0125 (A. Metspalu), 539/2010 A31592, 2014-2020.4.01.16-0030. Netherlands Heart Foundation: CVON 2011/B019 (S.W.v.d.L.), CVON 2017-20 (S.W.v.d.L.), NHS2010B233, NHS2010B280, CVON 2014–9 (M.R.). British Heart Foundation: Centre for Research Excellence (H.W.), RG/14/5/30893 (P. Deloukas), FS/14/66/3129 (O.G.), SP/04/002, SP/16/4/32697 (C.P.N.), CH/1996001/9454, 32334 (M. Kivimaki.), RG/17/1/32663, FS/13/43/30324 (P. Kirchhof), PG/17/30/32961 (P. Kirchhof), PG/20/22/35093 (P. Kirchhof). US Department of Veterans Affairs: Baltimore Geriatrics Research, Education, and Clinical Center; IK2-CX001780 (S.M.D.), I01-BX004821, MVP 001, IK2-CX001907 (S. Raghavan). American Heart Association: 18SFRN34250007 (S. A. Lubitz), 18SFRN34110082 (P.T.E.), 17IBDG33700328 (J.W.C.), 15GPSPG23770000 (J.W.C.), 15POST24470131 (C.N.S.), 17POST33650016 (C.N.S.), 19CDA34760258 (H.X.). Leducq Fondation: 'PlaqOmics' (Ather-Express, S.W.v.d.L), 14CVD01 (P.T.E.). Netherlands Organization for Scientific Research NWO: GB-MW 940-38-011, ZonMW Brainpower grant 100-001-004, ZonMw Risk Behavior and Dependence grant 60-60600-97-118, ZonMw Culture and Health grant 261-98-710, GB-MaGW 480-01-006, GB-MaGW 480-07-001, GB-MaGW 452-04-314, GB-MaGW 452-06-004, 175.010.2003.005, 481-08-013, 481-11-001, Vici 016.130.002, 453-16-007/2735, Gravitation 024.001.003, 480-05-003, NWO/SPI 56-464-14192, 480-15-001/674, ZonMW grant number 916.19.151 (H.H.H.A.), ZonMw grant 95103007, 175.010.2005.011, 911-03-012, ZonMW grant 6130.0031, VIDI 016-065-318 (D.P.), Vidi 016.096.309. European Research Council: ERC-2017-STG-757364, ERC-CoG-2015-681466, CoG-2015_681742_NASCENT (I.J.), ERC-2011-StG 280559-SEPI, ERC-STG-2015-679242, 742927, ERC-230374. Swedish Research Council: 2017-02554, 349-2006-237, 2009-1039, Linné grant number 349-2006-237, 2016-06830 (G.H.), 2017-00641, grant for the Swedish Infrastructure for Medical Population-based Life-course Environmental Research. Novo Nordisk Foundation: 12955 (B.F.), NNF18CC0034900, NNF15OC0015896, NNF18CC0034900, NNF15CC0018486, NNF20oC0062294 (T. Karaderi). Academy of Finland: 77299, 124243, 285547 EGEA, 100499, 205585, 118555, 141054, 264146, 308248, 312073, 265240, 263278, Center of Excellence in Complex Disease Genetics grant number 312062, 329202 (M. Kivimaki), 322098, 206374, 251360, 276861, 322098, 286284, 134309 (Eye), 126925, 121584, 124282, 129378 (Salve), 117787 (Gendi), and 41071 (Skidi), 263401 (L. Groop), 267882 (L. Groop), 312063 (L. Groop), 336822 (L. Groop), 312072 (T.T.), 336826 (T.T.). German Federal Ministry of Education and Research: 01ZZ9603, 01ZZ0103, 01ZZ0403, 03IS2061A, 03ZIK012, 01EA1801A (G.E.D.), 01ER0804 (K.-U.E.), BMBF 01ER1206 and BMBF 01ER1507 (I.M.H.), BMBF projects 01EG0401, 01GI0856, 01GI0860, 01GS0820_WB2-C, 01ER1001D, 01GI0205. Additional funding came from the following sources. The University of Newcastle Strategic Initiatives Fund; the Gladys M Brawn Senior Research Fellowship scheme; Vincent Fairfax Family Foundation; The Hunter Medical Research Institute; the Nagahama City Office and the Zeroji Club; the Center of Innovation Program, the Global University Project from the Ministry of Education, Culture, Sports, Science and Technology of Japan; the Practical Research Project for Rare/Intractable Diseases (ek0109070, ek0109283, ek0109196, ek0109348), and the Program for an Integrated Database of Clinical and Genomic Information (kk0205008), from the Japan Agency for Medical Research and Development; Takeda Medical Research Foundation; Astellas Pharma, Inc.; Daiichi Sankyo Co., Ltd.; Mitsubishi Tanabe Pharma Corporation; Otsuka Pharmaceutical Co., Ltd.; Taisho Pharmaceutical Co., Ltd.; Takeda Pharmaceutical Co., Ltd.; JSPS KAKENHI (22H00476), AMED (JP21gm4010006, JP22km0405211, JP22ek0410075, JP22km0405217, JP22ek0109594), JST Moonshot R&D (JPMJMS2021, JPMJMS2024) (Y.O.); Type 1 Diabetes Genetics Consortium; the French Ministry of Research; the Chief Scientist Office of the Scottish Government CZB/4/276 and CZB/4/710; Arthritis Research UK; Royal Society URF (J.F.W.); the Atlantic Philanthropies; the UK Economic and Social Research Council awards ES/L008459/1 and ES/L008459/1; the UKCRC Centre of Excellence for Public Health Northern Ireland; the Centre for Ageing Research and Development in Ireland; the Office of the First Minister and Deputy First Minister; the Health and Social Care Research and Development Division of the Public Health Agency; the Wellcome Trust/Wolfson Foundation; Queen's University Belfast; the Science Foundation Ireland-Department for the Economy Award 15/IA/3152 (NICOLA); NI HSC R&D division STL/5569/19 (L. J. Smyth); the Italian Ministry of Education, University and Research (MIUR) number 5571/DSPAR/2002 (OGP study); GlaxoSmithKline; the Faculty of Biology and Medicine of Lausanne; the Swiss National Science Foundation grants 33CSCO-122661, 33CS30-139468, 33CS30-148401 and 33CS30_177535/1; the Montreal Heart Institute Biobank; the Canadian Institutes of Health Research PJT 156248; the Canada Research Chair Program, Genome Quebec and Genome Canada, and the Montreal Heart Institute Foundation (G.L.); the Strategic Priority CAS Project grant number XDB38000000, Shanghai Municipal Science and Technology Major Project grant number 2017SHZDZX01 and the National Natural Science Foundation of China grant number 81970684; the National Medical Research Council (grants 0796/2003, 1176/2008, 1149/2008, STaR/0003/2008, 1249/2010, CG/SERI/2010, CIRG/1371/2013 and CIRG/1417/2015) and the Biomedical Research Council (grants 08/1/35/19/550 and 09/1/35/19/616) of Singapore; the Ministry of Health, Singapore; the National University of Singapore and the National University Health System, Singapore; the Agency for Science, Technology and Research, Singapore; Merck Sharp & Dohme Corp.; Kuwait Foundation for Advancements of Sciences (The KODGP); the Oogfonds, MaculaFonds, Landelijke Stichting voor Blinden en Slechtzienden, Stichting Blindenhulp, Stichting A.F. Deutman Oogheelkunde Researchfonds; in Mexico, the Fondo Sectorial de Investigación en Salud y Seguridad Social SSA/IMSS/ISSSTECONACYT project 150352; Temas Prioritarios de Salud Instituto Mexicano del Seguro Social 2014-FIS/IMSS/PROT/PRIO/14/34; the Fundación IMSS; Compute Ontario (https://www.computeontario.ca/) and the Digital Research Alliance of Canada (https://alliancecan.ca/); CIHR Operating grants and a CIHR New Investigator Award (E.J.P.); the Westlake Education Foundation (Jian Yang); AstraZeneca; a Miguel Servet contract from the ISCIII Spanish Health Institute number CP17/00142 and co-financed by the European Social Fund (M.S.-L.); the Dutch Ministry of Justice; the European Science Foundation EuroSTRESS project FP-006; Biobanking and Biomolecular Resources Research Infrastructure BBMRI-NL award CP 32; Accare Centre for Child and Adolescent Psychiatry; the Dutch Brain Foundation; the Federal Ministry of Science, Germany award 01 EA 9401; German Cancer Aid award 70-2488-Ha I; the participating Departments, the Division and the Board of Directors of the Leiden University Medical Centre and the Leiden University, Research Profile Area 'Vascular and Regenerative Medicine'; Research Project For Excellence IKY/SIEMENS; the Wake Forest School of Medicine grant M01

RR07122 and Venture Fund; the Greek General Secretary of Research and Technology award PENED 2003; the MRC-PHE Centre for Environment and Health; the Singapore Ministry of Health's National Medical Research Council under its Singapore Translational Research Investigator (STaR) Award NMRC/STaR/0028/2017 (J.C.C.); the German Research Foundation Project-ID 431984000 - SFB 1453 (M. Wuttke, A. Köttgen); the KfH Foundation for Preventive Medicine, and Bayer Pharma AG; the German Research Foundation grant KO 3598/5-1 (A. Köttgen); the Leipzig Research Center for Civilization Diseases; the Medical Faculty of the University of Leipzig; the Free State of Saxony; the Medical Research Funds from Kangbuk Samsung Hospital (H.-N.K.); the Division of Adult and Community Health, Centers for Disease Control and Prevention; AstraZeneca (P.M.R., D.I.C.); Amgen (P.M.R., D.I.C.); a gift from the Smilow family; the Perelman School of Medicine at the University of Pennsylvania; the University of Bristol; a comprehensive list of grants funding is available on the ALSPAC website; the US Centers for Disease Control and Prevention/Association of Schools of Public Health awards S043, S1734 and S3486, and US Centers for Disease Control and Prevention awards U01 DP003206 and U01 DP006266; the Ministry of Cultural Affairs and the Social Ministry of the Federal State of Mecklenburg-West Pomerania; Hjartavernd (the Icelandic Heart Association), and the Althingi (the Icelandic Parliament); Bristol Myers Squibb; the Netherlands Genomics Initiative's Netherlands Consortium for Healthy Aging grant 050-060-810; the Netherlands Heart Foundation grant 2001 D 032 (J.W.J.); the Chief Scientist Office of the Scottish Government Health Directorates award CZD/16/6, the Scottish Funding Council award HR03006; the Stiftelsen Kristian Gerhard Jebsen; Faculty of Medicine and Health Sciences, Norwegian University of Science and Technology; Central Norway Regional Health Authority; the Medical Research Council of Canada and the Canadian Institutes of Health Research grant FRN-CCT-83028 (The Quebec Family Study); Pfizer; the Servier Research Group; Leo Laboratories; Estonian Research Council grants PUT 1371, EMBO Installation grant 3573, and The European Regional Development Fund project no. 2014-2020.4.01.15-0012 (K. Lüll, A. Metspalu); the Estonian Research Council grants PUT PRG687, PRG1291 (EBB, T.E.); the University of Oulu grant number 24000692, Oulu University Hospital grant number 24301140; the Austrian Science Fond grant numbers P20545-P05 and P13180, the Austrian National Bank Anniversary Fund award number P15435, the Austrian Ministry of Science under the aegis of the EU Joint Programme-Neurodegenerative Disease Research (https://www. neurodegenerationresearch.eu/), the Austrian Science Fund P20545-B05, and the Medical University of Graz (ASPS); Wellcome Trust Sanger Institute; the Broad Institute; the Grant of National Center for Global Health and Medicine; the Core Research for Evolutional Science and Technology (CREST) from the Japan Science Technology Agency; the Program for Promotion of Fundamental Studies in Health Sciences, National Institute of Biomedical Innovation Organization; the Grant of National Center for Global Health and Medicine; the German Research Foundation awards HE 3690/7-1 (I.M.H.) and BR 6028/2-1; funds from THL and various domestic foundations (The FINRISK surveys); Business Finland through the Personalized Diagnostics and Care program, SalWe grant number 3986/31/2013; the Finnish Foundation for Cardiovascular Research, the Sigrid Juselius Foundation and University of Helsinki HiLIFE Fellow and Grand Challenge grants (S. Ralhan); the Finnish innovation fund Sitra and Finska Läkaresällskapet (E.W.); Netherlands Twin Registry Repository and the Biobanking and Biomolecular Resources Research Infrastructure awards BBMRI–NL, 184.021.007 and 184.033.111; Amsterdam Public Health and Neuroscience Campus Amsterdam; the Avera Institute for Human Genetics (The Netherlands Twin Register); the KNAW Academy Professor Award PAH/6635 (D.I.B.); the Netherlands Organization for Scientific Research Geestkracht program grant 10-000-1002; the Center for Medical Systems Biology, Biobanking and Biomolecular Resources Research Infrastructure; VU University's Institutes for Health and Care Research and Neuroscience Campus Amsterdam; University Medical Center Groningen; Leiden University Medical Center; the Genetic Association Information Network of the Foundation for the National Institutes of Health; the BiG Grid, the Dutch e-Science Grid; The Lundbeck Foundation; the Stanley Medical Research Institute; the Aarhus and Copenhagen universities and university hospitals; the Danish National Biobank resource supported by the Novo Nordisk Foundation; the Robert Dawson Evans Endowment of the Department of Medicine at Boston University School of Medicine and Boston Medical Center; the Economic & Social Research Council award ES/H029745/1; American Diabetes Association Innovative and Clinical Translational Award 1-19-ICTS-068 (J.M.M.); SIGMA; Consejo Naconal de Ciencia y Tecnologia CONACYT grants 2092, M9303, F677M9407, 251M 2005COI (C.G.-V.); the Danish National Research Foundation; the Danish Pharmacists' Fund; the Egmont Foundation; the March of Dimes Birth Defects Foundation; the Augustinus Foundation; the Health Fund of the Danish Health Insurance Societies; the Oak Foundation fellowship (B.F.); the Nordic Center of Excellence in Health-Related e-Sciences (Xueping Liu); Grants-in-Aid from MEXT numbers 24390169, 16H05250, 15K19242, 16H06277, 19K19434, 20K10514, 21H03206, and a grant from the Funding Program for Next-Generation World-Leading Researchers number LS056; Council of Scientific and Industrial Research, Ministry of Science and Technology, Govt. of India; the Lundbeck Foundation grant number R16-A1694; the Danish Ministry of Health grant number 903516; the Danish Council for Strategic Research grant number 0603-00280B; and The Capital Region Research Foundation; the Danish Research Council; the Danish Centre for Health Technology Assessment; Novo Nordisk; Research Foundation of Copenhagen County; Danish Ministry of Internal Affairs and Health; the Danish Heart Foundation; the Danish Pharmaceutical Association; the Ib Henriksen Foundation; the Becket Foundation; and the Danish Diabetes Association; the Velux Foundation; The Danish Medical Research Council; Danish Agency for Science, Technology and Innovation; The Aase and Ejner Danielsens Foundation; ALK-Abello A/S, Hørsholm, Denmark; and Research Centre for Prevention and Health, the Capital Region of Denmark; the Timber Merchant Vilhelm Bang's Foundation; the Danish Heart Foundation grant number 07-10-R61-A1754-B838-22392F; the Health Insurance Foundation (Helsefonden) grant number 2012B233 (Health2008); TrygFonden grant number 7-11-0213, the Lundbeck Foundation award R155-2013-14070; the Danish Research Council for Independent Research and by Region of Southern Denmark; the Heinz Nixdorf Foundation; the German Research Council DFG projects EI 969/2-3, ER 155/6-1;6-2, HO 3314/2-1;2-2;2-3;4-3, INST 58219/32-1, JO 170/8-1, KN 885/3-1, PE 2309/2-1, SI 236/8-1;9-1;10-1; the Ministry of Innovation, Science, Research and Technology, North Rhine-Westphalia; Academia Sinica; the Office of Population Studies Foundation in Cebu; the China-Japan Friendship Hospital; Ministry of Health, Chinese Human Genome Center at Shanghai; Beijing Municipal Center for Disease Prevention and Control; the National Institute for Nutrition and Health, China Center for Disease Control and Prevention; the Canadian Institutes of Health Research grant MOP-82893; WA Health, Government of Western Australia Future Health WA grant G06302; Safe Work Australia; the University of Western Australia (UWA); Curtin University; Women and Infants Research Foundation; Telethon Kids Institute; Edith Cowan University; Murdoch University; The University of Notre Dame Australia; The Raine Medical Research Foundation; the Italian Ministry of Health award ICS110.1/RF97.71; Hong Kong Kadoorie Charitable Foundation; National Natural Science Foundation of China award 91846303; National Key Research and Development Program of China awards 2016YFC 0900500, 0900501, 0900504, 1303904; the KfH Stiftung Präventivmedizin e.V. (C.A.B.); the Else Kröner-Fresenius-Stiftung (2012_A147); the University Hospital Regensburg; the Deutsche Forschungsgemeinschaft (DFG, German Research Foundation) Project-ID 387509280 – SFB 1350 (Subproject C6); the European Union/EFPIA/ JDRF Innovative Medicines Initiative 2 Joint Undertaking grant number 115974; German Research Foundation DFG BO 3815/4-1 (C.A.B.); the Swedish Foundation for Strategic Research; the Swedish Heart-Lung Foundation; Swedish Heart-Lung Foundation (A. Poveda); VIAgenomics number SP/19/2/344612; the Strategic Cardiovascular Program of Karolinska Institutet and Stockholm County Council; the Foundation for Strategic Research and the Stockholm County Council number 560283; the ALF/LUA research grant in Gothenburg; the Torsten Soderberg Foundation; the ESRC grants ES/S007253/1, ES/T002611/1, and ES/T014083/1 (M. Kumari); Beijing Municipal of Health Reform and Development Project 2019-4 (Beijing Eye Study); the Children's Hospital of Philadelphia; a Research Development Award from the Cotswold Foundation; the Children´s Hospital of Philadelphia Endowed Chair in Genomic Research; the Daniel B. Burke Endowed Chair for Diabetes Research; the Italian Ministry of Universities grant IDF SHARID ARS01_01270; the Assessorato Ricerca Regione Campania grant POR CAMPANIA 2000/2006 MISURA 3.16; the Dutch Ministry of Health, Welfare and Sport; the Dutch Ministry of Economic Affairs; the University Medical Center Groningen (UMCG the Netherlands); University of Groningen and the Northern Provinces of the Netherlands; the UMCG Genetics Lifelines Initiative supported by a Spinoza Grant from NWO; University of Michigan discretionary funds; National Institute of Health, Republic of Korea grants 4845–301, 4851–302, 4851–307; Korea National Institute of Health intramural grant 2019-NG-053-02; the Korea Healthcare Technology R&D Project, Ministry of Health and Welfare, Republic of Korea award A102065; the National Research Foundation of Korea grant 2020R1I1A2075302 (Y.S.C.); the National Research Foundation of Korea Grant NRF-2020R1A2C1012931; the Republic of Croatia Ministry of Science, Education and Sports research grant 108-1080315-0302; the Eye Birth Defects Foundation; the National Science Council, Taiwan grant NSC 98-2314-B-075A-002-MY3; the Taichung Veterans General Hospital, Taichung, Taiwan grant TCVGH-1003001C; AFNET; EHRA; German Centre for Cardiovascular Research (DZHK); German heart Foundation (DSF); the State of Brandenburg DZD grant 82DZD00302; Sanofi; Abbott; the Victor Chang Cardiac Research Institute; NSW Health; the Center for Translational Molecular Medicine, the University Medical Center Groningen; the Dutch Kidney Foundation grant E0.13; the Netherlands Cardiovascular Research Initiative; the Dell Loy Hansen Heart Foundation (M. J. Cutler); Biosense Webster, ImriCor, and ADAS software (S.N.); the Swedish Heart-Lung Foundation grant 2019-0526; Swedish Foundation for Strategic Research grant IRC15-0067; Skåne University Hospital; governmental funding of clinical research within the Swedish National Health Service; the Knut and Alice Wallenberg Foundation (J.G.S.); the Boettcher Foundation Webb Waring Biomedical Research Award (S. Raghavan); the Translational Genomics Research Institute; the Singapore National Medical Research Council grant 1270/2010, and the National Research Foundation, Singapore project 370062002; the Genetic Laboratory of the Department of Internal Medicine, Erasmus MC; the Research Institute for Diseases in the Elderly grant 014-93-015; the Netherlands Genomics Initiative (NGI)/Netherlands Organisation for Scientific Research (NWO) Netherlands Consortium for Healthy Aging project 050-060-810; the Dutch Dairy Association NZO; Netherlands Consortium Healthy Aging, Ministry of Economic Affairs, Agriculture and Innovation project KB-15-004-003; Wageningen University; VU University Medical Center; and Erasmus MC; The Folkhalsan Research Foundation; Nordic Center of Excellence in Disease Genetics; Finnish Diabetes Research Foundation; Foundation for Life and Health in Finland; Finnish Medical Society; Helsinki University Central Hospital Research Foundation; Perklén Foundation; Ollqvist Foundation; Narpes Health Care Foundation; Municipal Heath Care Center and Hospital in Jakobstad; and Health Care Centers in Vasa, Narpes and Korsholm; the Institute of Cancer Research and The Everyman Campaign; The Prostate Cancer Research Foundation; Prostate Research Campaign UK (now PCUK); The Orchid Cancer Appeal; Rosetrees Trust; The National Cancer Research Network UK; The National Cancer Research Institute (NCRI) UK; the Movember Foundation grants D2013-36 and D2013-17; the Morris and Horowitz Families Endowed Professorship; the Swedish Cancer Foundation; Ligue Nationale Contre le Cancer, Institut National du Cancer (INCa); Fondation ARC; Fondation de France; Agence Nationale de sécurité sanitaire de l'alimentation, de l'environnement et du travail (ANSES); Ligue départementale du Val de Marne; the Baden Württemberg Ministry of Science, Research and Arts; The Ronald and Rita McAulay Foundation; Cancer Australia; AICR Netherlands A10-0227; Cancer Council Tasmania; Cancer Councils of Victoria and South Australia; Philanthropic donation to Northshore University Health System; FWO Vlaanderen grants G.0684.12N and G.0830.13N; the Belgian federal government grant KPC_29_023; a Concerted Research Action of the KU Leuven grant GOA/15/017; the Spanish Ministry Council Instituto de Salud Carlos III-FEDER grants PI08/1770, PI09/00773-Cantabria, PI11/01889-FEDER, PI12/00265, PI12/01270, PI12/00715, PI15/00069,and RD09/0076/00036; the Fundación Marqués de Valdecilla grant API 10/09; the Spanish Association Against Cancer (AECC) Scientific Foundation; the Catalan Government DURSI grant 2009SGR1489; the Xarxa de Bancs de Tumors de Catalunya sponsored by Pla Director d'Oncologia de Catalunya (XBTC); the Spanish Ministry of Science and Innovation grant CEX2018-000806-S; the Generalitat de Catalunya; the VicHealth and Cancer Council Victoria; Programa Grupos Emergentes; Cancer Genetics Unit, CHUVI Vigo Hospital; Instituto de Salud Carlos III, Spain; Cancer Australia PdCCRS and Cancer Council Queensland; the California Cancer Research Fund grant 99-00527V-10182; US Public Health Service grants U10CA37429 and 5UM1CA182883; Canadian Cancer Society Research Institute Career Development Award in Cancer Prevention grant 2013-702108; the German Cancer Aid (Deutsche Krebshilfe); The Anthony DeNovi Fund; the Donald C. McGraw Foundation; and the St. Louis Men's Group Against Cancer; UK Biobank project 12505; Westlake Education Foundation (Jian Yang); a Burroughs Wellcome Fund Career Award, the Next Generation Fund at the Broad Institute of MIT and Harvard, and a Sloan Research Fellowship (P.-R.L.); the Consortium for Systems Biology (NCSB), the Netherlands Genomics Initiative (NGI)/Netherlands Organisation for

Scientific Research (NWO); the Government of Canada through Genome Canada and the Canadian Institutes of Health Research grant GPH-129344; the Ministère de l'Économie et de l'Innovation du Québec through Genome Québec grant PSRSIIRI-701; the Quebec Breast Cancer Foundation; the US Department of Defence grant W81XWH-10-1-0341; the Canadian Institutes of Health Research (CIHR) for the CIHR Team in Familial Risks of Breast Cancer; Komen Foundation for the Cure; the Breast Cancer Research Foundation; and the Ovarian Cancer Research Fund; the Economic and Social Research Council grant number ES/M001660/1; Wellcome Investigator and NIHR Senior Investigator (M.I.M.); Council of Scientific and Industrial Research, Government of India grant number BSC0122; the Department of Science and Technology, Government of India through PURSE II CDST/SR/PURSE PHASE II/11 provided to Jawaharlal Nehru University; the Deutsche Forschungsgemeinschaft (DFG, German Research Foundation) Projektnummer 209933838 – SFB 1052; B03, C01; SPP 1629 TO 718/2- 1; the Competitive Research Funding of the Tampere University Hospital grants 9M048 and 9N035; the Finnish Cultural Foundation; the Finnish Foundation for Cardiovascular Research; the Emil Aaltonen Foundation, Finland; Juho Vainio Foundation; Finnish Cardiac Research Foundation; Finnish Ministry of Education and Culture; Yrjö Jahnsson Foundation; C.G. Sundell Foundation; Special Governmental Grants for Health Sciences Research, Turku University Hospital; Foundation for Pediatric Research; and Turku University Foundation; the Social Insurance Institution of Finland; Competitive State Research Financing of the Expert Responsibility area of Kuopio, Tampere and Turku University Hospitals grant X51001; Paavo Nurmi Foundation; Signe and Ane Gyllenberg Foundation; Diabetes Research Foundation of Finnish Diabetes Association; Tampere University Hospital Supporting Foundation; and Finnish Society of Clinical Chemistry; the Italian Ministry of Health—RC 01/21 (M.P.C.) and D70-RESRICGIROTTO (G.G.); 5 per mille 2015 senses CUP: C92F17003560001 (P.G.); the Helmholtz Zentrum München –German Research Center for Environmental Health, which is funded by the German Federal Ministry of Education and Research (BMBF) and by the State of Bavaria; the Department of Innovation, Research, and University of the Autonomous Province of Bolzano-South Tyrol; the Croatian National Center of Research Excellence in Personalized Healthcare grant number KK.01.1.1.01.0010 (O. Polašek) and the Center of Competence in Molecular Diagnostics grant number KK.01.2.2.03.0006 (O. Polašek); the Norwegian Research Council Mobility Grant 24014 and Young Research Talent grant 287086; the South-Eastern Health Authorities PhD-grant 2019122; Vestre Viken Hospital Trust PhD-grant; afib.no - the Norwegian Atrial Fibrillation Research Network; 'Indremedisinsk Forskningsfond' at Bærum Hospital; the Foundation for the National Institutes of Health Accelerating Medicines Partnership award no. HART17AMP; the Dutch String of Pearls Initiative; the Amsterdam University Medical Center, Location VUmc; Academy of Medical Sciences–Wellcome Trust–Government Department of Business, Energy and Industrial Strategy–British Heart Foundation–Diabetes UK Springboard Award SBF006\1134 (A. R. Wood).

**Author contributions** Steering committee: G.R.A., T.L.A., S.I.B., M.B., D.I.C., Y.S.C., T.E., T.M.F., I.M.H., J.N.H., G.L., C.M.L., A.E.L., R.J.F.L., M.I.M., K.L.M., M.C.Y.N., K.E.N., C.J.O., Y.O., F. Rivadeneira, Y.V.S., E.S.T., C.J.W., U.T., P. M. Visscher and R.G.W. Conveners of GIANT working groups: S.I.B., P. Deloukas, J.N.H., A.E.J., G.L., C.M.L., R.J.F.L., E.M., K.L.M., K.E.N., Y.O., C.N.S., R.G.W., C.J.W., A. R. Wood and L. Yengo. Writing group (drafted, edited and commented on manuscript): E. Bartell, J.N.H., G.L., E.M., Y.O., S. Raghavan, S. Sakaue., S. Vedantam, P. M. Visscher, A. R. Wood and L. Yengo. Coordinated or supervised data collection or analysis specific to manuscript: A. Auton, P. Deloukas, T.E., T.M.F., S.E.G., J.N.H., A.E.J., G.L., A.E.L., P.-R.L., Y.O., K.S., U.T., P. M. Visscher, R.G.W., A. R. Wood, Jian Yang and L. Yengo. Data preparation group (checked and prepared data from contributing cohorts for meta-analyses): J. D. Arias, S.I.B., S.-H.C., T.F., S.E.G., M. Graff, H.M.H., Y. Ji, A.E.J., T. Karaderi, A.E.L., K. Lüll, D.E.M., E.M., C.M.-G., M.Mo., A. Moore, S. Rüeger, X.S., C.N.S., S. Vedantam, S. Vrieze, T.W.W., X.Y. and K.L.Y. Meta-analysis working group: J.N.H., E.M., S. Vedantam and L. Yengo. Primary height analysis working group (post meta-analysis): E. Bartell, A.D.B., M. Graff, Y. Jiang, M. Kanai, K. Lin, J. Miao, E.M., R. E. Mukamel, S. Raghavan, S. Sakaue, J. Sidorenko, S. Vedantam, A. R. Wood and L. Yengo. All other authors were involved in the design, management, coordination or analysis of contributing studies.

**Competing interests** Y. Jiang is employed by and holds stock or stock options in 23andMe. T.S.A. is a shareholder in Zealand Pharma A/S and Novo Nordisk A/S. G.C.-P. is an employee of 23andMe. M.E.K. is employed by SYNLAB Holding Deutschland GmbH. H.L.L. receives support from a consulting contract between Data Tecnica International and the National Institute on Aging (NIA), National Institutes of Health (NIH). As of January 2020, A. Mahajan is an employee of Genentech, and a holder of Roche stock. I.N. is an employee and stock owner of Gilead Sciences; this work was conducted before employment by Gilead Sciences. J. Shi is employed by and holds stock or stock options in 23andMe. C. Sidore is an employee of Regeneron. V. Steinthorsdottir is employed by deCODE Genetics/Amgen. Since completing the work contributed to this paper, D.J.T. has left the University of Cambridge and is now employed by Genomics PLC. G.T. is employed by deCODE Genetics/Amgen. S.W.v.d.L. has received Roche funding for unrelated work. H.B. has consultant arrangements with Chiesi Pharmaceuticals and Boehringer Ingelheim. M. J. Caulfield is Chief Scientist for Genomics England, a UK Government company. M. J. Cutler has served on the advisory board or consulted for Biosense Webster, Janssen Scientific Affairs and Johnson & Johnson. S.M.D. receives research support from RenalytixAI and personal consulting fees from Calico Labs, outside the scope of the current research. P.T.E. receives sponsored research support from Bayer AG and IBM Health, and he has served on advisory boards or consulted for Bayer AG, Quest Diagnostics, MyoKardia and Novartis. P. Kirchhof has received support from several drug and device companies active in atrial fibrillation, and has received honoraria from several such companies in the past, but not in the last three years. P. Kirchhof is listed as inventor on two patents held by University of Birmingham (Atrial Fibrillation Therapy WO 2015140571, Markers for Atrial Fibrillation WO 2016012783). G.D.K. has given talks, attended conferences and participated in trials sponsored by Amgen, MSD, Lilly, Vianex and Sanofi, and has also accepted travel support to conferences from Amgen, Sanofi, MSD and Elpen. S. A. Lubitz previously received sponsored research support from Bristol Myers Squibb, Pfizer, Bayer AG, Boehringer Ingelheim, Fitbit and IBM, and has consulted for Bristol Myers Squibb, Pfizer, Bayer AG and Blackstone Life Sciences. S. A. Lubitz is a current employee of Novartis Institute of Biomedical Research. W.M. reports grants and personal fees from AMGEN, BASF, Sanofi, Siemens Diagnostics, Aegerion Pharmaceuticals, Astrazeneca, Danone Research, Numares, Pfizer and Hoffmann LaRoche; personal fees from MSD and Alexion; and grants from Abbott Diagnostics, all outside the submitted work. W.M. is employed with Synlab Holding Deutschland. M.A.N. receives support from a consulting contract between Data Tecnica International and the National Institute on Aging (NIA), National Institutes of Health (NIH). S.N. is a scientific advisor to Circle software, ADAS software, CardioSolv and ImriCor and receives grant support from Biosense Webster, ADAS software and ImriCor. H. Schunkert has received honoraria for consulting from AstraZeneca, MSD, Merck, Daiichi, Servier, Amgen and Takeda Pharma. He has further received honoraria for lectures and/or chairs from AstraZeneca, BayerVital, BRAHMS, Daiichi, Medtronic, Novartis, Sanofi and Servier. P.S. has received research awards from Pfizer. The members of the 23andMe Research Team are employed by and hold stock or stock options in 23andMe. The views expressed in this article are those of the author(s) and not necessarily those of the NHS, the NIHR or the Department of Health. M. I. McCarthy has served on advisory panels for Pfizer, Novo Nordisk and Zoe Global, and has received honoraria from Merck, Pfizer, Novo Nordisk and Eli Lilly and research funding from Abbvie, AstraZeneca, Boehringer Ingelheim, Eli Lilly, Janssen, Merck, Novo Nordisk, Pfizer, Roche, Sanofi Aventis, Servier and Takeda. As of June 2019, M. I. McCarthy is an employee of Genentech, and a holder of Roche stock. C.J.O. is a current employee of Novartis Institute of Biomedical Research. U.T. is employed by deCODE Genetics (Amgen). K.S. is employed by deCODE Genetics (Amgen). A. Auton is employed by and holds stock or stock options in 23andMe. G.R.A. is an employee of Regeneron Pharmaceuticals and owns stock and stock options for Regeneron Pharmaceuticals. C.J.W.'s spouse is employed by Regeneron. A.E.L. is currently employed by and holds stock in Regeneron Pharmaceuticals. J.N.H. holds equity in Camp4 Therapeutics. The remaining authors declare no competing interests.

## Additional information

**Correspondence and requests for materials** should be addressed to Loïc Yengo, Yukinori Okada, Andrew R. Wood, Peter M. Visscher or Joel N. Hirschhorn.

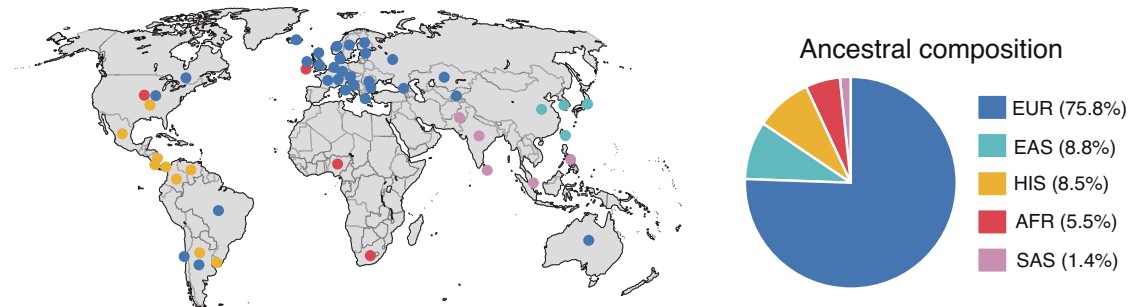

## GIANT consortium: Genetic Investigation of ANthropometric Traits

### Ancestral composition

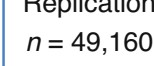

- EUR (75.8%)
- EAS (8.8%)
- HIS (8.5%)
- AFR (5.5%)
- SAS (1.4%)

### Ancestry-specific meta-analysis of height

| European | East Asian | Hispanic | African/ African American | South Asian |
|---|---|---|---|---|
| *n* = 4,080,687 | *n* = 472,730 | *n* = 455,180 | *n* = 293,593 | *n* = 77,890 |


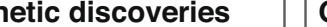

### GWAS meta-analysis of height in 281 studies

**Replication**
*n* = 49,160

**Genetic discoveries**
- Heritability estimation
- Conditional analysis
- Effect size comparison

**Genomic distribution**
- Signal density analysis
- OMIM enrichment

**Polygenic prediction**
- Out-of-sample prediction
- Trans-ancestry comparison
- Within-family analyses

**Saturation of discovery from GWAS**
- Down-sampling analysis
- Variant-, functional-, gene-, and pathway-based metrics
- Cross-population comparison

**Extended Data Fig. 1 | Broad ancestries composition.** Geographical mapping and ancestries composition of 281 studies meta-analysed in this study. Various analyses were performed including (1) dectection of height-associated SNPs (Genetic discoveries box), (2) quantification of the genomic distribution of height-associated loci (Genomic distribution box), (3) assessement of the performances of polygenic predictors of height (Polygenic prediction box), and (4) assessement of the relationship between GWAS sample size and discoveries (Saturation of discovery from GWAS box).

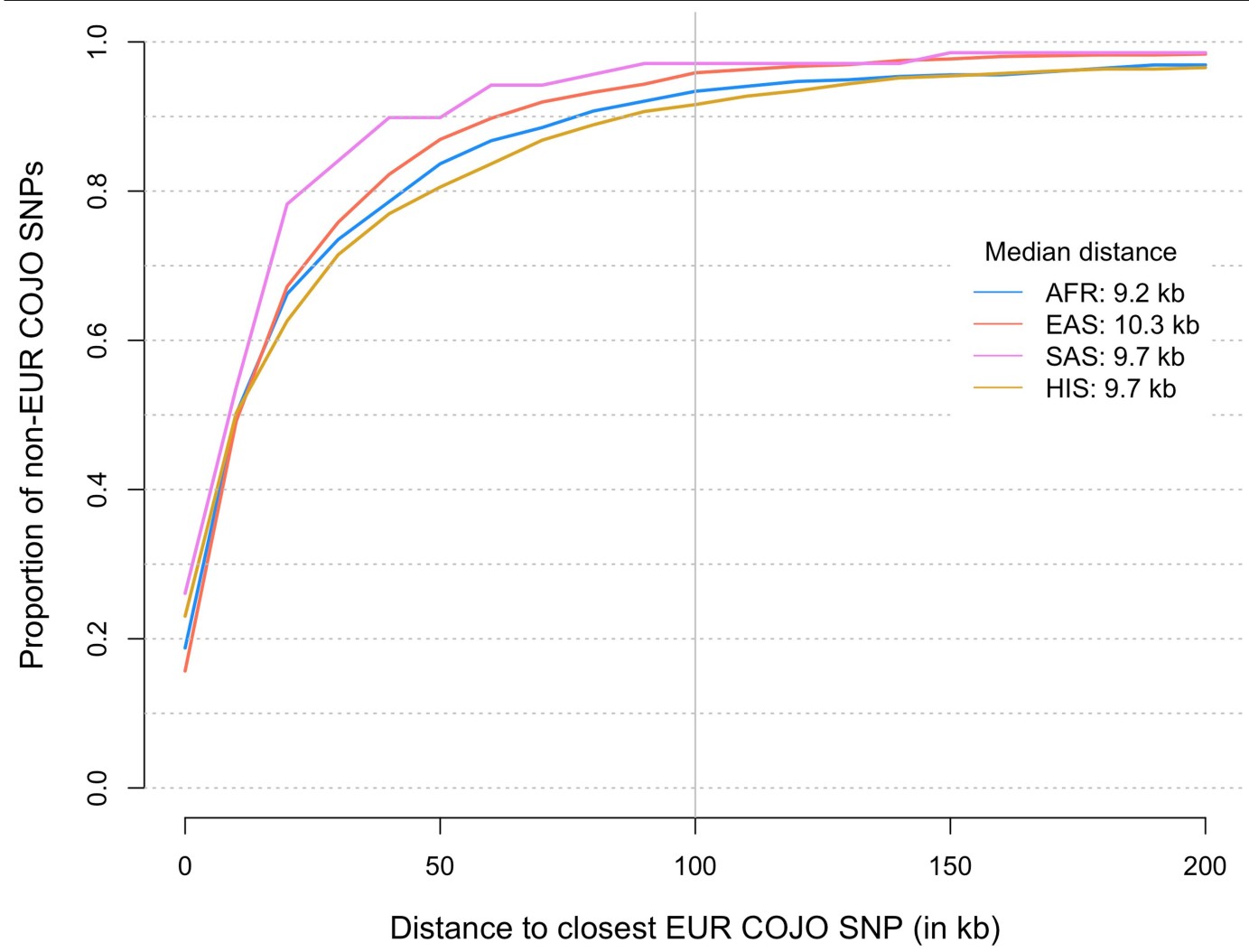

**Extended Data Fig. 2 | Colocalization of height-associated signals across ancestries.** Proportion (y-axis) of GWS SNPs identified in our GWAS meta-analyses of non-European (non-EUR: African – AFR; East Asian – EAS; South Asian – SAS; Hispanic – HIS) ancestry/ethnicity participants that are located within a certain distance (x-axis) of GWS SNPs identified in our GWAS meta-analysis of EUR participants only.

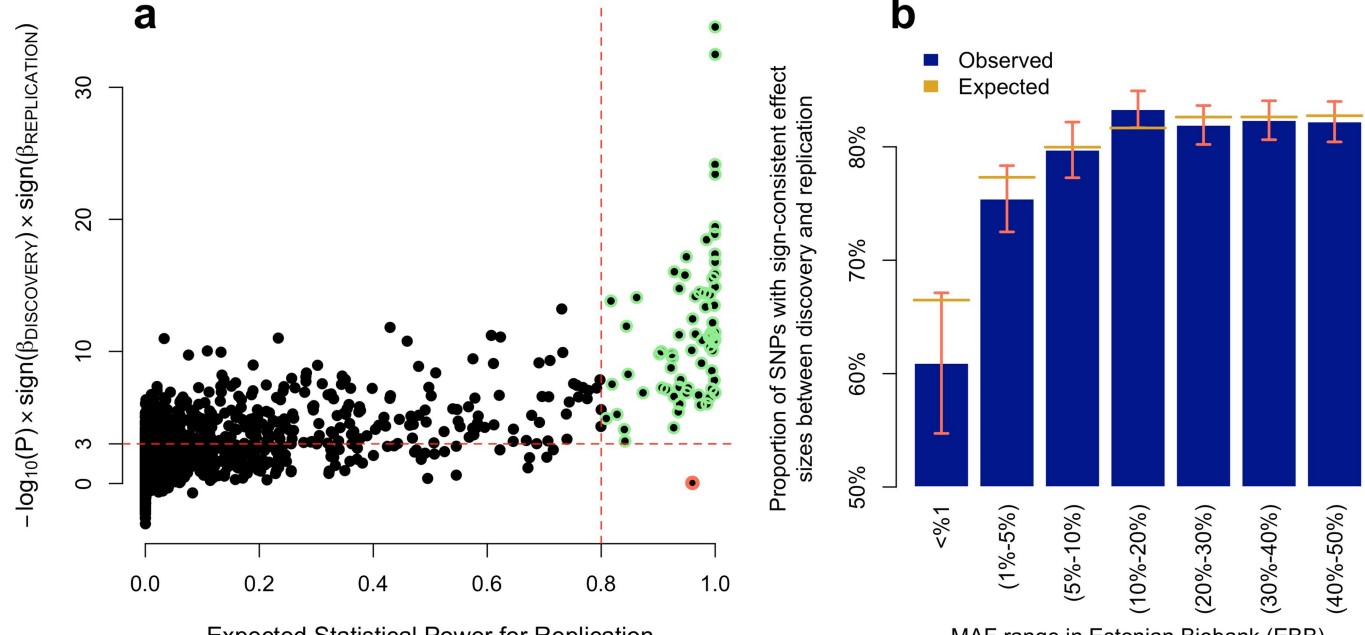

**Extended Data Fig. 3 | Replication of marginal associations in the EBB.**
**a**, Each dot represents one the 12,111 SNPs detected in our trans-ancestry meta-analysis. The x-axis represents the expected statistical power to replicate each association ($P<0.05/9,473 = 5.3\times10^{-6}$; where 9,473 is the number of associations reaching marginal genome-wide significance in our discovery trans-ancestry GWAS and with a minor allele frequency >1% in the EBB sample). The y-axis represents the $-\log_{10}$ of the association p-value in the EBB multiplied by the product of signs of estimated SNP effects in the discovery and in the EBB.

Horizontal dotted line represents replication at $P<0.001$ and the vertical dotted line indicates 80% of statistical power. SNPs highlighted in green have an expected statistical power for replication >80%. One outlier (rs11100870), highlighted in red, does not replicate in the EBB sample. **b**, Proportion ($P$) of SNPs with a sign-consistent estimated effect between discovery GWAS (N-5.3M) and EBB. Expected proportions ($E[P]$) are calculated using equation (2) in the Methods. Error bars are defined as $1.96\times\sqrt{P(1-P)/m}$, where $m$ is the number of SNPs in the corresponding MAF interval.

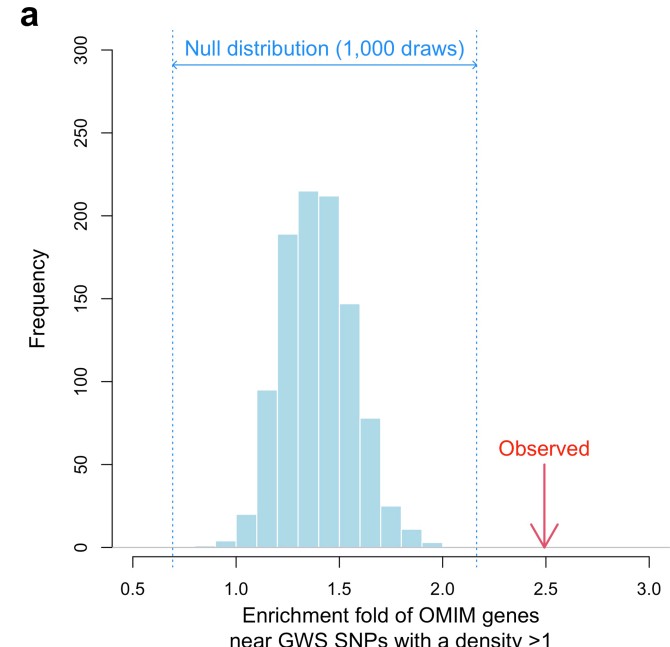

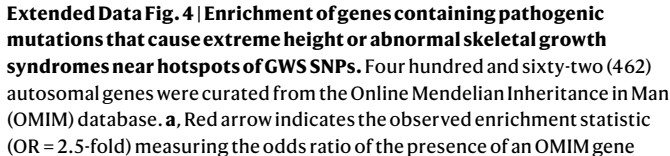

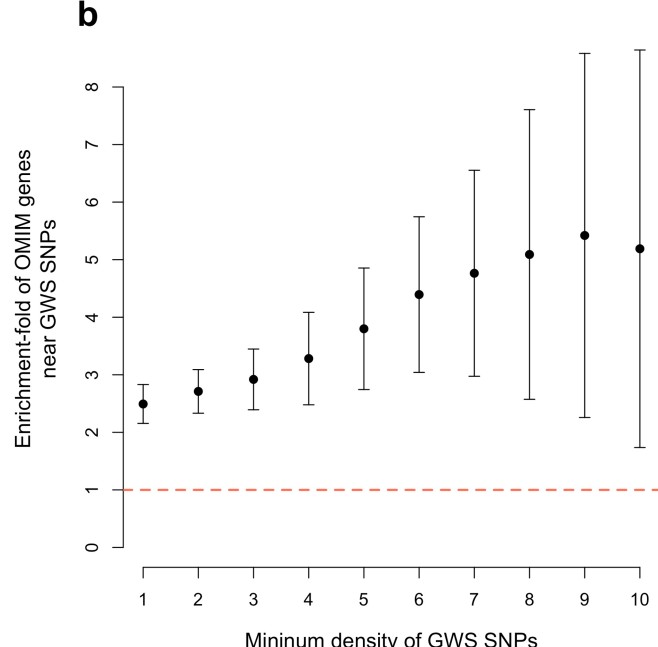

**Extended Data Fig. 4 | Enrichment of genes containing pathogenic mutations that cause extreme height or abnormal skeletal growth syndromes near hotspots of GWS SNPs.** Four hundred and sixty-two (462) autosomal genes were curated from the Online Mendelian Inheritance in Man (OMIM) database. **a**, Red arrow indicates the observed enrichment statistic (OR = 2.5-fold) measuring the odds ratio of the presence of an OMIM gene within 100 kb of a GWS SNPs with a density >1. The blue histogram represents the distribution of enrichment statistics from 1,000 random genes matched, which length distribution matches that of the OMIM genes. **b**, Enrichment of OMIM genes near high density GWS SNPs. High density is defined by on the x-axis by the minimum number of other independent GWS SNPs detected within 100 kb.

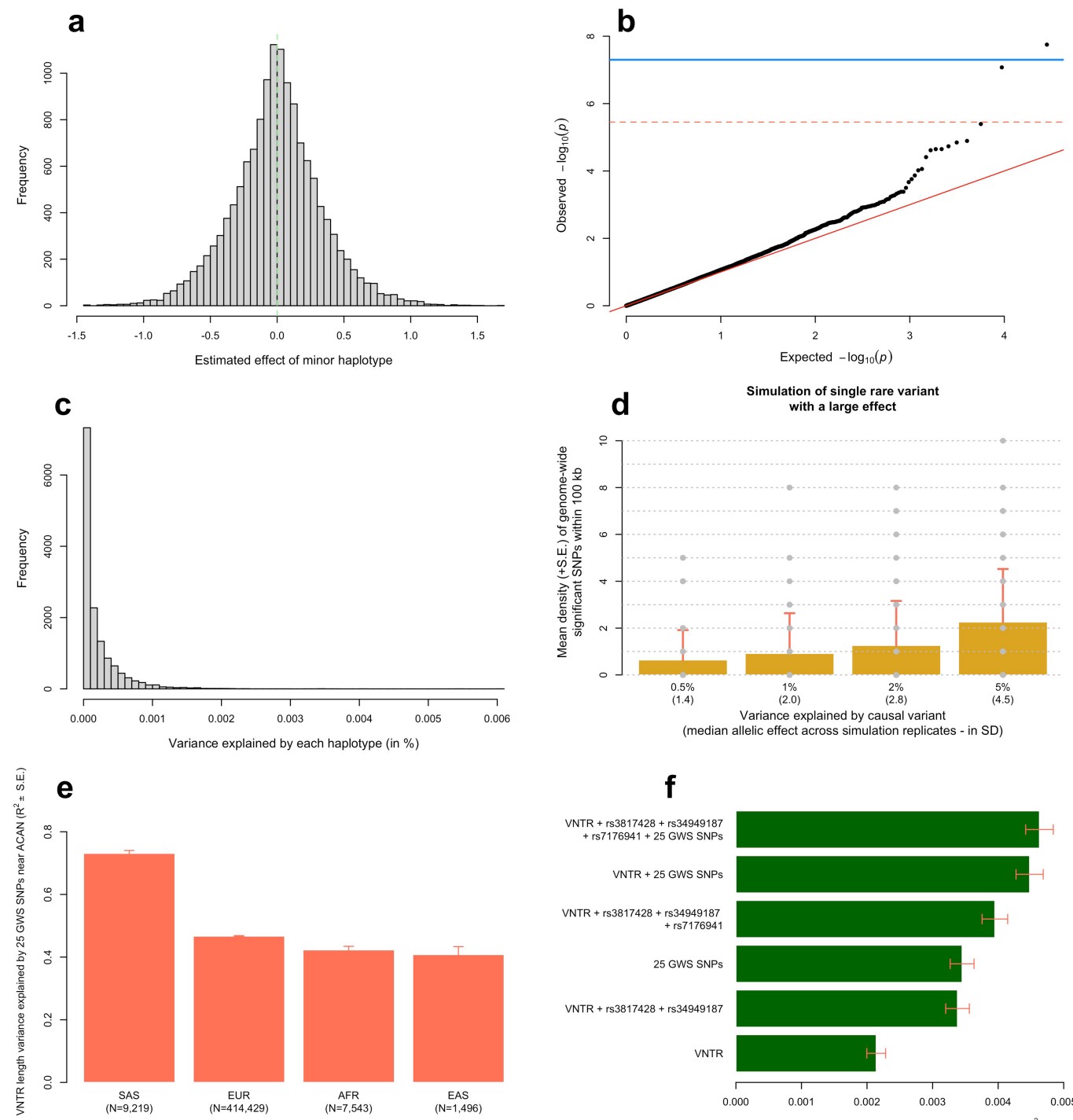

**Extended Data Fig. 5 | Haplotypic analysis at the *ACAN* locus. a**, Distribution of estimated haplotype effects from 14,117 haplotypes covering a 100 kb long genomic region near the *ACAN* gene (hg19 genomic coordinates: chr15:89,307,521-89,407,521). **b**, Quantile-quantile plot of associations between these 14,117 haplotypes and height. **c**, Distribution of the variance explained by each of the 14,117 haplotypes. **d**, Mean signals density (y-axis) across simulated data where 1 causal SNP within the locus explains between 0.5% and 5% (x-axis) of trait variance. Causal variants were sampled from a pool of 13 SNPs with a $1.4 \times 10^{-5} <$ MAF $< 1\%$ genotyped in 291,683 unrelated EUR participants of the UKB, with no missing values at these 13 SNPs. Standard errors were calculated as the standard deviation (s.d.) of signal density across

100 simulation replicates. GCTA-COJO analyses to identify independent signals were performed using a subset of 10,000 unrelated EUR participants of the UKB to mimic the large discrepancy between the size of the discovery GWAS and that of the LD reference used in our real data analyses. **e**, Proportion of VNTR length explained by 25 GWS SNPs identified near *ACAN* in 4 ancestries (European: EUR; South Asian: SAS; East Asian: EAS; African: AFR). **f**, Proportion of height variance explained in a sample of EUR UK Biobank participants by various sets of polymorphisms at the *ACAN* locus. rs3817428 and rs34949187 are two missense variants and rs7176941 is an intronic variant with high posterior causal probability identified in ref. [28]. In **e** and **f**, error bars represent standard error (s.e.).

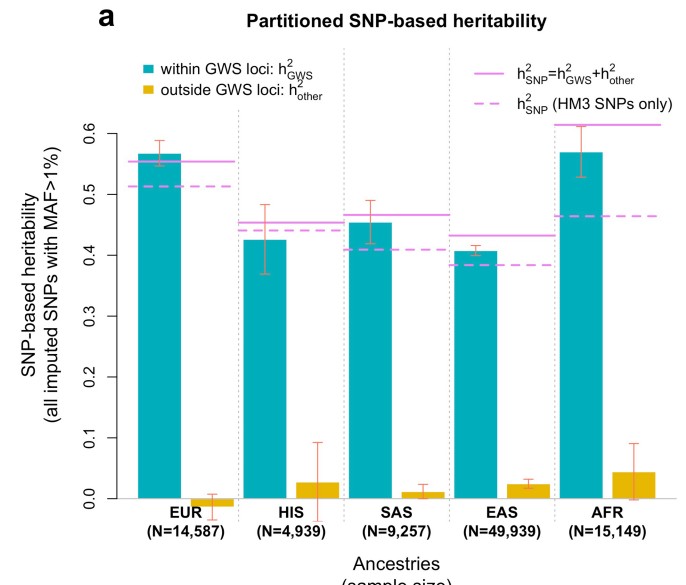

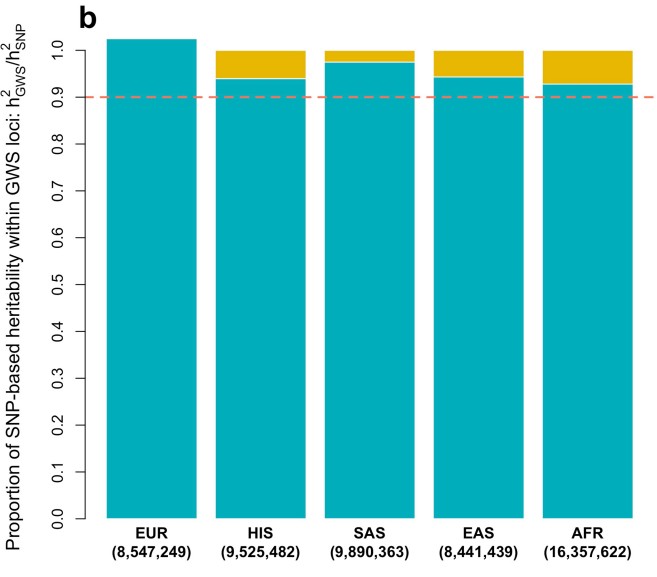

**Extended Data Fig. 6 | Variance of height explained by common SNPs within 35 kb of GWS SNPs.** Stratified SNP-based heritability ($h_{SNP}^2$) estimates were obtained from a partition of the genome into two sets of 1000 Genomes imputed SNPs with a minor allele frequency (MAF) >1%: (1) SNPs within +/− 35 kb of GWS (GWS loci) vs. all other SNPs. Analyses were performed in samples of five different ancestry groups: European (EUR; UK Biobank only), African (AFR), East Asian (EAS) and South Asian (SAS) as described in the legend of Fig. 3. Estimates from stratified analyses were compared with SNP-based heritability estimates obtained from analysing HM3 SNPs only (dotted horizontal violet bar).

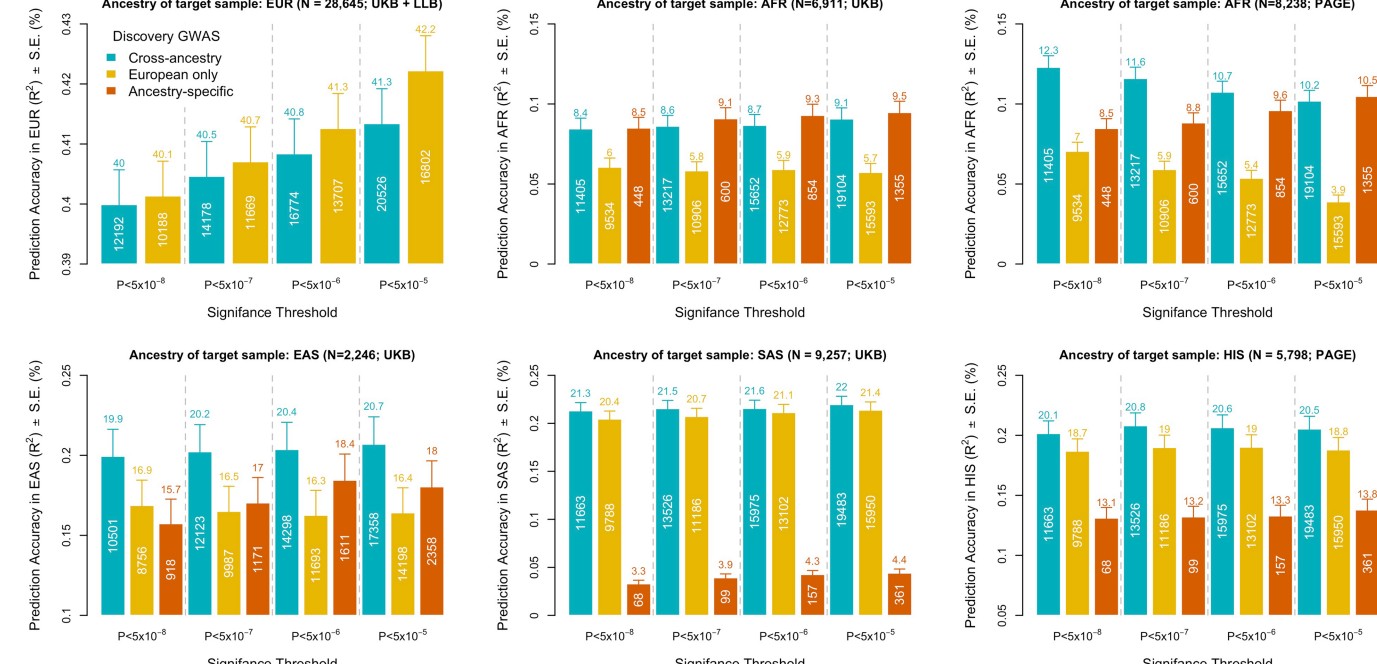

**Extended Data Fig. 7 | Accuracy of PGSs derived from joint effects of SNPs ascertained at various significance thresholds.** The six panels show on their y-axes the prediction accuracy ($R^2$) of multiple PGS across five target samples. The ancestry group and size of each target sample is indicated in the panel title. The top-left panel shows the averaged prediction accuracy in two European ancestry (EUR) target samples from the UK Biobank (UKB) and Lifelines Biobank (LLB). The other panels show prediction accuracies in individual target samples of African ancestry (AFR) from UKB and the PAGE study, East Asian ancestry (EAS) and South Asian ancestry (SAS) ancestry from the UKB and Hispanic ethnicity from the PAGE study. Each panel is divided in four columns representing the four significance levels used to ascertain SNPs using the GCTA-COJO algorithm. GCTA-COJO was applied to each ancestry-group specific GWAS meta-analysis with an ancestry-match linkage disequilibrium (LD) reference. We used genotypes from 50,000 (vs 350,000 for results reported in the main text) unrelated EUR participants as LD reference to run GCTA-COJO on the EUR- and the cross-ancestry GWAS meta-analysis. For the other ancestry groups, we used genotypes from 10,636 AFR individuals, 5,875 EAS individuals, 4,883 HIS individuals and 9,448 SAS individuals as LD reference (as described in Methods). Error bars are standard error (s.e.). The number of SNPs used in each PGS is indicated (in white) within each bar.

| cluster: | 1 | 2 | 3 | 4 | 5 | 6 | 7 | 8 | 9 | 10 | 11 | 12 | 13 | 14 | 15 | 16 | 17 | 18 | 19 | 20 |
|---|---|---|---|---|---|---|---|---|---|---|---|---|---|---|---|---|---|---|---|---|
| gene sets in cluster: | 498 | 391 | 187 | 348 | 669 | 489 | 384 | 1385 | 869 | 199 | 538 | 1976 | 274 | 1257 | 666 | 364 | 809 | 486 | 788 | 1885 |
| **DEPICT** N=0.13M | 0.62 | 2.60 * | 0.26 | 0.43 | 3.81 * | 0.38 | 0.02 | 0.32 | 0.11 | 1.40 | 5.40 * | 0.08 | 0.18 | 0.13 | 0.0 | 3.18 * | 3.11 * | 0.16 | 1.35 * | 0.64 |
| N=0.24M | 0.18 | 1.56 * | 0.21 | 0.25 | 3.48 * | 0.28 | 0.02 | 0.23 | 0.17 | 1.30 | 5.85 * | 0.04 | 0.07 | 0.10 | 0.0 | 3.70 * | 4.01 * | 0.20 | 1.21 | 0.73 |
| N=0.70M | 0.60 | 4.78 * | 0.37 | 0.25 | 4.23 * | 0.36 | 0.02 | 0.30 | 0.09 | 1.05 | 5.13 * | 0.02 | 0.07 | 0.07 | 0.0 | 2.66 * | 3.28 * | 0.06 | 0.85 | 0.61 |
| N=1.63M | 0.56 | 4.42 * | 0.32 | 0.17 | 4.12 * | 0.65 | 0.02 | 0.27 | 0.09 | 1.35 | 5.35 * | 0.02 | 0.03 | 0.07 | 0.0 | 2.58 * | 3.41 * | 0.16 | 0.82 | 0.55 |
| N=2.50M | 0.54 | 3.09 * | 0.48 | 0.11 | 3.96 * | 0.53 | 0.02 | 0.31 | 0.10 | 1.30 | 5.79 * | 0.02 | 0.21 | 0.08 | 0.0 | 3.35 * | 3.18 * | 0.34 | 0.97 | 0.56 |
| N=4.08M | 0.72 | 3.35 * | 0.42 | 0.31 | 4.30 * | 0.40 | 0.02 | 0.32 | 0.11 | 1.45 | 5.59 * | 0.03 | 0.25 | 0.10 | 0.0 | 2.69 * | 3.17 * | 0.24 | 1.01 | 0.49 |
| N=5.31M | 0.50 | 2.99 * | 0.69 | 0.40 | 3.85 * | 0.44 | 0.02 | 0.33 | 0.13 | 1.55 * | 5.79 * | 0.03 | 0.25 | 0.11 | 0.0 | 3.10 * | 3.16 * | 0.30 | 0.98 | 0.54 |
| **MAGMA** N=0.13M | 0.06 | 0.10 | 0.0 | 0.0 | 1.71 * | 0.08 | 0.0 | 0.15 | 0.10 | 0.65 | 4.81 * | 0.08 | 0.36 | 0.19 | 0.03 | 5.85 * | 5.30 * | 0.69 | 1.09 | 1.07 |
| N=0.24M | 0.06 | 0.15 | 0.05 | 0.0 | 1.47 * | 0.0 | 0.0 | 0.14 | 0.09 | 1.20 | 5.72 * | 0.03 | 0.25 | 0.04 | 0.0 | 5.71 * | 5.30 * | 0.72 | 1.14 | 1.03 |
| N=0.70M | 0.0 | 0.15 | 0.05 | 0.0 | 1.49 * | 0.10 | 0.0 | 0.10 | 0.11 | 0.90 | 5.42 * | 0.05 | 0.40 | 0.04 | 0.0 | 5.90 * | 5.04 * | 0.84 | 1.42 * | 1.04 |
| N=1.63M | 0.06 | 0.23 | 0.10 | 0.0 | 1.79 * | 0.10 | 0.0 | 0.07 | 0.13 | 1.05 | 5.40 * | 0.04 | 0.36 | 0.05 | 0.01 | 5.82 * | 5.10 * | 0.55 | 1.19 | 1.05 |
| N=2.50M | 0.06 | 0.17 | 0.05 | 0.0 | 1.65 * | 0.12 | 0.0 | 0.10 | 0.10 | 0.95 | 5.44 * | 0.04 | 0.40 | 0.07 | 0.01 | 5.98 * | 4.98 * | 0.72 | 1.33 * | 1.02 |
| N=4.08M | 0.06 | 0.17 | 0.10 | 0.0 | 1.50 * | 0.10 | 0.0 | 0.06 | 0.11 | 0.95 | 5.40 * | 0.03 | 0.40 | 0.06 | 0.0 | 5.98 * | 5.08 * | 0.88 | 1.25 | 1.07 |
| N=5.31M | 0.06 | 0.17 | 0.10 | 0.0 | 1.56 * | 0.10 | 0.0 | 0.07 | 0.13 | 0.95 | 5.52 * | 0.04 | 0.40 | 0.03 | 0.0 | 5.76 * | 5.01 * | 0.94 | 1.24 | 1.07 |
| OMIM genes | 1.24 | 1.52 * | 0.85 | 1.18 | 1.66 * | 2.56 * | 0.65 | 0.95 | 1.10 | 1.11 | 1.57 * | 0.66 | 0.50 | 0.45 | 0.40 | 2.27 * | 1.83 * | 0.63 | 0.98 | 0.92 |

**Extended Data Fig. 8 | Enrichment of height-associated genes identified at various GWAS sample sizes within 20 clusters of gene sets representing broad categories of biological pathways.** Gene-set enrichment was performed with MAGMA and DEPICT across seven GWAS with increasing sample sizes. Samples used (Lango Allen et al. (2010), $n = 0.13M$; Wood et al. (2014), $n = 0.24M$; Yengo et al. (2018), $n = 0.7M$; GIANT-EUR (no 23andMe), $n = 1.63M$; 23andMe-EUR, $n = 2.5M$; European-ancestry meta-analysis, $n = 4.08M$; and cross-ancestry meta-analysis, $n = 5.31M$) are described in Tables 1–2. The degree of enrichment of gene sets (MAGMA, DEPICT) of known skeletal growth disorder genes catalogued in the Online Mendelian Inheritance in Man (OMIM) database among 20 clusters of gene sets (see Methods section in Supplementary Note 5) is indicated by the blue-red colour scale. Enrichment for MAGMA and DEPICT was defined to be the number of prioritized gene sets (top 10% of gene sets) in each cluster divided by the 10% of the number of gene sets in the cluster. Enrichment for OMIM was defined to be the number of OMIM genes in a gene set (Z > 1.96) divided by the size of the gene set divided by the proportion of all genes in OMIM, then averaged across the cluster. Significant enrichment (compared to shuffled prioritization of gene sets or genes) is marked with *.

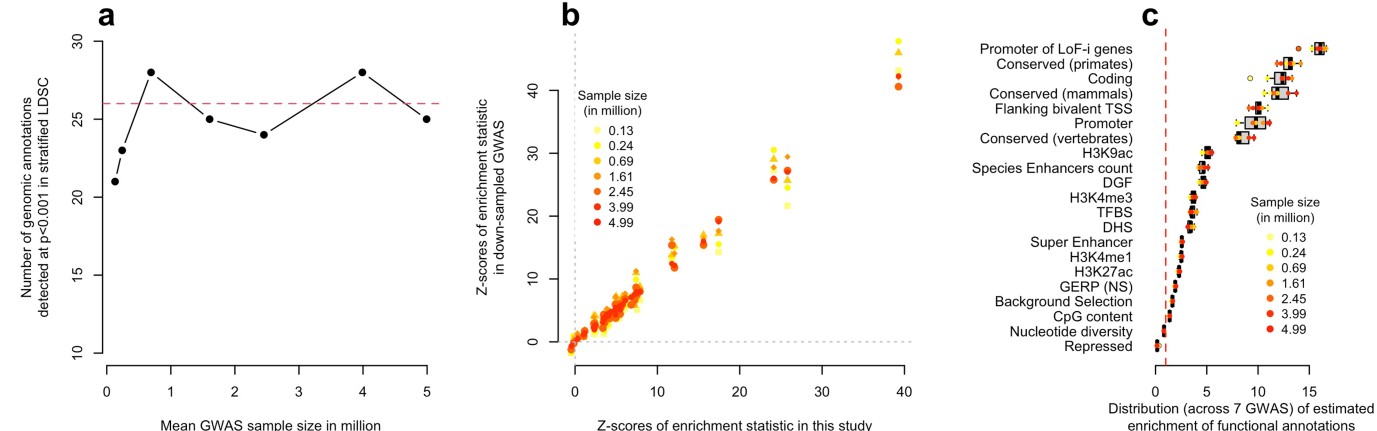

**Extended Data Fig. 9 | Annotation-level saturation of GWAS discoveries as a function of sample size.** Increase in sample size from ~4 million to ~5 million is achieved by including ~1 million participants of non-European ancestry. **a**, Number of annotations showing a significant heritability enrichment as function the function of the sample size of the GWAS used to estimate these enrichment. Heritability enrichment was detected using a stratified LD score regression (LDSC) analysis of 97 genomic annotations included in the "baseline+LD" model from Gazal et al. **b**, Correlation between Z-scores measuring the statistical significance of heritability enrichments of 97 annotations (each dot is an annotation) in our largest GWAS (x-axis) as compared to down-sampled GWAS (y-axis). Sample size is denoted by the colour-code. **c**, Distribution of estimated enrichment statistics for 21 annotations found significantly enriched ($P < 0.05/97$) in at least 6 of the 7 GWAS analysed here. LoF-i genes: Loss of function intolerant genes; TSS: Transcription Start Sites; DGF: Digital genomic footprint; TFBS: Transcription Factor Binding Sites; DHS: DNAse I hypersensitive sites; GERP (NS): GERP++ score (number of substitutions).

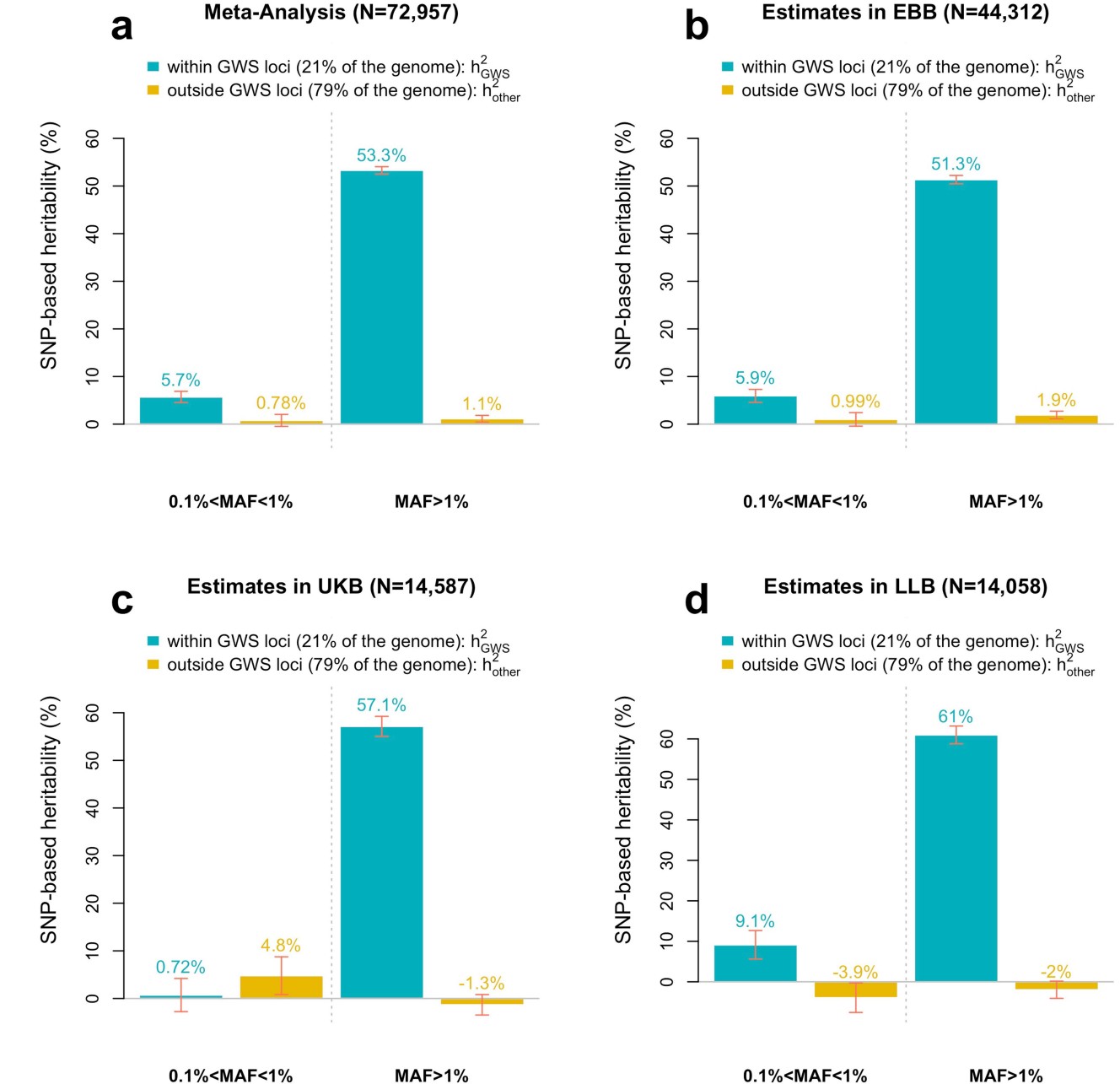

**Extended Data Fig. 10 | Partitioning of low-frequency SNP-based heritability within GWS loci.** Panels **b**–**d** represent partitioned SNP-based heritability estimates from three samples (EBB: Estonian Biobank; UKB: UK Biobank; LLB: Lifelines Biobank) of unrelated European ancestry individuals independent of our discovery GWAS. **a**, Partitioned SNP-based heritability estimates obtained from an inverse-variance weighted meta-analysis of estimates shown in **b**–**d**. SNPs were partitioned into four classes according to their minor allele frequency (MAF: 0.1% < MAF < 1% vs. MAF > 1%) and their position within versus outside GWS loci. The SNP-based heritability contributed by SNPs within GWS loci is denoted $h^2_{GWS}$, and that contributed by SNPs outside these loci is denoted $h^2_{other}$. These results are further discussed in Supplementary Note 6.

# Reporting Summary

## Statistics

For all statistical analyses, confirm that the following items are present in the figure legend, table legend, main text, or Methods section.

| n/a | Confirmed | |
|---|---|---|
| ☐ | ☒ | The exact sample size (*n*) for each experimental group/condition, given as a discrete number and unit of measurement |
| ☒ | ☐ | A statement on whether measurements were taken from distinct samples or whether the same sample was measured repeatedly |
| ☐ | ☒ | The statistical test(s) used AND whether they are one- or two-sided *Only common tests should be described solely by name; describe more complex techniques in the Methods section.* |
| ☐ | ☒ | A description of all covariates tested |
| ☐ | ☒ | A description of any assumptions or corrections, such as tests of normality and adjustment for multiple comparisons |
| ☐ | ☒ | A full description of the statistical parameters including central tendency (e.g. means) or other basic estimates (e.g. regression coefficient) AND variation (e.g. standard deviation) or associated estimates of uncertainty (e.g. confidence intervals) |
| ☐ | ☒ | For null hypothesis testing, the test statistic (e.g. *F*, *t*, *r*) with confidence intervals, effect sizes, degrees of freedom and *P* value noted *Give P values as exact values whenever suitable.* |
| ☐ | ☒ | For Bayesian analysis, information on the choice of priors and Markov chain Monte Carlo settings |
| ☒ | ☐ | For hierarchical and complex designs, identification of the appropriate level for tests and full reporting of outcomes |
| ☒ | ☐ | Estimates of effect sizes (e.g. Cohen's *d*, Pearson's *r*), indicating how they were calculated |

*Our web collection on statistics for biologists contains articles on many of the points above.*

## Software and code

Policy information about availability of computer code

| Data collection | Genotyping arrays and calling algorithms are listed for all cohorts in ST2. |
|---|---|
| Data analysis | Statistical analyses were performed with R version v4.21 (for simulations, visualization, regression and prediction analyses), PLINK v1.9 for standard quality control and GWAS analyses, GCTA v1.93 for estimating SNP-based heritability and for conditional analyses (COJO algorithm), GCTB v2.0 for running the SBayesC model used in prediction analyses, BOLT-LMM v2 and rvtest for mixed-model based GWAS analyses (versions listed in ST2), KING v2.2.5 for identifying relatives in certain cohorts, RAREMETAL v4.15.1. for running within-ancestry GWAS meta-analyses, LDSC v1.0.0 for LD score regression analyses, ImpG-Summary v1.0.1 for imputing GWAS summary statistics, SMR v1.03 for eQTL-based Mendelian Randomization analyses. Additional scripts used for analyses are listed in the Code Availability Statement. |

For manuscripts utilizing custom algorithms or software that are central to the research but not yet described in published literature, software must be made available to editors and reviewers. We strongly encourage code deposition in a community repository (e.g. GitHub). See the Nature Portfolio guidelines for submitting code & software for further information.

## Data

Policy information about availability of data

All manuscripts must include a data availability statement. This statement should provide the following information, where applicable:
- Accession codes, unique identifiers, or web links for publicly available datasets
- A description of any restrictions on data availability
- For clinical datasets or third party data, please ensure that the statement adheres to our policy

Summary statistics for ancestry-specific and multi-ancestry GWAS (excluding data from 23andMe) as well as SNP weights for polygenic scores derived in this study are made publicly available at https://portals.broadinstitute.org/collaboration/giant/index.php/GIANT_consortium_data_files. GWAS summary statistics derived

# Field-specific reporting

Please select the one below that is the best fit for your research. If you are not sure, read the appropriate sections before making your selection.

☒ Life sciences  ☐ Behavioural & social sciences  ☐ Ecological, evolutionary & environmental sciences

For a reference copy of the document with all sections, see nature.com/documents/nr-reporting-summary-flat.pdf

# Life sciences study design

All studies must disclose on these points even when the disclosure is negative.

| | |
|---|---|
| Sample size | We meta-analyzed GWAS data from 281 cohorts, with the aim to reach the largest possible sample size (here, N=5.4 million). Our sample size is larger than any previous GWAS of height, thereby expected to deliver additional associations. |
| Data exclusions | Samples were excluded based on heterozygosity, low call rate, relatedness relative to other participants of the same cohort. Details for each cohort are given in ST2. We analyzed SNPs with a minor allele frequency >1% in at least one of five ancestry groups. Other quality control exclusions are listed in ST2. |
| Replication | We replicated associations detected in our study in N=49,160 individuals in the Estonian Biobank. Quantification of variance explained by height-associated SNPs and prediction accuracy was performed in 61,095 individuals independent of our discovery dataset from four cohorts (hold-out sample from the UK Biobank, the Lifelines Study, the Chinese Kadoorie Biobank and the PAGE Study). |
| Randomization | N/A. Our study is observational and used data from all available participants. No intervention was implemented in any of the study participants, therefore randomization was not required. |
| Blinding | N/A. No intervention was implemented in any of the study participants, therefore blinding was not required. |

# Reporting for specific materials, systems and methods

We require information from authors about some types of materials, experimental systems and methods used in many studies. Here, indicate whether each material, system or method listed is relevant to your study. If you are not sure if a list item applies to your research, read the appropriate section before selecting a response.

### Materials & experimental systems

| n/a | Involved in the study |
|---|---|
| ☒ | ☐ Antibodies |
| ☒ | ☐ Eukaryotic cell lines |
| ☒ | ☐ Palaeontology and archaeology |
| ☒ | ☐ Animals and other organisms |
| ☐ | ☒ Human research participants |
| ☒ | ☐ Clinical data |
| ☒ | ☐ Dual use research of concern |

### Methods

| n/a | Involved in the study |
|---|---|
| ☒ | ☐ ChIP-seq |
| ☒ | ☐ Flow cytometry |
| ☒ | ☐ MRI-based neuroimaging |

## Human research participants

Policy information about studies involving human research participants

| | |
|---|---|
| Population characteristics | The mean/standard deviation/ minimum and maximum values for height and age is given separately for males and females of each cohort in ST3. |
| Recruitment | Study participants were recruited under various designs including population based, prospective studies, birth cohorts, family-based, hospital-based, (nested) case-controls, case-cohort, clinical trials. Designs for each cohort is listed in ST1. |
| Ethics oversight | Written informed consent was obtained from every participant in each study, and the study was approved by relevant ethics committees for each cohort. We provide a list of Institutional Review Boards in ST1. |

Note that full information on the approval of the study protocol must also be provided in the manuscript.

