## [Peer Review File · Nature]

Manuscript Title: A Saturated Map of Common Genetic Variants Associated with Human Height

Reviewer Comments & Author Rebuttals

Reviewer Reports on the Initial Version:

Referee expertise:

Referee #1: statistical genetics

Referee #2: quantitative genetics

Referee #3: human genetics

Referees' comments:

Referee #1 (Remarks to the Author):

The authors performed the largest GWAS to date in 5.38M individuals for human height. They report that over 18% of the genome is covered by a loci associated with height in EUR samples but <5% for non-EUR samples. This is probably due to sample size of the non-EUR ancestry in the analysis since the authors determined that over 87% of the associations detected in non-EUR near in a EUR identified loci. Additionally, overall, the common genetic variation found captures nearly all of the expected genetic variability in height. The authors insights into where increasing GWAS sample sizes and ancestry diversity helps the discovery of variants, genes, and biological pathways is informative for other complex traits. By looking at the correlation between effect sizes in the various ancestry groups, they determined that the height associations are substantially shared across the ancestry groups. And, using a PGS across all samples outperformed using ancestry specific PGS. These last two points are not novel for complex traits but support other publications that have shown similar results.

The manuscript is well written and an interesting read. The abstract appropriate highlights the results, and the discussion is appropriate. Appropriate limitations are noted including that HapMap 3 project variation is being used. However, the authors argue that HapMap 3 captures the regions of interest for height suggesting that rarer variation outside of the identified loci would not contribute to variation in height. This has not been shown in this manuscript. Is there no rarer variation associated with height outside the 20% of the genome identified in this study? Particularly, non-EUR populations may have lower frequency variation that is contributing to height that would increase the percentage of the genome covered.

The authors indicate no evidence of confounding by population stratification in their EUR associations in Suppl. Fig. 3, but it does look like there is a positive slope. Is there confidence bands

for the slopes? The slope for GIANT-EUR is not as problematic as for Wood et al, but it is hard to say it is null. And, what about the other population groups? Despite there being lower sample sizes in these groups, there could still be uncorrected population stratification in these groups and across the main META_FE.

Minor comments

1. Main figures, in general – barplots are used for point statistics. Scatter points are more appropriate for displaying point statistics.
2. Suppl Tables 1 – 3; Why is there is missing statistics? These should be reported.
3. Table 1 – Why was +/- 35 kb used for the loci definition and then to determine the signal density, 100kb was used. Why define the loci differently?
4. Page 3, line 96 – include threshold used for marginal significance in EUR. Does this mean significant in the marginal results or marginally significant?
5. Page 4, line 179 states that the mean signal density is 2. What is the median? Is this statistic normal distributed?
6. The custom R script that was used for the fixed-effect inverse variance weighted meta-analysis should be provided as a supplemental note.

Referee #2 (Remarks to the Author):

- A. The paper describes a large genome-wide association study for human height. It finds circa 12k independent SNPs associated with height. These SNPs colocalise in ~7k non-overlapping genomic regions, with an median size of 90kb. That is, about 21% of the genome explains about 40% of the variation in human height.
- B. This is an impressive study of unprecedented size that uses human height as an exemplar of a highly polygenic trait for which all genetic variation captured by the $h^2(\text{SNP})$ can be explained by the identified loci.
- C. Data and methodology follow well defined protocols developed in GWAS for the last decade and a half. Improvements on the presentation of the data could be made and a list is presented below.
- D. The statistics are appropriate.
- E. Some of the conclusions are perhaps a bit far-fetched.
- F. I have no major concerns. Suggested improvements for improvement are:

Main Manuscript

L91- SF9, I imagine the Yengo referenced is this paper, but it is not clear is the previous Yengo height paper.

L96, talks about marginal effects, whilst I believe SF4a are COJO results.

L99 – GWSs are not randomly distributed, hence selecting randomly sampled HM3 SNPs doesn't strike to me as the right comparison. Perhaps, the authors could show try to mimic the distribution of GWS SNPs in relation to gene distance, patterns of LD and other functional annotations.

L101- Could the authors show the cross-ancestry correlation of allele substitution effects for non-GWS SNPs stratified by significance levels?

L151. This sentence is a bit confusing, it implies little benefit from the inclusion of other ancestries, but N is much smaller for non-Europeans. What's the point the authors wish to make?

L158. I think the reader would benefit from seeing how many significant SNPs there are in the

beta(meta) vs beta(EBB) comparison.

L187 SF13b, why use 100kb if the median size of linked genomic segments is 90kb?

L209. I don't understand how this sentence can be true without identifying the causal variants.

Maybe tone down.

L213. Is it mean or median, or are the same in this case?

L232. As before I am unconvinced random SNPs is the right background here.

L251. This sentence is misleading, I'd argue that can be purely explained by the difference in N.

L253. The reader would benefit from seeing $h^2(\text{SNP})$ and $R^2(\text{GWS})$ estimated on the same samples. Globally (genome-wide) and locally (7k genomic regions) to ensure one is not comparing pears and apples.

L262. This sentence seems to be pulled out of thin air. Where is the data to support that sentence shown? Although I agree that this is the likely explanation I don't think the authors show that.

L262. I also found there is a bit of a mix and match selection of samples which will make reproducibility quite cumbersome.

L277. Notation, I imagine $R^2(\text{SNP})=R^2(\text{GWS})$. See SF19.

L277. The authors simulate 1,000 variants when they are arguing there are 12k. Please, update the simulations to a more realistic number.

Supplementary information

Please, read carefully. URLs are missing (P42, P43, etc) , some of the legends are unclear and not self-contained. For instance:

ST1, has a cohort with N=5.

ST4 9,863 COJO SNPs, but main text talks about marginal effects. Which one is it?

ST5, LD with next SNP, what SNP, what LD measure? In what population was estimated? With what sample size was estimated?

ST11. What is the length group?

ST12. Variance explained $2p(1-p)bx_bJ$? I imagine the J is a typo.

SF19 has two vertical lines, but only one is described.

ST16. I'd keen to hear how the authors interpret that an eQTL in the skin or salivary gland might mediate or be linked to height. Perhaps, a bit of discussion in the MS would be welcomed.

In my view, if the authors wish to uphold the best standards and lead the way, they should consider releasing the following data to the research community:

1. All results for imputed HM3 SNPs per cohort (rsID, beta, SE(beta), Ref Allele, N, Imputation Accuracy, etc) . Cohorts included in the meta-analyses and replication.
2. All summary statistics for all HM3 SNPs from the GWAS meta-analyses (FE, EUR, EAS, HIS, AFR, SAS).

G. References are largely OK, but they missed some of the references to prediction of height and transferability to other populations in UKB.

H. The manuscript is clear.

Referee #3 (Remarks to the Author):

In this manuscript, Yengo et al. conduct the largest GWAS to date on height, a well-established trait representative of a highly polygenic genetic architecture. The manuscript is well written and the analyses are appropriate and comprehensive. Notably, the authors include individuals from diverse ancestries, and are thoughtful in their discussion of cross-ancestry findings. A prime finding of the manuscript is with respect to saturation of GWAS loci for height, namely they did not find uniform coverage across the entire genome at this massive sample size, but that top loci are clustered in functionally relevant regions that explain effectively all heritable variation (at least in EUR). This is to my knowledge the most direct test of this saturation phenomenon to date, and speaks to the likely trends in genetic architecture for complex traits at large. In this manuscript the authors also demonstrate the utility of combining familial information and PGS in risk prediction, as well as demonstrate that smaller GWAS can implicate the correct functional annotation categories sooner than reaching variant level saturation. Finally, they show the benefit of meta-analyzing results from across ancestry groups for the generation of better performing PGS for diverse populations. Overall I was very impressed with this paper and think it will have a large impact on the GWAS community. I do, though, have a few suggestions that I hope may strengthen it even further:

1. The investigation into residual population stratification in the EUR results is interesting and well warranted given prior findings regarding confounding of height GWAS in Europe, as the authors discuss. Given the large sample sizes of the non-EUR ancestry groups, it would also be well worth a check that population stratification isn't affecting the results in those GWAS as well. Despite the groups being of much relative smaller size than EUR, they are still among the largest GWAS of diverse cohorts yet published. Additionally, many of these non-EUR groups also have high levels of admixture, making PS particularly important to control for in them. I do not think that as comprehensive a test as what was done in EUR is necessary for these other groups, but some further due diligence to confirm that PS isn't driving some signals (especially considering the authors highlight the novel contributions from other ancestries) is warranted, e.g. one of the strategies the authors describe in Supplementary Note 1.

1a. Related, I was surprised to see no PC plots in this manuscript. Some additional supplemental figures with the PCs for the different ancestry groups would be useful for context as to the composition of these groups.

2. The authors justify their use of a EUR LD reference panel for COJO analysis, noting that it appears to 'reasonably approximate' the LD observed in their cross-ancestry meta-analysis. Would findings be improved by including some diverse participants in the generation of an LD reference panel? Though your sample is indeed driven by the large number of EUR individuals, including representatives from other ancestries in the generation of the LD scores may aid results contributed by other ancestry groups.

3. Figure 1, the 'Brisbane plot' is an intriguing way to visualize the GWAS results seen here, and I would bet this presentation will become widely used as GWAS sample sizes continue to grow. I

imagine the authors made an intentional decision not to include standard Manhattan plots in this manuscript (likely overloaded with results) but some way to highlight the loci contributed by each ancestry group would be interesting to see.

4. This reviewer could not find a description of the imputation procedure done on the component cohorts. Relevantly, Supplementary Figure 2 shows strikingly how much more poorly captured less common variants are in AFR. Though this site frequency spectrum is likely driven by the genotype arrays used to generate data, it may be improved with the inclusion of more diverse participants in the imputation reference panel for these cohorts. At least a discussion of imputation procedures done in the various cohorts and the reference panels used should be added.

5. Additionally, though the authors do limit to HM3 SNPs that should be quite well behaved, it may be useful to confirm that the reasonably permissive INFO threshold of 0.3 is appropriate for all cohorts here. This is related to the concerns over population stratification noted above (e.g. if imputation is performing worse in some ancestries than others).

6. The authors comment on their analyses here for height speaking to the broader implications for saturation as it pertains to other complex traits. Height is extremely polygenic even among complex traits – could the authors comment on the likely saturation thresholds for traits with other levels of polygenicity?

7. Only GWS SNPs appear to have been used in the PGS analyses. Some justification for this choice would be helpful, and even better to see the PGS R² at different p value thresholds, as is commonly presented for this type of analysis.

8. Given the unprecedented size/completeness of this GWAS, I would love to see the distribution of effect sizes as they relate to MAF. This would be a useful figure to add that would speak more broadly to what empirical data is bearing out with respect to these two variant dimensions.

Author Rebuttals to Initial Comments:

Referee #1 (Remarks to the Author):

The authors performed the largest GWAS to date in 5.38M individuals for human height. They report that over 18% of the genome is covered by a loci associated with height in EUR samples but <5% for non-EUR samples. This is probably due to sample size of the non-EUR ancestry in the analysis since the authors determined that over 87% of the associations detected in non-EUR near in a EUR identified loci. Additionally, overall, the common genetic variation found captures nearly all of the expected genetic variability in height. The authors insights into where increasing GWAS sample sizes and ancestry diversity helps the discovery of variants, genes, and biological pathways is informative for other complex traits. By looking at the correlation between effect sizes in the various ancestry groups, they determined that the height associations are substantially shared across the ancestry groups. And, using a PGS across all samples outperformed using ancestry specific PGS. These last two points are not novel for complex traits but support other publications that have shown similar results. The manuscript is well written and an interesting read. The abstract appropriate highlights the results, and the discussion is appropriate. Appropriate limitations are noted including that HapMap 3 project variation is being used.

We thank Reviewer #1 for the accurate summary of our work and for acknowledging the quality of our manuscript. We address below each point raised by the referee.

However, the authors argue that HapMap 3 captures the regions of interest for height suggesting that rarer variation outside of the identified loci would not contribute to variation in height. This has not been shown in this manuscript. Is there no rarer variation associated with height outside the 20% of the genome identified in this study? Particularly, non-EUR populations may have lower frequency variation that is contributing to height that would increase the percentage of the genome covered.

We thank the referee for this comment and question. Our study identified genomic regions that contain the relevant common variants (MAF>1%) contributing to height heritability in European ancestry (EUR) populations. However, we also showed that ~5% to 10% of the SNP-based heritability of height in non-EUR populations could be explained by variants outside these genomic regions (Fig. 3 and Fig. S25). Therefore, it is possible that genetic variants (common and rare) outside the ~7000 loci identified would contribute to the heritability of height, particularly in those populations.

Among the future directions mentioned in our Discussion section, we listed testing “whether rare genetic variants associated with height are also concentrated within the same loci”. We previously did not provide any prior evidence supporting that hypothesis. However, building upon the referee’s comment we performed two additional analyses. First, we stratified SNPs (not only HapMap3 SNPs)

that are common in African ancestries (AFR) participants into two groups: those with a MAF>1% in EUR vs. those with a MAF<1% in EUR. In this analysis, we chose AFR as a reference for all non-EUR populations because it has the largest number of common variants (i.e., ~16 M SNPs vs ~8-9M in other non-EUR groups; Fig S25). Next, we quantified the enrichment of SNP-based heritability within our 7000 genome-wide significant (GWS) loci for both groups of variants.

Suppl. Fig. 35a-b

We found that height-associated SNPs that are rare in EUR but common in AFR are also concentrated within our 7209 genome-wide significant (GWS) loci. Therefore, under the assumption that causal variants are shared across ancestries, this observation suggests that a certain class of rare variants in EUR that are associated with height will likely also be found within these regions. Note that such rarer variants in EUR could, in principle, be detected with larger AFR GWAS.

Secondly, we analysed ~72K unrelated European ancestry individuals (14k samples from the UK Biobank, 14k samples from the Lifelines study and 44k samples from the Estonian Biobank) independent of our discovery GWAS to quantify the enrichment of SNP-based heritability for rarer variants with a MAF between 0.1% and 1%. As large sample size is required to accurately quantify the contribution of rarer variants, we focused this analysis on EUR.

Suppl. Fig. 34a

We found that ~88% (i.e. $5.7/(5.7 + 0.78)$) of the SNP-based heritability of height explained by variants with $0.1\% < \text{MAF} < 1\%$ is also due to SNPs within our 7209 height-associated loci. This analysis also suggests that rarer variants associated with height would likely be detected in (or near) these loci. It is noteworthy that these analyses are based upon imputed SNPs with an imputation accuracy score > 0.3 , which is only a fraction of SNPs within that MAF range (approximately 80%, Wainschtein et al. 2022).

In conclusion, we provide suggestive evidence that both common and rare SNP-based heritability in EUR is strongly enriched within our identified loci. We have added these results in the main text (L410-416) and Supplementary Note 6 (Fig. S34-35), and acknowledge that large samples with whole-genome sequence data will be required to test this hypothesis with more statistical power.

The authors indicate no evidence of confounding by population stratification in their EUR associations in Suppl. Fig. 3, but it does look like there is a positive slope. Is there confidence bands for the slopes? The slope for GIANT-EUR is not as problematic as for Wood et al, but it is hard to say it is null. And, what about the other population groups? Despite there being lower sample sizes in these groups, there could still be uncorrected population stratification in these groups and across the main META_FE.

We thank the referee for these questions. Panel b in (former Suppl. Fig. 3, now Suppl. Fig. 6) shows the estimated slopes for different height GWAS with error bars corresponding to standard errors.

We can conclude from these standard errors that the population stratification (PS) slope of our EUR GWAS (N=4M) is not significantly different from 0 (P=0.1). This P-value was previously reported in Suppl. Note 1 (now Suppl. Note 2).

Following the referee's suggestions (and that of Reviewer #3), we extended our assessment of population by calculating PS slopes for all non-EUR GWAS. We found some evidence of uncorrected PS in our SAS- and HIS-specific GWAS (Fig S7).

Suppl. Fig. 7

Suppl. Fig. 8

However, the levels of residual stratifications were at least 2-fold lower than that of the Wood et al. (2014) study, which triggered the development of this metric. Importantly, PS slopes from our cross-ancestry meta-analyses (Fig S8) suggested that the residual PS in SAS and HIS had only a limited impact on our combined analysis. We have added these new results to Supplementary Note 2 and now echo them in the main text as well (L94 – 96): “Therefore, we specifically investigated confounding effects in all ancestry specific GWAS and found that our results are minimally affected by population stratification (Suppl. Note 2, Suppl. Fig. 6-8).”

Minor comments

1. Main figures, in general – barplots are used for point statistics. Scatter points are more appropriate for displaying point statistics.

We thank the referee for this point. Barplots are classically utilised in the field to represent heritability and prediction accuracy estimates (e.g. Wood et al. 2014; Yengo et al. 2018, etc.). Therefore, we believe that these plots are appropriate for consistency with prior studies.

2. Suppl Tables 1 – 3; Why is there is missing statistics? These should be reported.

In most cases where missing data exists (ST3), some studies provided association data only for sex-combined, male-specific or female-specific analyses. In those instances, we have inserted “NA” into cells that are not related to analyses provided by the respective studies. In other instances, missing or problematic data could not be recovered as it was not provided to GIANT. For several studies where insufficient summary data was provided to GIANT (even after re-contacting the relevant studies to request data for this revision), we have now taken these out of the ST1-3 tables as appropriate and listed the studies in the respective table legends with a comment that data was not available from the study.

3. Table 1 – Why was +/- 35 kb used for the loci definition and then to determine the signal density, 100kb was used. Why define the loci differently?

We apologise if there was any confusion because of our various windows definitions. These two analyses used different windows because they are addressing somewhat different questions. The 35kb window has a particular meaning in our analyses because this is when we start to see saturation (i.e., we explain all the SNP-based heritability) in EUR. Therefore, we found sensible to define GWS loci using this threshold. Note that loci were not defined in the manuscript before presenting partitioned SNP-based heritability analyses. For signal density, we are asking how far apart different associations are from each other (without any implication that the regions in between signals account for any heritability), and somewhat arbitrarily chose 100 kb as a distance at which it is still quite plausible that signals could be acting through the same effector gene, leading them to be nonrandomly clustered. This new per-SNP measure is useful to quantify clustering of associations within in a relatively small window; we also note that correlation of signal density within 100 kb vs density within 35 kb is >0.69 across all sets of GWS SNPs.

4. Page 3, line 96 – include threshold used for marginal significance in EUR. Does this mean significant in the marginal results or marginally significant?

We apologise for the confusion. We meant significance in the marginal (one SNP at the time) results. We have now clarified this in the main text (L101: $(P_{\text{GWAS}} < 5 \times 10^{-8})$).

5. Page 4, line 179 states that the mean signal density is 2. What is the median? Is this statistic normal distributed?

We thank the referee for this question. Signal density is not normally distributed. We now show a histogram of signal density in Suppl. Fig 19. and added the median density in Fig. 2.

Suppl. Fig. 19. Distribution of signal density.

6. The custom R script that was used for the fixed-effect inverse variance weighted meta-analysis should be provided as a supplemental note.

We thank the referee for this request. We now provide the requested R script (based on the R package “meta”) on github (https://github.com/loic-yengo/ScriptsForYengo2022_HeightGWAS).

Referee #2 (Remarks to the Author):

A. The paper describes a large genome-wide association study for human height. It finds circa 12k independent SNPs associated with height. These SNPs colocalise in ~7k non-overlapping genomic regions, with a median size of 90kb. That is, about 21% of the genome explains about 40% of the variation in human height.

B. This is an impressive study of unprecedented size that uses human height as an exemplar of a highly polygenic trait for which all genetic variation captured by the $h^2(\text{SNP})$ can be explained by the identified loci.

C. Data and methodology follow well defined protocols developed in GWAS for the last decade and a half. Improvements on the presentation of the data could be made and a list is presented below.

D. The statistics are appropriate.

E. Some of the conclusions are perhaps a bit far-fetched.

F. I have no major concerns. Suggested improvements for improvement are:

Points A – D. We thank the referee for the accurate summary of our study and for acknowledging the good quality of our contributions.

Point E. We have now clarified or toned down each of the conclusions flagged by the referee (details below).

Point F. We are grateful for all the suggestions made by the referee to further improve the quality of our study.

We address below each point raised by the referee.

Main Manuscript

L91- SF9, I imagine the Yengo referenced is this paper, but it is not clear is the previous Yengo height paper.

We apologise for the confusion. We have now added the year (i.e. Yengo (2018)) to clarify which study we were referring to.

L96, talks about marginal effects, whilst I believe SF4a are COJO results.

We did consider both statistics: (1) the proportion of GWS COJO SNPs identified in non-EUR GWAS, which reached marginal genome-wide significance (i.e. $P < 5 \times 10^{-8}$) in our EUR GWAS and (2) the colocalization of GWS COJO SNPs identified in non-EUR vs. those identified in EUR. Former Figure S4a (now Suppl. Fig. 9a) refers to the latter. More generally, we used COJO to ascertain SNPs jointly reaching genome-wide significance. In a few cases, we focused certain analyses on a subset of those COJO SNPs (i.e. ~80%-90% of them) that were also marginally GWS.

L99 – GWSs are not randomly distributed, hence selecting randomly sampled HM3 SNPs doesn't strike to me as the right comparison. Perhaps, the authors could show try to mimic the distribution of GWS SNPs in relation to gene distance, patterns of LD and other functional annotations.

We thank the referee for this suggestion. We have now repeated this analysis using random SNPs matched on 24 functional annotations from Finucane et al. 2015 (e.g., coding, conserved regions across mammals, DNase I hypersensitive sites, etc.) and rephrased our sentence as (L102-105) “**In contrast, a randomly sampled HM3 SNP (matched with GWS SNPs identified in non-EUR on 24 functional annotations; Suppl. Methods) falls within 100 kb of a EUR GWS SNP 55% of the time on average (standard deviation=1% over 1,000 draws).**” This enrichment seems stronger than previously reported). However, we highlight that our previous result considered location relative to SNPs reaching either marginal or conditional genome-wide significance. We have now simplified this comparison to minimize confusion (as underlined by this referee) and ensure consistency with Suppl. Fig 4a (now Suppl. Fig. 9a), which focused on GWS COJO SNPs. Therefore, the “55%” reported above is to be compared with “49%” obtained for a random HM3 SNP not matched on anything. In other words, matching on functional annotations increases of 55%-49%=6% the probability to colocalize with a GWS COJO SNP detected in our EUR GWAS.

L101- Could the authors show the cross-ancestry correlation of allele substitution effects for non-GWS SNPs stratified by significance levels?

We thank the referee for this suggestion. We re-ran COJO in all ancestry specific GWAS using lower significance thresholds: 5×10^{-7} , 5×10^{-6} and 5×10^{-5} . We then calculated the correlation of marginal SNP effects for all pairs of ancestry groups as previously shown in former Suppl. Fig.6-9 (now Suppl. Fig. 10 - 14). The results are shown below and in the main manuscript as Suppl. Fig. 15.

Suppl. Fig. 15

Overall, we found a monotonic relationship between strength of significance and the cross-ancestry correlation of marginal SNP effects, which remain >0.5 across all comparisons. We have added a reference to Suppl. Fig. 15 in the main text (L111).

L151. This sentence is a bit confusing, it implies little benefit from the inclusion of other ancestries, but N is much smaller for non-Europeans. What's the point the authors wish to make?

We apologise for the confusion. The point of this sentence was to highlight what happened to the 131 non-EUR specific signals after being meta-analysed with a large EUR GWAS. We found that 59/131 of these signals reached marginal significance in the METAFE but only 24 of them overlapped with our list of 12,111 COJO SNPs. These 24 signals contributed by non-EUR GWAS are now highlighted in the Fig. 2 (Former Fig. 1). We have rephrased the sentence as (L162 - 165): "Of the GWS SNPs obtained from the non-EUR meta-analyses above that were conditionally independent of the EUR GWS SNPs, 0/2 in SAS, 5/17 in AFR, 27/49 in EAS, and 27/63 in HIS were marginally significant in our METAFE (Suppl. Table 9) and 24 of those (highlighted in Fig. 2) overlapped with our list of 12,111 quasi-independent GWS SNPs."

L158. I think the reader would benefit from seeing how many significant SNPs there are in the beta(meta) vs beta(EBB) comparison.

We previously reported the number of SNPs used for the comparison with EBB in the main text (L176-179): “Over the 11,529/11,847 SNPs with a MAF>1% in the EBB, we found a correlation of marginal SNP effects of $\rho_b=0.93$ (jackknife standard error; S.E. 0.01) and a correlation of conditional SNP effects using the same LD reference panel of $\rho_b=0.80$ ”.

L187 SF13b, why use 100kb if the median size of linked genomic segments is 90kb?

Genome-wide significant loci were defined using a 35kb window, corresponding to when saturation of the SNP-based heritability was observed in EUR. These 35 kb windows were used around each GWS SNPs and overlapping windows were then merged subsequently. 90 kb corresponds to the mean size after merging overlapping segments. As described in our response to reviewer 1, the different sized windows were used in analyses designed to answer different questions: the 35 kb window to define loci that saturated the capture of heritability, and the somewhat arbitrary 100 kb window to assess non-random clustering of signals that could plausibly be acting through the same effector gene (regardless of whether the region between the signals captures heritability).

L209. I don't understand how this sentence can be true without identifying the causal variants. Maybe tone down.

We have rephrased the sentence as (L223-224) “We chose this size window because it was predicted that causal variants are located within 35 kb of GWS SNPs with probability >80% (ref.²³).”

L213. Is it mean or median, or are the same in this case?

We thank the referee for pointing out this inconsistency. 90kb is the average length (the median is 70 kb). We have corrected the abstract, which wrongly referred to the median length.

L232. As before I am unconvinced random SNPs is the right background here.

In these analyses SNPs were actually matched on MAF and LD. We previously wrote “a random set of SNPs with similar EUR MAF and LD scores as the 12,111 height-associated GWS SNPs.” We have rephrased the sentence to reflect this better (L246-247): “Furthermore, we repeated our analysis using a random set of SNPs matched with the 12,111 height-associated GWS SNPs on EUR MAF and LD scores”.

L251. This sentence is misleading, I'd argue that can be purely explained by the difference in N.

We agree with the referee that sample size is a key determinant of prediction accuracy, which could explain this observation. Therefore, we now emphasize that sample size difference between

ancestry-specific GWAS is a likely explanation for this observation (L267-269): “PGS_{GWS-METAPE} yielded prediction accuracies larger or equal to that of all other PGS_{GWS} (Fig. 4a), partly reflecting sample size differences between ancestry-specific GWAS and also consistent with previous studies.²⁷”. Importantly, we also show (after correcting our COJO results in AFR for collinearity issues) that the PGS based on 9863 SNPs identified in our EUR GWAS (N=4M) no longer outperforms that based on 453 SNPs detected in our AFR GWAS (N=293K), despite a substantial sample size difference (Updated Figure 3 – Now Fig. 4).

Figure 4.

L253. The reader would benefit from seeing $h^2(\text{SNP})$ and $R^2(\text{GWS})$ estimated on the same samples. Globally (genome-wide) and locally (7k genomic regions) to ensure one is not comparing pears and apples.

We did use the same samples for prediction (former Fig. 3, now Fig. 4) and estimation of $h^2(\text{SNP})$ (former Fig. 2, now Fig. 3). Fig. 4 shows multiple estimates of prediction accuracy for the same ancestry group (e.g., UKB and Lifelines for EUR and UKB and PAGE for AFR), while Fig. 3 shows the meta-analysis of $h^2(\text{SNP})$ estimates across these samples. The only exception is that we did not show $R^2(\text{GWS})$ in the Chinese Kadoorie Biobank (CKB). This is because we could not share GWAS summary statistics including data from 23andMe data participants with our CKB collaborators who do not have ethics approval to access the data.

L262. This sentence seems to be pulled out of thin air. Where is the data to support that sentence shown? Although I agree that this is the likely explanation I don't think the authors show that.

We have now removed that sentence.

L262. I also found there is a bit of a mix and match selection of samples which will make reproducibility quite cumbersome.

See answer to comment above referring to L253. We assessed the accuracy of PGS in various samples (sometimes within the same ancestry group) to demonstrate the robustness of our results and their replicability across samples.

L277. Notation, I imagine $R^2(\text{SNP})=R^2(\text{GWS})$. See SF19.

The derivations presented in Supplementary Note 4 are generalisable to any PGS, not just those based on GWS SNPs. This is the reason why we used a different (more general) notation. We now highlight the change of notations in the figure legend.

L277. The authors simulate 1,000 variants when they are arguing there are 12k. Please, update the simulations to a more realistic number.

We thank the referee for this comment. However, the point of this simulation was merely to validate our mathematical derivations. Choosing different simulation parameters (e.g., using 12K SNPs as suggested by this referee) will not change the consistency between our theoretical results and the simulations under the same assumptions. We have now re-emphasised the purpose of these simulations, while underlining that a discrepancy with our empirical results will not impact our conclusions. We added the following sentences to Supplementary Note 4: "These simulations use an arbitrary number of SNPs included in the PGS and are not designed to match the number of SNPs used in various PGS presented in the main text. Nevertheless, our conclusions are general and applicable to empirical data under the assumption that each SNP in the PGS contributes about the same amount of genetic variance."

Supplementary information

Please, read carefully. URLs are missing (P42, P43, etc), some of the legends are unclear and not self-contained. For instance: ST1, has a cohort with $N=5$.

We thank the reviewer for noticing this error and have corrected the value for the sample size accordingly.

ST4 9,863 COJO SNPs, but main text talks about marginal effects. Which one is it?

We report all COJO SNPs in Supplementary Tables ST4 – ST9. However, some analyses in the main text (e.g., correlation of SNP effects across ancestries or for replication analyses) are based upon a subset of these COJO SNPs, which marginal effect (i.e., from fitting 1 SNP at a time) also reaches genome-wide significance (i.e. $P_{\text{GWAS}} < 5E-8$). Comparison of marginal effects are easier to interpret because of LD differences between populations.

ST5, LD with next SNP, what SNP, what LD measure? In what population was estimated? With what sample size was estimated?

We apologise for the lack of clarity. We have now clarified this in the table legend. The column flagged by the reviewer refers to the sample correlation of allele count between the focal COJO SNP and the one upstream. This correlation is estimated in the corresponding LD reference used for the COJO analysis. We also highlight the sample size of the LD reference in the legend.

ST11. What is the length group?

We grouped genes into length group (i.e. genes of similar length are grouped together) to sample the null distribution used to generate Suppl. Fig. 20.

ST12. Variance explained $2p(1-p)bxJ$? I imagine the J is a typo.

No this is not a typo. The justification for this is given in Yang et al. (2012), which introduced the COJO methodology.

SF19 has two vertical lines, but only one is described.

We now describe the grey vertical line, which corresponds to $R^2(\text{SNP}) = 0.5 * h^2$ (i.e. PGS explains half the total heritability).

ST16. I'd keen to hear how the authors interpret that an eQTL in the skin or salivary gland might mediate or be linked to height. Perhaps, a bit of discussion in the MS would be welcomed.

We thank the referee for this comment. We agree that this observation may be somewhat counterintuitive at first glance. Previous studies have found that the genetic control of gene expression can be shared across tissues (e.g., Qi et al. Nat. Comm. 2018; GTEx consortium, Science 2020). In other words, a variant controlling the expression of a given gene in one tissue may also be an eQTL for the same gene in another tissue. Therefore, our results do not imply that salivary gland and skin are the causal tissues for height but may still be relevant to prioritise genes. We previously

wrote in the main text: “Note that saturation of SMR genes is partly affected by the biological relevance and statistical power of eQTL studies.” “Biological relevance” in that sentence alluded to the comment made by the referee. We have now rephrased and complemented the sentence as (L358-360): “Note that saturation of SMR genes is partly affected by the statistical power of current eQTL studies, which do not always survey biologically relevant tissues and cell types for height.”

In my view, if the authors wish to uphold the best standards and lead the way, they should consider releasing the following data to the research community:

1. All results for imputed HM3 SNPs per cohort (rsID, beta, SE(beta), Ref Allele, N, Imputation Accuracy, etc) . Cohorts included in the meta-analyses and replication.
2. All summary statistics for all HM3 SNPs from the GWAS meta-analyses (FE, EUR, EAS, HIS, AFR, SAS).

We agree with the referee on the importance of sharing GWAS summary statistics and have done so in previous efforts. However, a substantial fraction of the data used in this study were provided (under a specific agreement) by 23andMe Inc. This agreement prevents us from sharing all the data requested by the referee, as always seen in other published GWAS papers incorporating 23andMe data. Also, sharing data for individual cohorts is covered by study-specific agreements. Nevertheless, we will make the following datasets available upon publication of our manuscript (should it be accepted for publication):

- 1) HM3 SNPs Summary statistics for multi-ancestry and each ancestry-specific GWAS meta-analyses excluding data from 23andMe (N=2M in total)
- 2) COJO results for each ancestry group + from the cross-ancestry meta-analysis (Supp. Tables 4 - 9)
- 3) SNP weights for 5 polygenic scores using all HM3 SNPs derived from our 5 largest GWAS meta-analyses (incl. 23andMe)

G. References are largely OK, but they missed some of the references to prediction of height and transferability to other populations in UKB.

We thank the referee for this comment. We have now increased the number of relevant references. For example, Bitarello et al. G3 2020 & Prive et al. AJHG 2022 for transferability and Lello et al. Genetics 2018 and Marquez-Luna et al. Nat. Comm 2022 for prediction.

H. The manuscript is clear.

We thank the referee for this comment.

Referee #3 (Remarks to the Author):

In this manuscript, Yengo et al. conduct the largest GWAS to date on height, a well-established trait representative of a highly polygenic genetic architecture. The manuscript is well written and the analyses are appropriate and comprehensive. Notably, the authors include individuals from diverse ancestries, and are thoughtful in their discussion of cross-ancestry findings. A prime finding of the manuscript is with respect to saturation of GWAS loci for height, namely they did not find uniform coverage across the entire genome at this massive sample size, but that top loci are clustered in functionally relevant regions that explain effectively all heritable variation (at least in EUR). This is to my knowledge the most direct test of this saturation phenomenon to date, and speaks to the likely trends in genetic architecture for complex traits at large. In this manuscript the authors also demonstrate the utility of combining familial information and PGS in risk prediction, as well as demonstrate that smaller GWAS can implicate the correct functional annotation categories sooner than reaching variant level saturation. Finally, they show the benefit of meta-analyzing results from across ancestry groups for the generation of better performing PGS for diverse populations. Overall I was very impressed with this paper and think it will have a large impact on the GWAS community.

We thank Reviewer #3 for the detailed and accurate summary of our work and for acknowledging the quality of our manuscript.

I do, though, have a few suggestions that I hope may strengthen it even further:

We thank Reviewer #3 for their suggestions to strengthen our study. We address each of their points below.

1. The investigation into residual population stratification in the EUR results is interesting and well warranted given prior findings regarding confounding of height GWAS in Europe, as the authors discuss. Given the large sample sizes of the non-EUR ancestry groups, it would also be well worth a check that population stratification isn't affecting the results in those GWAS as well. Despite the groups being of much relative smaller size than EUR, they are still among the largest GWAS of diverse cohorts yet published. Additionally, many of these non-EUR groups also have high levels of admixture, making PS particularly important to control for in them. I do not think that as comprehensive a test as what was done in EUR is necessary for these other groups, but some further due diligence to confirm that PS isn't driving some signals (especially considering the authors highlight the novel contributions from other ancestries) is warranted, e.g. one of the strategies the authors describe in Supplementary Note 1.

We thank the referee for this point, which has also been raised by Reviewer #1. We have expanded Supplementary Note 2, which now presents additional assessment of residual population stratification in all GWAS meta-analyses of non-EUR cohorts. Overall, we find some evidence of residual population stratification in our non-EUR GWAS analyses, but the magnitude is much smaller than in previous GWAS of height in EUR (e.g., Wood et al. 2014). We refer Reviewer #3 to our more detailed response to Reviewer #1 above and to extended Suppl. Note 2.

1a. Related, I was surprised to see no PC plots in this manuscript. Some additional supplemental figures with the PCs for the different ancestry groups would be useful for context as to the composition of these groups.

Following the referee's suggestion, we now provide PCA plots (shown below) showing ancestry similarities and differences between cohorts included in our meta-analyses alongside 26 sub-populations from the 1000 Genomes project (Suppl. Fig. 2) Overall, these plots show a high degree of homogeneity within the 5 ancestry groups included in our meta-analysis and their clear concordances with the corresponding 1000 Genomes sub-populations.

Suppl. Fig. 2. (PCA analysis of cohorts included in the meta-analysis).

2. The authors justify their use of a EUR LD reference panel for COJO analysis, noting that it appears to ‘reasonably approximate’ the LD observed in their cross-ancestry meta-analysis. Would findings be improved by including some diverse participants in the generation of an LD reference panel? Though your sample is indeed driven by the large number of EUR individuals, including representatives from other ancestries in the generation of the LD scores may aid results contributed by other ancestry groups.

We thank the referee for this suggestion. Accordingly, we re-analysed summary statistics from our cross-ancestry GWAS meta-analysis using two LD reference sets. First, we randomly selected 37900

EUR, 4400 EAS, 4250 HIS, 2750 AFR and 700 SAS individuals (i.e. 50000 individuals in total) to form a LD reference set with ancestries proportions matching that in our cross-ancestry meta-analysis. The second set contained 50000 individuals with EUR ancestries. We restricted analyses with both LD reference sets to 882,755 HM3 SNPs passing quality control (Hardy-Weinberg Equilibrium test, missingness and imputation quality) in all 5 ancestry groups.

We found that COJO based on the multi-ancestry LD panel detected 3635 (3380 using a collinearity-threshold of 0.1) independent associations vs 11,001 associations using the EUR LD reference. The latter number is much smaller than the 12,111 reported in the main text but consistent with a ~10-15% smaller number of HM3 SNPs used as input. We also repeated these analyses only using the 37900 EUR individuals as LD reference and found that COJO detect 11,065 SNPs. Altogether, these results demonstrate that COJO with a composite LD reference panel does not improve the detection of associations in our cross-ancestry GWAS meta-analysis. We emphasize that extending the COJO methodology for analysing multi-ancestry GWAS is an independent research question, beyond the scope of our study. We now describe these additional results in Supplementary Note 1.

3. Figure 1, the 'Brisbane plot' is an intriguing way to visualize the GWAS results seen here, and I would bet this presentation will become widely used as GWAS sample sizes continue to grow. I imagine the authors made an intentional decision not to include standard Manhattan plots in this manuscript (likely overloaded with results) but some way to highlight the loci contributed by each ancestry group would be interesting to see.

We thank the referee for this comment. We now highlight 24 / 12,111 SNPs which only reached genome-wide significance (marginally or jointly) in non-EUR and were independent of signals detected in our EUR GWAS. These 24 SNPs are listed among the 131 non-EUR specific signals reported in ST9. The remaining $131-24=107$ SNPs did not overlap with the 12,111 SNPs selected after applying COJO to our cross-ancestry signals because either a proxy SNP was selected instead or their effects were masked in the meta-analysis.

Fig. 2.

4. This reviewer could not find a description of the imputation procedure done on the component cohorts. Relevantly, Supplementary Figure 2 shows strikingly how much more poorly captured less common variants are in AFR. Though this site frequency spectrum is likely driven by the genotype arrays used to generate data, it may be improved with the inclusion of more diverse participants in the imputation reference panel for these cohorts. At least a discussion of imputation procedures done in the various cohorts and the reference panels used should be added.

A description of the imputation procedure (pre- and post-imputation QC + software used) is provided in Supp. Table 2.

5. Additionally, though the authors do limit to HM3 SNPs that should be quite well behaved, it may be useful to confirm that the reasonably permissive INFO threshold of 0.3 is appropriate for all cohorts here. This is related to the concerns over population stratification noted above (e.g. if imputation is performing worse in some ancestries than others).

We thank the referee for this question. We have added a new panel to (Supp. Fig 3) showing the distribution of imputation accuracy (averaged for each SNP within ancestry groups) across HM3. Overall, we confirm the HM3 SNPs are well imputed in all ancestry groups with >98% of them having an imputation accuracy >0.3 and >83% an accuracy >0.9.

Suppl. Fig. 3. Frequency and Imputation Accuracy Distribution of HapMap3 SNPs

6. The authors comment on their analyses here for height speaking to the broader implications for saturation as it pertains to other complex traits. Height is extremely polygenic even among complex traits – could the authors comment on the likely saturation thresholds for traits with other levels of polygenicity?

We thank the referee for this question. We have now added the following paragraph, which underlines some of the challenges of predicting saturation thresholds and makes a few predictions for other complex traits and diseases (L456-469): “Recent studies using UKB data predicted that GWAS sample sizes just over 3M individuals are required to identify 6000-7000 GWS SNPs explaining >90% of the SNP-based heritability of height.⁵⁰ We showed empirically that these predictions are downwardly biased given that ~10,000 independent associations are, in fact, required to explain 80-90% of the SNP-based heritability of height in EUR individuals. Discrepancies between observed and predicted levels of saturation could be explained by multiple factors such as (i) heterogeneity of SNP effects between cohorts and background ancestries, which may have reduced the statistical power of our study as compared to a homogenous sample like UKB, (ii) inconsistent definitions of GWS SNPs (using COJO in this study vs. standard clumping in ref.⁵⁰), and, most importantly (iii) misspecification of the SNP-effects distribution assumed to make these predictions. Nevertheless, if these predictions reflect proportional levels of saturation between traits, then we could expect that 2- to 10-fold larger samples would be required for GWAS of inflammatory bowel disease (×2, i.e. N= 10M), schizophrenia (×7; i.e. N=35M) or BMI (×10; i.e. N=50M) to reach a similar saturation of 80-90% of SNP-based heritability.”

7. Only GWS SNPs appear to have been used in the PGS analyses. Some justification for this choice would be helpful, and even better to see the PGS R2 at different p value thresholds, as is commonly presented for this type of analysis.

We thank the referee for this suggestion. We now present predictions based on a PGS including all HM3 SNPs (PGS weights calculated using the SBayesC method) as well as various prediction analyses using PGS based on COJO SNPs identified at lower significance thresholds (5e-7, 5e-6 and 5e-5). These results are described in the main text (L258-293), in Fig. 4 below

Fig. 4. Accuracy of various genetic predictor of height

and in Suppl. Fig. 26. below

Suppl. Fig. 26. Accuracy of PGS of height based on SNPs ascertained various significance thresholds

8. Given the unprecedented size/completeness of this GWAS, I would love to see the distribution of effect sizes as they relate to MAF. This would be a useful figure to add that would speak more broadly to what empirical data is bearing out with respect to these two variant dimensions.

We thank the referee for this question. We now provide this Figure in the main text as Fig. 1. Fig. 1 below shows the relationship between MAF in our cross-ancestry meta-analysis and joint SNP effects from COJO applied to summary statistics from METAFE.

Fig. 1. Relationship between frequency and effect sizes of 12,111 genome wide significant SNPs.

Reviewer Reports on the First Revision:

Referees' comments:

Referee #1 (Remarks to the Author):

The authors have put in substantial effort to revise the paper and clarify the reviewer concerns. I am additionally curious for some further clarification around the results based on a comment from Review #3 suggesting using a multi-ancestry LD reference panel with the COJO results. The authors found that using the multi-ancestry LD reference panel detected substantially less independent associations and claim that the multi-ancestry LD panel leads to underestimation by COJO of the number of associations. But, why isn't the EUR only reference panel leading to an overestimation of the number of independent associations? If you used a AFR LD panel, do you still have ~11K independent SNPs? Can you expand on this a little more?

Referee #2 (Remarks to the Author):

The authors have addressed all my comments. I have nothing else to add. Nice work.

Referee #3 (Remarks to the Author):

Prior summary:

In this manuscript, Yengo et al. conduct the largest GWAS to date on height, a well-established trait representative of a highly polygenic genetic architecture. The manuscript is well written and the analyses are appropriate and comprehensive. Notably, the authors include individuals from diverse ancestries, and are thoughtful in their discussion of cross-ancestry findings. A prime finding of the manuscript is with respect to saturation of GWAS loci for height, namely they did not find uniform coverage across the entire genome at this massive sample size, but that top loci are clustered in functionally relevant regions that explain effectively all heritable variation (at least in EUR). This is to my knowledge the most direct test of this saturation phenomenon to date, and speaks to the likely trends in genetic architecture for complex traits at large. In this manuscript the authors also demonstrate the utility of combining familial information and PGS in risk prediction, as well as demonstrate that smaller GWAS can implicate the correct functional annotation categories sooner than reaching variant level saturation. Finally, they show the benefit of meta-analyzing results from across ancestry groups for the generation of better performing PGS for diverse populations. Overall I was very impressed with this paper and think it will have a large impact on the GWAS community.

I find the statistical tests conducted by the authors to be appropriate, comprehensive, and well presented. In their revisions, the authors have thoroughly addressed my comments and concerns. Specifically they have now:

- Addressed the question of residual PS in non-EUR populations, as was brought up by myself and another reviewer
- Included numerous additional figures, including perhaps most significantly the relationship

between MAF and effect size (very interesting and broadly informative!), as well as PCA plots, an updated Brisbane plot highlighting contributions from non-EUR ancestry groups, and the imputation accuracy for different populations.

- Tested whether including more diverse populations in LD panel generation improves their COJO results, finding that “COJO with a composite LD reference panel does not improve the detection of associations in our cross-ancestry GWAS meta-analysis.”
- Described the implications of their findings for other complex traits with different levels of polygenicity
- Conducted PGS with a variety of different methodologies and R2 thresholds

In sum, I remain highly impressed with this manuscript and am satisfied with the changes that the authors have done during this revision.

Author Rebuttals to First Revision:

Referee #1 (Remarks to the Author):

The authors have put in substantial effort to revise the paper and clarify the reviewer concerns.

We thank Reviewer #1 for acknowledging the improvement of our revised manuscript.

I am additionally curious for some further clarification around the results based on a comment from Review #3 suggesting using a multi-ancestry LD reference panel with the COJO results. The authors found that using the multi-ancestry LD reference panel detected substantially less independent associations and claim that the multi-ancestry LD panel leads to underestimation by COJO of the number of associations. But, why isn't the EUR only reference panel leading to an overestimation of the number of independent associations? If you used a AFR LD panel, do you still have ~11K independent SNPs? Can you expand on this a little more?

We thank Reviewer #1 for these additional questions, which give us the opportunity to clarify our findings. To answer the first referee's question, we ran a COJO analysis of our cross-ancestry GWAS using LD information from 10,636 AFR individuals (i.e. same LD panel as in our AFR GWAS meta-analysis). Note that 242,891 SNPs were filtered out by GCTA prior to analysis because of expected large differences in allele frequencies between our cross-ancestry GWAS (incl. >75% of EUR individuals) and the AFR LD panel (by default GCTA excludes SNPs with an absolute frequency difference >0.1). Nevertheless, we detected 5,701 quasi-independent joint associations, explaining 24.6%, 13.2%, 10.9% and 3.5% of height variance in EUR, SAS, EAS and AFR individuals respectively.

To answer the second question asked by the referee, we compared different sets of COJO SNPs using the prediction accuracy of PGSs calculated from their estimated joint effects. Across ancestries, we found that the PGS based on 11,001 COJO SNPs detected using a EUR LD panel explains a significantly larger amount of height variance than that of the PGS based on only 3,380 COJO SNPs detected with a multi-ancestry panel (EUR: 38.2% vs 26.4%; SAS: 20.3% vs 13.4%; EAS: 19.5% vs 13.3% and AFR: 9.0% vs 5.0%). In other words, the set of 11,001 COJO SNPs contains additional information not captured by the 3,380 COJO SNPs, which implies that using a mix-ancestry panel misses relevant associations and, therefore, underestimates the actual number of signals that can be detected.

We have now added the following sentences to Suppl. Note 1.

“This conclusion is supported by the fact that a PGS based on 11,001 COJO SNPs detected using a EUR LD panel explains a significantly larger amount of height variance than that of a PGS based on only 3,380 COJO SNPs detected with a multi-ancestry panel (EUR: 38.2% vs 26.4%; SAS: 20.3% vs 13.4%; EAS: 19.5% vs 13.3% and AFR: 9.0% vs 5.0%). Moreover, we ran another COJO analysis of our cross-ancestry GWAS using LD information from 10,636 AFR individuals (i.e. same LD panel for our AFR GWAS meta-analysis). Note that 242,891 / 882,755 (i.e. 27.5%) SNPs were filtered out by GCTA prior to analysis because of expected large differences in allele frequencies between our cross-ancestry GWAS including >75% of EUR individuals and the AFR LD panel (by default GCTA exclude SNPs with an absolute frequency difference is >0.1). Nevertheless, we detected 5,701 quasi-independent joint associations (i.e. more associations than using a mix-ancestry panel), explaining 24.6%, 13.2%, 10.9% and 3.5% of height variance in EUR, SAS, EAS and AFR individuals respectively. The latter predictive performances are lower to that obtained with a PGS from 3,380 COJO SNPs.”

Referee #2 (Remarks to the Author):

The authors have addressed all my comments. I have nothing else to add. Nice work.

We thank Reviewer #2 for acknowledging the quality of our revised manuscript and again for their previous suggestions, which have contributed to improving the overall quality of our study.

Referee #3 (Remarks to the Author):

I find the statistical tests conducted by the authors to be appropriate, comprehensive, and well presented. In their revisions, the authors have thoroughly addressed my comments and concerns.

Specifically they have now:

- Addressed the question of residual PS in non-EUR populations, as was brought up by myself and another reviewer
- Included numerous additional figures, including perhaps most significantly the relationship between MAF and effect size (very interesting and broadly informative!), as well as PCA plots, an updated Brisbane plot highlighting contributions from non-EUR ancestry groups, and the imputation accuracy for different populations.
- Tested whether including more diverse populations in LD panel generation improves their COJO results, finding that “COJO with a composite LD reference panel does not improve the detection of associations in our cross-ancestry GWAS meta-analysis.”
- Described the implications of their findings for other complex traits with different levels of

polygenicity

- Conducted PGS with a variety of different methodologies and R2 thresholds

In sum, I remain highly impressed with this manuscript and am satisfied with the changes that the authors have done during this revision.

We thank Reviewer #3 for the detailed and accurate summary of our work and for acknowledging the quality of our revised manuscript.